

# Energy magnetization and transport in systems with a non-zero Berry curvature in a magnetic field

Archisman Panigrahi[1,2] and Subroto Mukerjee[1*]

**1** Department of Physics, Indian Institute of Science, Bangalore 560012, India
**2** Department of Physics, Massachusetts Institute of Technology, Cambridge, MA 02139, USA

* smukerjee@iisc.ac.in

## Abstract

We demonstrate that the well-known expression for the charge magnetization of a sample with a non-zero Berry curvature can be obtained by demanding that the Einstein relation holds for the electric transport current. We extend this formalism to the transport energy current and show that the energy magnetization must satisfy a particular condition. We provide a physical interpretation of this condition, and relate the energy magnetization to circulating energy currents in Chern insulators due to chiral edge states. We further recover the expression for the energy magnetization with this alternative formalism. We also solve the Boltzmann Transport Equation for the non-equilibrium distribution function in 2D for systems with a non-zero Berry curvature in a magnetic field. This distribution function can be used to obtain the regular Hall response in time-reversal invariant samples with a non-zero Berry curvature, for which there is no anomalous Hall response.

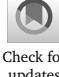

# 1   Introduction

The effect of the Berry curvature on thermoelectric transport in crystals has received a lot of attention in recent years. Following the seminal work of Berry on the adiabatic evolution of quantum states [1], it was realized that the geometric phase identified by him has important consequences for the semiclassical dynamics of electron wavepackets in crystals. In particular, the Berry curvature associated with this phase contributes an anomalous term to the velocity of the wavepacket [2–7]. This anomalous velocity can give rise to a Hall response without an external magnetic field, resulting in the anomalous Hall effect [8], the Magnus Hall Effect [9–11], as well as many other interesting electronic transport phenomena [12]. Similarly, in response to a temperature gradient, without any external magnetic field, it generates a transverse Hall voltage, known as the anomalous Nernst effect [9,13–15]. Coupled with the Boltzmann transport theory, the modified semiclassical equations have been employed to study transport in topological insulators [16,17], Chern Insulators [18], Weyl Semi-Metals [14,15, 19–25], Kondo Insulators [26], Rashba systems [27,28], triple-component Fermionic systems [29], optical lattices and quasicrystals [30,31], superconductors [32], non-Hermitian systems [33,34], as well as in various other systems [35–41]. Non-linear effects in transport have also been studied within this formalism [42–51].

In the linear response regime of thermoelectric transport, the electric current and heat current are given by the expressions,

$$
\begin{aligned}
j^e &= \overleftrightarrow{L}_{11} \cdot \left( E + \frac{\nabla\mu}{e} \right) + \overleftrightarrow{L}_{12} \cdot (-\nabla T), \\
j^Q &= \overleftrightarrow{L}_{21} \cdot \left( E + \frac{\nabla\mu}{e} \right) + \overleftrightarrow{L}_{22} \cdot (-\nabla T),
\end{aligned}
\tag{1}
$$

where the heat current $\boldsymbol{j}^Q$ is related to the energy current $\boldsymbol{j}^E$ and the number density current $\boldsymbol{j}^N$ by the relation $\boldsymbol{j}^Q = \boldsymbol{j}^E - \mu \boldsymbol{j}^N$ [52], $\mu$ being the chemical potential, and the electric current $\boldsymbol{j}^e$ is related to the number density current $\boldsymbol{j}^N$ by the relation $\boldsymbol{j}^e = -e\boldsymbol{j}^N$, where $-e$ is the electronic charge. Here $\overset{\leftrightarrow}{L}_{11}$ is the electric conductivity tensor, $\overset{\leftrightarrow}{L}_{22}$, the thermal conductivity tensor,[1] and $\overset{\leftrightarrow}{L}_{12}$ and $\overset{\leftrightarrow}{L}_{21}$ are the Peltier conductivity coefficients. The coefficients $\overset{\leftrightarrow}{L}_{12}$ and $\overset{\leftrightarrow}{L}_{21}$ are related via the Onsager relation [53–55],

$$\overset{\leftrightarrow}{L}_{21} = T\, \overset{\leftrightarrow}{L}_{12} \ . \tag{2}$$

It is important to note that the Onsager relation holds only for the coefficients that relate *transport* currents to the applied gradients and not the *total* currents. The total currents for systems with broken time reversal symmetry typically have diamagnetic contributions, which need to be subtracted in order to obtain the transport currents [55]. This has been explicitly demonstrated for systems with a non-zero Berry curvature [12, 56].

An additional set of relations are the Einstein relations which require the transport currents generated by an electric field $\boldsymbol{E}$ to be equal to those due to a chemical potential gradient $\boldsymbol{\nabla}\mu$ of strength $e\boldsymbol{E}$ [55,56]. It is well known that the bound electric current in a magnetic sample can be expressed as the curl of the charge magnetization [57]. Similarly, there can be circulating energy currents in a sample, which can be expressed as the curl of a quantity called the energy magnetization [55]. It has been demonstrated that the Einstein relation holds for the electric, heat, and energy currents in systems with a non-zero Berry curvature [56, 58], by employing various interesting techniques like introduction of a fictitious inhomogeneous disorder field. However, as shown by Onsager [53, 54], the Einstein relation should always hold due to the principle of detailed balance, and should not depend on the microscopic details of the sample. In this paper, we show that assuming the Einstein relation holds, one can conveniently find the forms of the charge magnetization and the energy magnetization, without introducing an inhomogeneous disorder field, as was done in [58]. Here we show that the energy magnetization has to satisfy a certain specific condition, and provide an interpretation of this condition for a Chern insulator in terms of the number of chiral edge modes.

As will be shown in section 4, the transport currents contain pieces which depend on the equilibrium and the non-equilibrium distribution functions. The equilibrium distribution function (Fermi function) contributes to the intrinsic anomalous Hall and Nernst responses, which occur without any external magnetic field, for a system with non-zero Berry curvature. The non-equilibrium distribution function, which is calculated in section 4 up to leading order in the external potential and temperature gradients for a two-dimensional system, is responsible for the regular Hall and Nernst responses and captures the effects of Berry curvature on these. Here by the term "regular Hall" response, we mean the Hall response due to a magnetic field, and we *do not* specifically mean the linear Hall response. Similar expressions for the non-equilibrium part of the distribution were obtained in several papers [21, 59, 60], but only in the context of chiral magnetic effects, which are absent in two dimensions. When the Berry curvature $\Omega = 0$, the non-equilibrium parts of the distributions obtained in the above-mentioned papers only lead to regular Ohmic transport, but not the regular Hall effect. Neglecting parts of the non-equilibrium distribution function is justified for systems like Weyl semimetals, where the anomalous Hall effect is much stronger than the regular Hall effect.

---

[1]Here the thermal conductivity tensor is denoted as the heat current per unit temperature gradient at zero electric field or zero chemical potential gradient. Sometimes, thermal conductivity is denoted as the heat current per unit temperature gradient when the net electric current is zero (an internal electrochemical potential gradient is generated to maintain zero electric current, but that gradient in turn contributes to the heat current). In that case, it can be shown that (see Eq. (13.56) of Ref. [52]) that the heat conductivity is $\overset{\leftrightarrow}{L}_{22} - \overset{\leftrightarrow}{L}_{21} (\overset{\leftrightarrow}{L}_{11})^{-1} \overset{\leftrightarrow}{L}_{12}$. But this additional term is a small correction of the order $\left(\frac{k_B T}{\varepsilon_F}\right)^2$.

However, there are systems (e.g., bilayer graphene) with non-zero Berry curvature, which do not show the anomalous Hall effect due to intrinsic time reversal symmetry. In such samples, the Berry curvature modifies the regular Hall effect, which can be calculated with the solution (Eq. (26)) of the Boltzmann Transport Equation obtained in this paper. Note that there is no fundamental change the regular Hall response. Rather, the existing response is modified due to the Berry curvature.

There are two main results of this paper. The first, obtained through the calculations of sections 2-3, is a derivation of the charge and energy magnetization assuming the validity of the Einstein relation, and the interpretation of a condition on the energy magnetization. While the expression for the energy magnetization has been obtained previously from microscopic considerations, our derivation is based on general arguments involving the Einstein relation. This, in our opinion, simplifies the physical understanding of the energy magnetization, which is a rather opaque quantity. The second result, obtained in section 4, is a complete solution of the Boltzmann Transport Equation in two-dimensions in the linear response regime, which can be used to calculate transport currents for a two-dimensional system with a non-zero Berry curvature and in the presence of a magnetic field.

The paper is organized as follows: We describe the overall formalism of calculating currents from the semiclassical equations in section 2, and further discuss how the orbital magnetization affects the electric and energy currents. While the effects of orbital magnetization on these currents have been addressed in the literature before [43, 56], we emphasize some of the salient aspects of the physics which are important for us to derive the central results of our paper described in Secs. 3.2 and 4. We first illustrate how the validity of the Einstein relation for the charge and energy currents can be exploited to obtain expressions for the charge and energy magnetizations. While the expressions for these magnetizations have been obtained from microscopics recently [58, 61], our approach has the virtue of simplicity. Our focus is the energy magnetization but as a warm-up, we first employ our method in section 3.1, to obtain the known expression for the more commonly studied, charge magnetization. Section 3.2 contains one of the two central results of this paper. Here we employ the same method as for the charge magnetization to find a condition that the energy magnetization has to obey, which has a physical intuitive interpretation. We then use it to find the expression of the energy magnetization. In section 4, we derive the complete expressions for the transport heat current density, and solve the Boltzmann transport equation for the electron distribution function up to linear order in the potential and temperature gradient in the presence of a magnetic field and non-zero Berry curvature. As mentioned above, the distribution function can be used to obtain the regular Hall and Nernst responses.

## 2 Formalism

The semiclassical equations of motion for the position and crystal momentum of a Bloch wavepacket are [3, 4, 7]

$$\dot{r} = \frac{1}{\hbar} \frac{\partial \varepsilon_k}{\partial k} - \dot{k} \times \Omega(k),$$

$$\hbar \dot{k} = -e(E + \dot{r} \times B).$$

(3)

Here, $\Omega(k) = i \langle \nabla_k u_k | \times | \nabla_k u_k \rangle$ is the Berry curvature in the reciprocal space, and $u_k$ is the periodic part of the Bloch wavefunction. The energy eigenvalues are modified due to the orbital magnetic moment [56] $m_k$ of an wavepacket, as well as Zeeman splitting,

$$\varepsilon_k = \varepsilon_0(k) - m_k \cdot B - m_s \cdot B,$$

(4)

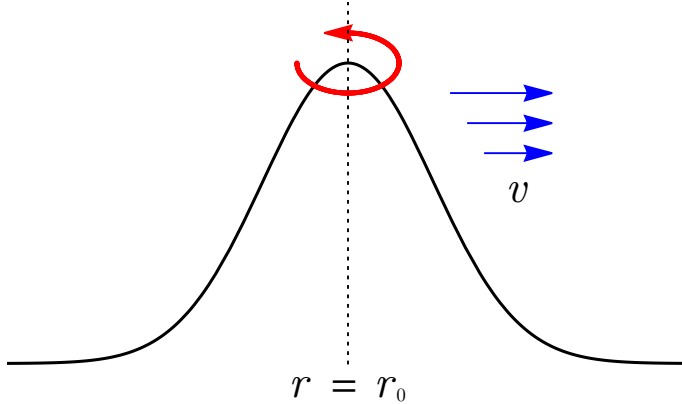

Figure 1: In addition to its velocity, a Bloch wavepacket can have an angular momentum about its center, which gives rise to an orbital magnetic moment.

where $\varepsilon_0(\mathbf{k})$ is the original band structure energy at zero magnetic field, and $\mathbf{m}_s$ is the spin magnetic moment. Here, $\varepsilon_k$ is a function of the spin $s$, but that is not explicitly written due to notational convenience.

It has been shown that the equations of motion (Eq. (3)) violate Liouville's theorem, and for a phase space volume element $\Delta V$, the quantity $\Delta V \left(1 + \frac{e}{\hbar}\mathbf{B} \cdot \mathbf{\Omega}(\mathbf{k})\right)$ remains a constant of motion [62]. As a result, the calculation of the expectation value of any operator $\hat{\mathcal{O}}$ over all states requires the introduction of an additional factor of $\left(1 + \frac{e}{\hbar}\mathbf{B} \cdot \mathbf{\Omega}(\mathbf{k})\right)$ in the integrand,

$$\langle \hat{\mathcal{O}} \rangle = \sum_s \int \frac{d\mathbf{k}}{(2\pi)^d} \left(1 + \frac{e}{\hbar}\mathbf{B} \cdot \mathbf{\Omega}(\mathbf{k})\right) \langle \hat{\mathcal{O}} \rangle_k \tilde{f}_k. \tag{5}$$

The sum $\sum_s$ accounts for the two spin species. In the above expression, $\tilde{f}_k$ has implicit dependence on spin due to the Zeeman splitting. Here, $\tilde{f}_k$ is the distribution function and in thermodynamic equilibrium, it reduces to the Fermi distribution function, $f_k = \frac{1}{e^{\beta(\varepsilon_k - \mu)} + 1}$. When an external electromagnetic field, chemical potential gradient or temperature gradient is applied, the distribution function is modified to $\tilde{f}_k = f_k + g_k$, where $g_k$ is the non-equilibrium contribution. We calculate $g_k$ for a two dimensional system within the Boltzmann transport formalism in Section 4. Up to linear order in the external fields, $g_k$ contributes to regular Ohmic conduction, the regular Hall effect, and the regular Nernst effect. In the presence of a non-zero Berry curvature, the equilibrium part of the distribution contributes to the anomalous Hall effect and the anomalous Nernst effect, up to linear order in the external electric field, temperature gradient, and chemical potential gradient.

Let us briefly review the calculation of the orbital magnetic moment of a Bloch wavepacket, which is responsible for the circulating magnetization energy currents and electric currents. While a Bloch wavepacket [4,7] is localized at a point (say, $\mathbf{r}_0$), the electron is not necessarily localized there, and can have an angular momentum due to the motion about the center of the wavepacket (see Fig. 1), giving rise to an orbital magnetic moment given by the expression [56,63–65],

$$\mathbf{m}_k = -\frac{e}{2m} \left\langle \psi_{k,r_0} \left| (\hat{\mathbf{r}} - \mathbf{r}_0) \times \hat{\mathbf{p}} \right| \psi_{k,r_0} \right\rangle. \tag{6}$$

Note that $m$ is the bare electron mass, not the effective mass of the Bloch state. Since the Bloch wavepacket is not completely localized, the wavepacket centered at one point, can contribute to the currents at another point, and the orbital magnetic moment is involved in these parts of the electric and energy currents. The *total* electric current density [12,40] and energy current

density [56, 58] are given by,

$$j^e_{\text{total}} = \sum_s \left[ -e \int \frac{d\boldsymbol{k}}{(2\pi)^d} \left[ g_{\boldsymbol{k}} \frac{1}{\hbar} \frac{\partial \varepsilon_{\boldsymbol{k}}}{\partial \boldsymbol{k}} + f_{\boldsymbol{k}} \frac{e}{\hbar} (\boldsymbol{E} \times \boldsymbol{\Omega}(\boldsymbol{k})) \right] + \boldsymbol{\nabla} \times \int \frac{d\boldsymbol{k}}{(2\pi)^d} f_{\boldsymbol{k}} \boldsymbol{m}_{\boldsymbol{k}} \left( 1 + \frac{e}{\hbar} \boldsymbol{B} \cdot \boldsymbol{\Omega}(\boldsymbol{k}) \right) \right], \tag{7a}$$

$$j^E_{\text{total}} = \sum_s \left[ \int \frac{d\boldsymbol{k}}{(2\pi)^d} \varepsilon_{\boldsymbol{k}} \left[ g_{\boldsymbol{k}} \frac{1}{\hbar} \frac{\partial \varepsilon_{\boldsymbol{k}}}{\partial \boldsymbol{k}} + \frac{e}{\hbar} f_{\boldsymbol{k}} (\boldsymbol{E} \times \boldsymbol{\Omega}(\boldsymbol{k})) \right] - \boldsymbol{E} \times \int \frac{d\boldsymbol{k}}{(2\pi)^d} f_{\boldsymbol{k}} \boldsymbol{m}_{\boldsymbol{k}} \left( 1 + \frac{e}{\hbar} \boldsymbol{B} \cdot \boldsymbol{\Omega}(\boldsymbol{k}) \right) \right.$$
$$\left. + \boldsymbol{\nabla} \times \int \frac{d\boldsymbol{k}}{(2\pi)^d} f_{\boldsymbol{k}} \varepsilon_{\boldsymbol{k}} \left( \frac{\boldsymbol{m}_{\boldsymbol{k}}}{-e} \right) \left( 1 + \frac{e}{\hbar} \boldsymbol{B} \cdot \boldsymbol{\Omega}(\boldsymbol{k}) \right) \right]. \tag{7b}$$

See Appendix A for a self-contained derivation of how the orbital magnetic moment appears in Eq. (7a). Similar calculations produce the terms involving orbital magnetization in Eq. (7b). In both Equations (7a) and (7b), the terms in the first lines, those does not involve the orbital magnetization $\boldsymbol{m}_{\boldsymbol{k}}$, can be considered as the contribution due to the movement of the center of the Bloch wavepacket, while the other terms are the contribution of the movement of the electron about the center of the wavepacket.

## 3 Charge and energy magnetization from the Einstein relation

### 3.1 Charge magnetization

In this section we demonstrate that the known expression of the charge magnetization can be recovered by demanding that the Einstein relation holds for the electric transport current. The electric transport current can be obtained by subtracting the charge magnetization current from the total electric current. The charge magnetization current is the curl of the charge magnetization, whose expression has been obtained in Ref. [56]. Here we show that the same expression can be obtained by using this alternative formalism. In the next section, we will employ the same strategy to find the expression for the energy magnetization (whose curl is the bound energy current). The expression for the electric magnetization current is

$$j^e_M = \boldsymbol{\nabla} \times \boldsymbol{M}^e = \boldsymbol{\nabla} \mu \times \frac{\partial \boldsymbol{M}^e}{\partial \mu} + \boldsymbol{\nabla} T \times \frac{\partial \boldsymbol{M}^e}{\partial T}. \tag{8}$$

Consequently, we can calculate the transport current,

$$j^e_{\text{transport}} = j^e_{\text{total}} - j^e_M$$
$$= \sum_s \left[ -e \int \frac{d\boldsymbol{k}}{(2\pi)^d} \left[ g_{\boldsymbol{k}} \frac{1}{\hbar} \frac{\partial \varepsilon_{\boldsymbol{k}}}{\partial \boldsymbol{k}} + f_{\boldsymbol{k}} \frac{e}{\hbar} (\boldsymbol{E} \times \boldsymbol{\Omega}(\boldsymbol{k})) \right] + \boldsymbol{\nabla} \mu \times \int \frac{d\boldsymbol{k}}{(2\pi)^d} \frac{\partial f_{\boldsymbol{k}}}{\partial \mu} \boldsymbol{m}_{\boldsymbol{k}} \left( 1 + \frac{e}{\hbar} \boldsymbol{B} \cdot \boldsymbol{\Omega}(\boldsymbol{k}) \right) \right.$$
$$\left. + \boldsymbol{\nabla} T \times \int \frac{d\boldsymbol{k}}{(2\pi)^d} \frac{\partial f_{\boldsymbol{k}}}{\partial T} \boldsymbol{m}_{\boldsymbol{k}} \left( 1 + \frac{e}{\hbar} \boldsymbol{B} \cdot \boldsymbol{\Omega}(\boldsymbol{k}) \right) \right] - \boldsymbol{\nabla} \mu \times \frac{\partial \boldsymbol{M}^e}{\partial \mu} - \boldsymbol{\nabla} T \times \frac{\partial \boldsymbol{M}^e}{\partial T}. \tag{9}$$

Demanding that the terms can only depend on the combination $(e\boldsymbol{E} + \boldsymbol{\nabla} \mu)$, we get the condition

$$\frac{\partial \boldsymbol{M}^e}{\partial \mu} = \sum_s \int \frac{d\boldsymbol{k}}{(2\pi)^d} \left[ \frac{\partial f_{\boldsymbol{k}}}{\partial \mu} \boldsymbol{m}_{\boldsymbol{k}} \left( 1 + \frac{e}{\hbar} \boldsymbol{B} \cdot \boldsymbol{\Omega}(\boldsymbol{k}) \right) + f_{\boldsymbol{k}} \frac{e}{\hbar} \boldsymbol{\Omega}(\boldsymbol{k}) \right]. \tag{10}$$

After integrating and applying the boundary condition that the charge magnetization should be zero for an empty band the constant of integration turns out to be zero. We get

$$M^e = \sum_s \int \frac{d\boldsymbol{k}}{(2\pi)^d}\left[\boldsymbol{m}(k)f_{\boldsymbol{k}}\left(1 + \frac{e}{\hbar}\boldsymbol{B}\cdot\boldsymbol{\Omega}(k)\right) + k_B T\frac{e\boldsymbol{\Omega}}{\hbar}\log\left(1 + e^{-\beta(\varepsilon_{\boldsymbol{k}}-\mu)}\right)\right]. \tag{11}$$

The intrinsic part of the charge magnetization is obtained by taking the limit of zero magnetic field, when the Zeeman splitting also vanishes. Even though the inversion symmetry maybe broken in a system like biased bilayer graphene, there is no spin orbit coupling to lift the spin degeneracy at zero magnetic field. Summing over both the spin species, we obtain

$$M^e\big|_{B=0} = \int \frac{2d\boldsymbol{k}}{(2\pi)^d}\left[\boldsymbol{m}(k)f_{\boldsymbol{k}} + k_B T\frac{e\boldsymbol{\Omega}}{\hbar}\log\left(1 + e^{-\beta(\varepsilon_0(\boldsymbol{k})-\mu)}\right)\right], \tag{12}$$

which is the same as the expression obtained in Ref. [56]. Similar expressions (in slightly different notations) may also be found in Refs. [63, 64]. The expression in Eq. (12) is for a single band. To get the total charge magnetization, we have to sum it over all the bands.

## 3.2 Energy magnetization

The expression in Eq. (7b) is the total energy current density, which is due to the sum of the transport and magnetization parts of the energy current. The magnetization energy current is the curl of energy magnetization, which is [55], $\boldsymbol{M}^E = \boldsymbol{M}_0^E - (\boldsymbol{E}\cdot\boldsymbol{r})\boldsymbol{M}^e$. Here $\boldsymbol{M}_0^E$ is the energy magnetization at zero external electric field, and $\boldsymbol{M}^e$ is the usual charge magnetization[2] (whose curl is the bound electric current density).

In the presence of an electric field, we need to add this additional term because the charges carry the potential whose gradient gives rise to the field. Both the bare energy magnetization $\boldsymbol{M}_0^E$ and the charge magnetization $\boldsymbol{M}^e$ are functions of the chemical potential $\mu$ and temperature $T$. Then, the circulating, bound energy current density is

$$\boldsymbol{j}_M^E = \boldsymbol{\nabla}\times\boldsymbol{M}_0^E - \boldsymbol{E}\times\boldsymbol{M}^e - \underbrace{(\boldsymbol{E}\cdot\boldsymbol{r})\boldsymbol{\nabla}\times\boldsymbol{M}^e}_{\text{2nd order quantity}}. \tag{13}$$

Here $\boldsymbol{\nabla}\times\boldsymbol{M}^e$ is already a function of $\boldsymbol{\nabla}\mu$ and $\boldsymbol{\nabla}T$, and hence, the term $(\boldsymbol{E}\cdot\boldsymbol{r})\boldsymbol{\nabla}\times\boldsymbol{M}^e$ is of second order, and we drop it. It follows that the transport energy current density is (see Appendix C)

$$\begin{aligned}
\boldsymbol{j}_{\text{transport}}^E &= \boldsymbol{j}_{\text{total}}^E - \boldsymbol{j}_M^E \\
&= \sum_s\Bigg[\int \frac{d\boldsymbol{k}}{(2\pi)^d}\varepsilon_{\boldsymbol{k}}\frac{1}{\hbar}\frac{\partial\varepsilon_{\boldsymbol{k}}}{\partial\boldsymbol{k}}g_{\boldsymbol{k}} - \boldsymbol{\nabla}\mu\times\frac{\partial}{\partial\mu}\int \frac{d\boldsymbol{k}}{(2\pi)^d}\varepsilon_{\boldsymbol{k}}f_{\boldsymbol{k}}\left(\frac{\boldsymbol{m}_{\boldsymbol{k}}}{e}\right)\left(1 + \frac{e}{\hbar}\boldsymbol{B}\cdot\boldsymbol{\Omega}(k)\right) \\
&\qquad - \boldsymbol{\nabla}T\times\frac{\partial}{\partial T}\int \frac{d\boldsymbol{k}}{(2\pi)^d}\varepsilon_{\boldsymbol{k}}f_{\boldsymbol{k}}\left(\frac{\boldsymbol{m}_{\boldsymbol{k}}}{e}\right)\left(1 + \frac{e}{\hbar}\boldsymbol{B}\cdot\boldsymbol{\Omega}(k)\right)\Bigg] \\
&\quad + e\boldsymbol{E}\times\boldsymbol{V}_1 - \boldsymbol{\nabla}\mu\times\frac{\partial\boldsymbol{M}_0^E}{\partial\mu} - \boldsymbol{\nabla}T\times\frac{\partial\boldsymbol{M}_0^E}{\partial T},
\end{aligned} \tag{14a}$$

where

$$\boldsymbol{V}_1 = \sum_s \int \frac{d\boldsymbol{k}}{(2\pi)^d}\frac{\boldsymbol{\Omega}(k)}{\hbar}\left[\varepsilon_{\boldsymbol{k}}f_{\boldsymbol{k}} + k_B T\log\left(1 + e^{-\beta(\varepsilon_{\boldsymbol{k}}-\mu)}\right)\right]. \tag{14b}$$

---

[2]In Ref. [55], the authors use the notation $\boldsymbol{M}^E = \boldsymbol{M}_0^E + \phi(\boldsymbol{r})\boldsymbol{M}^N$, where $\boldsymbol{M}^N$ (whose curl is bound number density current) is $\frac{\boldsymbol{M}^e}{-e}$, and $\phi(\boldsymbol{r}) = e\boldsymbol{E}\cdot\boldsymbol{r}$ is the potential energy due to the (constant) electric field.

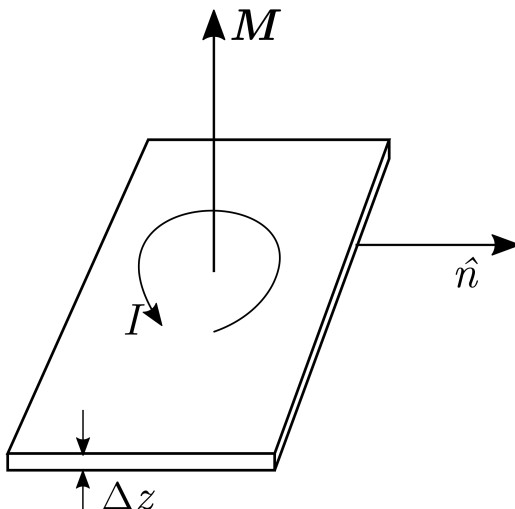

Figure 2: The circulating current and the magnetization in an effectively two-dimensional sample of thickness $\Delta z$ are related as $I = \Delta z |\boldsymbol{M} \times \hat{n}|$.

In Eq. (14a), there is an asymmetry between $e\boldsymbol{E}$ and $\boldsymbol{\nabla}\mu$. For the *Einstein relation* to hold for the energy current, the allowed terms can only depend on the combination $(e\boldsymbol{E} + \boldsymbol{\nabla}\mu)$. As shown in Sec. 4, $g_{\boldsymbol{k}}$ is proportional to this particular combination. Thus, for the Einstein relation to hold, we must have

$$-\frac{\partial \boldsymbol{M}_0^E}{\partial \mu} = \sum_s \left[ \int \frac{d\boldsymbol{k}}{(2\pi)^d} \varepsilon_k \frac{\partial f_{\boldsymbol{k}}}{\partial \mu} \left( \frac{\boldsymbol{m}_{\boldsymbol{k}}}{e} \right) \left( 1 + \frac{e}{\hbar} \boldsymbol{B} \cdot \boldsymbol{\Omega}(\boldsymbol{k}) \right) \right.$$
$$\left. + \int \frac{d\boldsymbol{k}}{(2\pi)^d} \frac{\boldsymbol{\Omega}(\boldsymbol{k})}{\hbar} \left[ \varepsilon_k f_{\boldsymbol{k}} + k_B T \log \left( 1 + e^{-\beta(\varepsilon_k - \mu)} \right) \right] \right]. \tag{15}$$

When the temperature scale ($k_B T$) is negligible compared to the chemical potential $\mu$ (more precisely, negligible compared to ($\mu - \varepsilon_{\min}$), where $\varepsilon_{\min}$ is the band minima), which happens for ordinary metals in room temperature, and there are multiple bands, it can be shown that (See Appendix D), up to leading order (zeroth order) in temperature,

$$-\frac{\partial \boldsymbol{M}_0^E}{\partial \mu} \underset{T \to 0}{\approx} \sum_{n,s} \frac{\mu}{\hbar} \int \frac{d\boldsymbol{k}}{(2\pi)^d} \left[ \boldsymbol{\Omega}_n(\boldsymbol{k}) f_{n\boldsymbol{k}} + \delta(\varepsilon_{n,\boldsymbol{k}} - \mu) \left( \frac{\hbar \boldsymbol{m}_{\boldsymbol{k}}}{e} \right) \left( 1 + \frac{e}{\hbar} \boldsymbol{B} \cdot \boldsymbol{\Omega}(\boldsymbol{k}) \right) \right], \tag{16}$$

where $n$ is the band index. This has a particularly simple interpretation in a *two-dimensional* Chern Insulator, where the chemical potential falls on a band gap and hence, the term containing the Dirac Delta function can be dropped. The magnitude of the circulating, bound energy current along an edge (not to be confused with the energy current density $\boldsymbol{j}^E$, which resides in the bulk) is given by $I_0^E = \Delta z |\boldsymbol{M}_0^E \times \hat{n}|$ (see Fig. 2).

Then, at zero magnetic field, the result in the above equation can be rewritten as

$$\frac{\Delta I_0^E}{\Delta \mu} \bigg|_{\text{Chern}} = \Delta z \sum_n \frac{\mu}{\hbar} \int \frac{2 d\boldsymbol{k}}{(2\pi)^2} \boldsymbol{\Omega}_n(\boldsymbol{k}) f_{n\boldsymbol{k}}. \tag{17}$$

A similar result for the charge magnetization and the circulating electric current was shown in Refs. [64, 66], using the known expression (Eq. (12)) for the charge magnetization. Note that we obtained this condition on the energy magnetization current from a necessary condition (Eq. (15)) so that the Einstein relation would hold. Let us discuss several cases, similar to the discussion on the circulating electric current by R. Resta [66].

For a Chern insulator, the integral of the Berry curvature over the filled bands (with all the factors of $2\pi$ included) is the sum of the Chern numbers of those bands, which can be interpreted as the number of chiral edge states [67, 68]. We get

$$\frac{\Delta I_0^E}{\Delta \mu}\bigg|_{\text{Chern}} = \Delta z \left(\frac{\mu}{h}\right) \left| N_\circlearrowleft - N_\circlearrowright \right|, \tag{18}$$

where $N_\circlearrowleft$ ($N_\circlearrowright$) is the number of chiral edge states circulating along the anti-clockwise (clockwise) direction. As the chemical potential is varied, per unit change in chemical potential, each of the chiral edge states contributes to the circulating energy current by an amount $\frac{\mu}{h}$ per unit thickness of the sample. It is worthwhile to note that in Eq. (16) and Eq. (18), the absolute value of the chemical potential appears and not its relative value with respect to the minima of the band. However, this is not unphysical at all. When we change the chemical potential by a small amount (i.e. $|\frac{\Delta \mu}{\mu - \epsilon_{\min}}| \ll 1$) by adding new electrons to the system, each of them will have energy in the order of $\mu$. Even if these newly added electrons belong to a bulk state, the circulating energy current at the edges will increase, and this change will be in the order of $\mu$. The net contribution is, as if, only the chiral edge states are contributing to the circulating energy current. A similar result holds for the circulating electric current [66]. For example, if the chemical potential is at zero energy with the band minima would be at some negative energy, and new electrons are added to the system, they would have almost zero energy, and the circulating energy current at the edges would not change, a result which agrees with Eq. (18).

When there are no chiral edge states (e.g. in a trivial insulator), the right hand side of Eq. (18) is zero. Also, when there are equal number of chiral edge states circulating in the clockwise and counter-clockwise direction, their contributions cancel.

In a metallic sample, the result would be similar to Eq. (18) for the filled bands, but the Fermi surface would contribute through the delta function term, and if the partially filled conduction band is of topological nature, it would also partially contribute.

It has been shown [65, 66] that the derivative of the charge magnetization with respect to the chemical potential has a term, $\frac{e}{\hbar} \sum_n \int \frac{d\boldsymbol{k}}{(2\pi)^d} f_{n\boldsymbol{k}} \Omega_n(\boldsymbol{k})$. The sign difference with Eq. (16) is because, when an electron moves, the local energy current density and the electric current density would be in opposite directions, as the electron bears a negative charge.

The expression for the bare energy magnetization at a non-zero magnetic field can be calculated by numerically integrating Eq. (15),

$$\begin{aligned}
\boldsymbol{M}_0^E = \sum_{n,s} \bigg[ & \int \frac{d\boldsymbol{k}}{(2\pi)^d} \varepsilon_{n,k} f_{n\boldsymbol{k}} \left(\frac{\boldsymbol{m}_k}{-e}\right) \left(1 + \frac{e}{\hbar} \boldsymbol{B} \cdot \boldsymbol{\Omega}(\boldsymbol{k})\right) \\
& - \int \frac{d\boldsymbol{k}}{(2\pi)^d} \frac{\Omega_n(\boldsymbol{k})}{\hbar} \int_{\tilde{\mu}=-\infty}^{\mu} d\tilde{\mu} \left\{ \varepsilon_{n,k} f_{n\boldsymbol{k}}(\tilde{\mu}) + k_B T \log\left(1 + e^{-\beta(\varepsilon_{n,k} - \tilde{\mu})}\right) \right\} \bigg],
\end{aligned} \tag{19}$$

with the initial condition that the energy magnetization would be zero when the chemical potential is at negative infinity, as then there would be no electrons to generate the circulating currents. Here, the first term can be regarded as the contribution of the individual orbital magnetic moments of the Bloch wavepackets, and the second term involving the Berry curvature arises due to the modification of the phase space volume. The intrinsic part (at zero magnetic field) of the bare energy magnetization is

$$\begin{aligned}
\boldsymbol{M}_0^E \big|_{B=0} = \sum_n \bigg[ & \int \frac{2d\boldsymbol{k}}{(2\pi)^d} \varepsilon_0(\boldsymbol{k}) f_{n\boldsymbol{k}} \left(\frac{\boldsymbol{m}_k}{-e}\right) \\
& - \int \frac{2d\boldsymbol{k}}{(2\pi)^d} \frac{\Omega_n(\boldsymbol{k})}{\hbar} \int_{\tilde{\mu}=-\infty}^{\mu} d\tilde{\mu} \left\{ \varepsilon_0(\boldsymbol{k}) f_{n\boldsymbol{k}}(\tilde{\mu}) + k_B T \log\left(1 + e^{-\beta(\varepsilon_0(\boldsymbol{k}) - \tilde{\mu})}\right) \right\} \bigg].
\end{aligned} \tag{20}$$

This expression agrees with the results obtained in previous scientific literature using different methods, namely, gauge theories of gravity [69], introduction of an inhomogeneous disorder field [58], and introduction of a gravitomagnetic field [61].

# 4 Complete expressions for transport currents in a two-dimensional sample with non-zero Berry curvature

The electric transport current density can be obtained by subtracting the curl of the charge magnetization from the total electric current density [56]. After substituting the expressions of charge and energy magnetization and subsequent simplification (See Appendix C), the expression for the transport electric current density, and the heat current density[3] are

$$
\boldsymbol{j}^e_{\text{transport}} = \sum_s \left[ -e \int \frac{d\boldsymbol{k}}{(2\pi)^d} g_{\boldsymbol{k}} \frac{1}{\hbar} \frac{\partial \varepsilon_{\boldsymbol{k}}}{\partial \boldsymbol{k}} - \int \frac{d\boldsymbol{k}}{(2\pi)^d} f_{\boldsymbol{k}} \frac{e^2}{\hbar} \left( \left[ \boldsymbol{E} + \frac{\boldsymbol{\nabla}\mu}{e} \right] \times \boldsymbol{\Omega}(\boldsymbol{k}) \right) \right] - \frac{\boldsymbol{\nabla}T}{T} \times e\boldsymbol{V}_2 ,
$$

(21a)

$$
\begin{aligned}
\boldsymbol{j}^Q_{\text{transport}} &= \boldsymbol{j}^E_{\text{transport}} - \mu \boldsymbol{j}^N_{\text{transport}} \\
&= \sum_s \int \frac{d\boldsymbol{k}}{(2\pi)^d} (\varepsilon_{\boldsymbol{k}} - \mu) \frac{1}{\hbar} \frac{\partial \varepsilon_{\boldsymbol{k}}}{\partial \boldsymbol{k}} g_{\boldsymbol{k}} + (e\boldsymbol{E} + \boldsymbol{\nabla}\mu) \times \boldsymbol{V}_2 + \boldsymbol{\nabla}T \times \frac{\partial \boldsymbol{V}_3}{\partial T} ,
\end{aligned}
$$

(21b)

where

$$
\boldsymbol{V}_2 = \sum_s \int \frac{d\boldsymbol{k}}{(2\pi)^d} \frac{\boldsymbol{\Omega}(\boldsymbol{k})}{\hbar} \left[ (\varepsilon_{\boldsymbol{k}} - \mu) f_{\boldsymbol{k}} + k_B T \log\left(1 + e^{-\beta(\varepsilon_{\boldsymbol{k}} - \mu)}\right) \right] ,
$$

(21c)

and

$$
\begin{aligned}
\boldsymbol{V}_3 = \sum_s \int \frac{d\boldsymbol{k}}{(2\pi)^d} \frac{\boldsymbol{\Omega}(\boldsymbol{k})}{\hbar} \Big[ &-\mu k_B T \log\left(1 + e^{-\beta(\varepsilon_{\boldsymbol{k}} - \mu)}\right) \\
&+ \int_{\tilde{\mu}=-\infty}^{\mu} d\tilde{\mu} \left[ \varepsilon_{\boldsymbol{k}} f_{\boldsymbol{k}}(\tilde{\mu}) + k_B T \log\left(1 + e^{-\beta(\varepsilon_{\boldsymbol{k}} - \tilde{\mu})}\right) \right] \Big] .
\end{aligned}
$$

(21d)

These expressions match with those previously obtained by introducing an inhomogeneous disorder field [58]. The second term of Eq. (21a) can produce a linear Hall response that does not depend on the relaxation timescale $\tau$ [72, 73]. Since the Fermi distribution is modified as $f_{\boldsymbol{k}} = \frac{1}{e^{\beta(\varepsilon_0(k) - \boldsymbol{m}_{\boldsymbol{k}} \cdot \boldsymbol{B} - \boldsymbol{m}_s \cdot \boldsymbol{B} - \mu)} + 1}$ when there is a magnetic field, the second term of Eq. (21a) can be Taylor expanded as, $\int \frac{d\boldsymbol{k}}{(2\pi)^d} \boldsymbol{\Omega}(\boldsymbol{k}) f_{\boldsymbol{k}} \approx \int \frac{d\boldsymbol{k}}{(2\pi)^d} \boldsymbol{\Omega}(\boldsymbol{k}) [f_{0\boldsymbol{k}} - \boldsymbol{m}_{\boldsymbol{k}} \cdot \boldsymbol{B} \frac{\partial f_{0\boldsymbol{k}}}{\partial \varepsilon_0}]$, where $f_{0\boldsymbol{k}} = \frac{1}{e^{\beta(\varepsilon_0(k) - \mu)} + 1}$, and the other first order term (which is linear in $\boldsymbol{B}$) proportional to the spin magnetic moment would vanish when integrated. The first order term proportional to the orbital magnetic moment, generates the aforementioned Hall response. Also note that $\boldsymbol{V}_2 \to 0$ as $T \to 0$, implying $\overleftrightarrow{L}_{21} = 0$ at $T = 0$, which is consistent with the Onsager relation, Eq. (2).

Once the band structure is known, the value of $\boldsymbol{\Omega}(\boldsymbol{k})$ can be obtained from the Bloch wavefunctions. Then, if we can find the expression of the out of equilibrium part of the distribution

---

[3]In general, there is no "heat magnetization" (whose curl is the heat current) analogous to the charge magnetization or the energy magnetization [70]. However, one can define such a quantity $\boldsymbol{M}^h = \boldsymbol{M}_0^E - \mu \boldsymbol{M}^N$, when the chemical potential throughout the sample is constant [58, 61, 69, 71].

function, $g_k$, the transport parts of the electric and the heat currents can be calculated up to linear order. We can calculate $g_k$ with the Boltzmann transport equation [74]. Being away from the "hydrodynamic regime", transport can be described in terms of the interaction of electrons (or quasiparticles) with phonons and other impurities. The timescale associated with this is much shorter than the one obtained from the sheer viscosity of electrons, which arises from the interactions among the electrons. When the external fields are time independent, we can drop the explicit partial derivatives with respect to time, and the equation takes the form

$$\frac{g_k}{\tau_k} + \dot{k} \cdot \frac{\partial}{\partial k} g_k + \dot{r} \cdot \frac{\partial}{\partial r} g_k = -\dot{k} \cdot \frac{\partial}{\partial k} f_k - \dot{r} \cdot \frac{\partial}{\partial r} f_k. \tag{22}$$

In general, the scattering timescale $\tau_k$ may depend on the crystal momentum of the electron. However, only the electrons near the Fermi surface can be effectively scattered, and if the scattering timescale only depends on the energy of the Bloch wavefunction, then we can consider it to be a constant. The following calculation remains valid even when $\tau_k$ varies with the crystal momentum. Using the decoupled equations of motion in two-dimensions,[4] the right hand side of the above equation simplifies to (see Appendix E.1)

$$\frac{\partial f_k}{\partial \varepsilon_k} \frac{1}{1 + \frac{e}{\hbar} B \cdot \Omega(k)} \frac{1}{\hbar} \frac{\partial \varepsilon_k}{\partial k} \cdot \left[ eE + \nabla \mu + \nabla T \frac{\varepsilon_k - \mu}{T} \right]. \tag{23}$$

In the above term, we can exchange $eE \leftrightarrow \nabla\mu$, i. e., the Einstein relation holds for the currents generated by the non-equilibrium distribution. We will show that the Onsager relation holds as well. We assume that the fields are uniform in space, and *drop the term*[5] containing $\frac{\partial g_k}{\partial r}$. Then, the BTE takes the form

$$\frac{g_k}{\tau_k} - \frac{\frac{e}{\hbar}E + \frac{e}{\hbar^2}\frac{\partial \varepsilon_k}{\partial k} \times B}{1 + \frac{e}{\hbar}B \cdot \Omega} \cdot \frac{\partial}{\partial k} g_k = \frac{\partial f}{\partial \varepsilon_k} \frac{1}{1 + \frac{e}{\hbar}B \cdot \Omega} \frac{1}{\hbar} \frac{\partial \varepsilon_k}{\partial k} \cdot \left[ eE + \nabla \mu + \nabla T \frac{\varepsilon_k - \mu}{T} \right]. \tag{24}$$

We can further simplify this equation. If we neglect the term $E \cdot \frac{\partial g_k}{\partial k}$ in the left-hand side (LHS), we would find (in the next section) that $g_k$ is a linear function of the electric field, the chemical potential gradient, and the temperature gradient. Then, $E \cdot \frac{\partial g_k}{\partial k}$ would be quadratic in the applied fields, and the solution would remain self-consistent up to the first order if we neglect it. Finally, the equation takes the form

$$\frac{g_k}{\tau_k} - \underbrace{\frac{\frac{e}{\hbar^2}\frac{\partial \varepsilon_k}{\partial k} \times B}{1 + \frac{e}{\hbar}B \cdot \Omega} \cdot \frac{\partial}{\partial k} g_k}_{\text{treated as a perturbation}} = \frac{\partial f}{\partial \varepsilon_k} \frac{1}{1 + \frac{e}{\hbar}B \cdot \Omega} \frac{1}{\hbar} \frac{\partial \varepsilon_k}{\partial k} \cdot \left[ eE + \nabla \mu + \nabla T \frac{\varepsilon_k - \mu}{T} \right]. \tag{25}$$

We can treat the second term in the LHS as a perturbation,[6] when the magnetic field is much smaller compared to $B_{crit} = \frac{m^*}{\tau e}$. If we take the effective mass to be the bare electron mass, and the scattering timescale to be that of a typical metal, $10^{-14}s$, the critical magnetic field

---

[4]The equations of motion, Eq. (3) can be decoupled [59], and for a two dimensional sample, they simplify to $\dot{r} = \frac{\frac{1}{\hbar}\frac{\partial \varepsilon_k}{\partial k} + \frac{e}{\hbar}(E \times \Omega)}{1 + \frac{e}{\hbar}B \cdot \Omega}$, and $\dot{k} = -\frac{\frac{e}{\hbar}E + \frac{e}{\hbar^2}\frac{\partial \varepsilon_k}{\partial k} \times B}{1 + \frac{e}{\hbar}B \cdot \Omega}$.

[5]When we solve the equation after discarding the term $\dot{r} \cdot \frac{\partial g_k}{\partial r}$, and substitute the solution (see Eq. (26)) in this term, we would find that this term is second order in the applied fields. Then up to linear order, we can discard it self-consistently.

[6]We have, $\dot{k} \cdot \frac{\partial}{\partial k} g \sim \dot{p} \cdot \frac{\partial}{\partial p} g \sim e(v \times B) \cdot \frac{\partial}{m \partial v} g \sim \omega_c \times v \cdot \frac{\partial g}{\partial v} \sim \omega g$. Then, if $\omega \ll \frac{1}{\tau}$ (in the limit of low magnetic field), it is justified to treat $\omega g$ as a perturbation over $\frac{g}{\tau}$.

turns out to be 570 T, which is much higher than any magnetic field accessible in the current laboratory setups. In typical metals, $m^* > m_e$, and the critical magnetic field is even larger.

The second term in the LHS of Eq. (25) gives rise to the (ordinary) Hall effect in samples with a non-zero Berry curvature, and that is why we would not completely neglect it. Up to linear order in external fields, the solution of Eq. (25) is (see Appendix E)

$$
\begin{aligned}
g_k =\,& \frac{\partial f}{\partial \varepsilon_k} \frac{1}{1+\frac{e}{\hbar}\boldsymbol{B}\cdot\boldsymbol{\Omega}} \frac{\tau_k}{\hbar} \frac{\partial \varepsilon_k}{\partial \boldsymbol{k}} \cdot \boldsymbol{S} + \frac{\frac{e\tau_k}{\hbar^2}}{1+\frac{e}{\hbar}\boldsymbol{B}\cdot\boldsymbol{\Omega}} \frac{\partial \varepsilon_k}{\partial \boldsymbol{k}} \cdot \boldsymbol{B} \times \left[ \frac{\frac{\partial f}{\partial \varepsilon_k}\tau_k}{1+\frac{e}{\hbar}\boldsymbol{B}\cdot\boldsymbol{\Omega}} \left[ \left( \frac{1}{\hbar}\boldsymbol{S}\cdot\frac{\partial}{\partial \boldsymbol{k}} \right) \frac{\partial \varepsilon_k}{\partial \boldsymbol{k}} \right] \right] \\
& + \frac{\frac{e\tau_k}{\hbar^2}}{1+\frac{e}{\hbar}\boldsymbol{B}\cdot\boldsymbol{\Omega}} \frac{\partial \varepsilon_k}{\partial \boldsymbol{k}} \cdot \boldsymbol{B} \times \left[ \frac{\partial}{\partial \boldsymbol{k}} \left[ \frac{\frac{\partial f}{\partial \varepsilon_k}\tau_k}{1+\frac{e}{\hbar}\boldsymbol{B}\cdot\boldsymbol{\Omega}} \right] \left( \frac{\boldsymbol{S}}{\hbar}\cdot\frac{\partial \varepsilon_k}{\partial \boldsymbol{k}} \right) \right],
\end{aligned}
\tag{26}
$$

where $\boldsymbol{S} = e\boldsymbol{E} + \boldsymbol{\nabla}\mu + \frac{\varepsilon_k - \mu}{T}\boldsymbol{\nabla}T$.[7] Having obtained Eq. (26), $g_k$ has to be substituted in the expressions of transport currents, Eq. (21a) and Eq. (21b), and the transport coefficients $\overleftrightarrow{L}_{11}$, $\overleftrightarrow{L}_{12}$, $\overleftrightarrow{L}_{21}$ and $\overleftrightarrow{L}_{22}$ can be read off. However, the actual computation will require the band structure and the Bloch eigenfunctions (to calculate the Berry curvature).

Similar solutions of the Boltzmann Transport Equation have been obtained in scientific literature [75–78], but the solution obtained in Eq. (26) takes into account the possibilities of the momentum dependence of the scattering timescale $\tau_k$, which the aforementioned works did not consider. Moreover, in this solution, the $\frac{1}{1+\frac{e}{\hbar}\boldsymbol{B}\cdot\boldsymbol{\Omega}}$ term has not been Taylor expanded, because the Berry curvature becomes a non-analytic function at the Dirac point (or the Weyl point), and needs to be treated carefully. In fact, it can be argued that the semi-classical treatment breaks down when the Fermi energy approaches a Dirac (or Weyl) point because of the vanishing of the density of states and a full quantum treatment is required. This has to take into account Landau quantization of the electron levels even at reasonably low values of the magnetic field. This quantum intervention ensures that transport coefficients remain well defined in the limit of the Fermi energy approaching the Dirac (or Weyl) point even as the Berry curvature diverges [79,80].

Let us discuss the role of each term in this solution. The first term is linear in $\tau_k$, and contributes to the regular Ohmic response. The second term, quadratic in the scattering timescale, generates the usual regular Hall response in the absence of Berry curvature. When there is a non-zero Berry curvature, the nature of the regular Hall response does not fundamentally change, rather it gets modified as the Berry curvature appears in the denominator of the third term. It may seem that since the Berry curvature only appears in the denominator multiplied with the Magnetic field, it contributes to the quadratic and higher order Hall responses. However, while the magnetic field in the laboratory maybe a small quantity (compared to some characteristic scale), the Berry curvature can exhibit singular behavior near the Dirac points of a system with effectively linear dispersion. In such systems, the third term of $g_k$ becomes non-analytic near the Dirac points, and we cannot simply Taylor expand it in the Magnetic field. The actual response can only be numerically calculated by plugging in this solution into the expressions of the electric and heat currents, Eq. (21a) and Eq. (21b). The third term, (also quadratic in the scattering time) captures the effects of the variation of the scattering time at different points in the Fermi surface.

---

[7]It can be easily verified that when there is no Berry curvature, and the band is parabolic with an effective mass $m^*$, and the scattering time $\tau$ is independent of $\boldsymbol{k}$, this solution produces the same electric conductivity tensor which is also obtained from the Drude model, $\overleftrightarrow{L}_{11} = \overleftrightarrow{\sigma} = \frac{ne^2\tau}{m^*}\begin{pmatrix} 1 & -\omega\tau \\ \omega\tau & 1 \end{pmatrix}$, where $\omega = \frac{eB}{m^*}$ is the cyclotron frequency.

In this solution, $\nabla T$ is always multiplied with a factor of $\frac{\varepsilon_k - \mu}{T}$, and the expression of heat current has a factor of $(\varepsilon_k - \mu)$ (Eq. (21b)), from which it follows that the Onsager relation holds for the contribution due to the non-equilibrium part of the distribution function in presence of Berry curvature. It has previously been demonstrated in Ref. [56] that the Onsager relation holds for the contribution due to the equilibrium part of the distribution.

## 5 Conclusions

The Einstein relation has been shown to hold from certain microscopic theories for the electric, energy, and heat transport current in systems with non-zero Berry curvature [56, 58]. In this paper, we employ a complementary approach to first demonstrate that, assuming that the Einstein relation holds (whose validity can be established from thermodynamic arguments [53, 54], irrespective of the underlying microscopic theory), an expression for the charge magnetization can be obtained in a relatively straightforward manner which agrees with the expression obtained for this quantity previously. We then extend this argument to the transport energy current and the heat current, to obtain a condition that the energy magnetization has to obey. We have used it to obtain an expression for the energy magnetization, which has been previously obtained using other methods. Moreover, we have found a physical interpretation of this condition, in terms of the circulating chiral edge modes in a Chern insulator. We have also solved the Boltzmann transport equation up to linear order in potential and temperature gradients for a two-dimensional system, which can be used to obtain the regular Hall response in systems like bilayer graphene, which possess a non-zero Berry curvature, but display no anomalous Hall Effect due to time reversal invariance.

## Acknowledgements

We thank Vijay Shenoy for discussions regarding the absolute value of the chemical potential appearing in several expressions. We thank the anonymous referees for their thoughtful comments and constructive criticism.

**Funding information** A. P. acknowledges support from the KVPY programme and S. M. thanks the Department of Science and Technology, Government of India for support.

## A Self contained derivation of Eq. (7a)

 **GOAL**: *The goal of this appendix is to demonstrate how the rotation of Bloch wavepacket about its center can induce terms in the electrical current density.*

The wavefunction of a Bloch wavepacket located at $r_0$, and peaked at crystal momentum $k_0$ is [4, 7] (here we follow the notation from [7])

$$\psi(\boldsymbol{r})_{k_0} = \int d\boldsymbol{k}\, w(\boldsymbol{k} - \boldsymbol{k}_0) e^{iA(\boldsymbol{k}_0)\cdot(\boldsymbol{k} - \boldsymbol{k}_0)} e^{-i\boldsymbol{k}\cdot\boldsymbol{r}_0} \left( e^{i\boldsymbol{k}\cdot\boldsymbol{r}} u_{\boldsymbol{k}}(\boldsymbol{r}) \right). \tag{A.1}$$

Here $w(\boldsymbol{k} - \boldsymbol{k}_0)$ is an (arbitrary) weight function sharply peaked at $\boldsymbol{k} = \boldsymbol{k}_0$, $u_{\boldsymbol{k}}(\boldsymbol{r})$ is the periodic part of the Bloch wavefunction with crystal momentum $\boldsymbol{k}$, and $A(\boldsymbol{k}) = i \langle u_{n,\boldsymbol{k}} | \nabla_{\boldsymbol{k}} | u_{n,\boldsymbol{k}} \rangle$ is the



Berry connection. $u_k(r)$ is normalized such that

$$\langle u_k | u_k \rangle = \int_{\text{unit cell}} dr \, |u_k(r)|^2 = 1 \,. \tag{A.2}$$

It is to be noted that the calculations in Appendix B do not depend on the actual form of the weight function $w(k - k_0)$.

An electron in a Bloch wavepacket centered at $r_0$ may be found at another point $r_1$ ($\neq r_0$), and it can contribute to the local current density at $r_1$. As an analogy, the total electron density near an atom in an insulator is not just the electron density of that atom, but the sum of electron densities belonging to all the atoms, evaluated at that point. Of course, atoms far away from the point would contribute very less, but nearby atoms can contribute significantly. Let us try write down the expectation value of a general operator $\mathcal{O}$ at a point $r_1$. We would consider the sum of the contributions from all the wavepackets centered at a point $r_0$, and sum over all $r_0$ [12]. That is

$$\langle \hat{O} \rangle (r_1) = \sum_s \int dr_0 \int \frac{dk}{(2\pi)^d}(f+g)\left(1 + \frac{e}{\hbar}B \cdot \Omega\right)\langle \psi_{k,r_0}| \frac{1}{2}\{\hat{O}, \delta(\hat{r} - r_1)\} |\psi_{k,r_0}\rangle \,. \tag{A.3}$$

Here $\hat{r}$ is the position operator that acts on the states. $r_1$ is just a parameter, and here $r_1$ acting on a state has to be understood as the operator $r_1 \hat{\mathcal{I}}$, with $\hat{\mathcal{I}}$ being the identity operator. And, $\frac{1}{2}\{\hat{O}, \delta(\hat{r} - r_0)\}$ denotes the Hermitized operator $\frac{\hat{O}\delta(\hat{r}-r_1)+\delta(\hat{r}-r_1)\hat{O}}{2}$, in case $\hat{r}$ and $\hat{O}$ do not commute (for example, to calculate the electric current, we need to find the expectation value of the velocity operator, which does not commute with the position operator).

Now, since the state is centered around $r_0$, we can take that into account by expanding the delta function (up to leading order) [81],

$$\delta(r_1 - \hat{r}) = \delta((r_1 - r_0) - (\hat{r} - r_0)) \approx \delta(r_1 - r_0) - (\hat{r} - r_0) \cdot \nabla_{r_1} \delta(r_1 - r_0)$$
$$= \delta(r_1 - r_0) - \nabla_{r_1} \cdot [(\hat{r} - r_0)\delta(r_1 - r_0)] \,. \tag{A.4}$$

Using this, we get

$$\langle \hat{O} \rangle (r_1) = \sum_s \left[ \int \frac{dk}{(2\pi)^d}(f+g)\left(1 + \frac{e}{\hbar}B \cdot \Omega\right)\langle \psi_{k,r_1}| \hat{O} |\psi_{k,r_1}\rangle \right.$$
$$\left. - \nabla_{r_1} \cdot \int \frac{dk}{(2\pi)^d}(f+g)\left(1 + \frac{e}{\hbar}B \cdot \Omega\right)\langle \psi_{k,r_1}| \frac{1}{2}\{\hat{O}, (\hat{r} - r_1)\} |\psi_{k,r_1}\rangle \right]. \tag{A.5}$$

We would work with this leading order expansion of the Delta function. For the current density operator, we need to separately take the three components of $\hat{j} = -e\frac{\hat{p}}{m}$ to be $\hat{O}$.

Let us define the tensor,

$$\mathcal{M}_{\mu\nu} = \langle \psi_{k,r_1}| \frac{1}{2}\{-e\frac{\hat{p}_\nu}{m}, (\hat{r} - r_1)_\mu\} |\psi_{k,r_1}\rangle \,, \tag{A.6}$$

where $\mu, \nu$ runs from $1, 2, \ldots d$. It can be shown that $\mathcal{M}_{\mu\nu}$ is completely anti-symmetric (see Appendix B.1 for proof), a fact stated without proof in [40]. Due to anti-symmetry of $\mathcal{M}$, we can further write this as (see Appendix B.2 and B.3 for proof)

$$\mathcal{M}_{\mu\nu} = \epsilon_{\alpha\mu\nu} m_\alpha \,, \tag{A.7}$$

where $m_\alpha$ is the $\alpha$-th component of the orbital magnetic moment $m_k$, defined in Eq. (6).

Note that we can rewrite the second term in Eq. (A.5) as

$$-\nabla_{r_1} \cdot \int \frac{d\boldsymbol{k}}{(2\pi)^d}(f+g)\Big(1+\frac{e}{\hbar}\boldsymbol{B}\cdot\boldsymbol{\Omega}\Big)\langle\psi_{\boldsymbol{k},r_1}|\frac{1}{2}\{\hat{\mathcal{O}},(\hat{\boldsymbol{r}}-\boldsymbol{r}_1)\}|\psi_{\boldsymbol{k},r_1}\rangle$$
$$=-\partial_\alpha \int \frac{d\boldsymbol{k}}{(2\pi)^d}(f+g)\Big(1+\frac{e}{\hbar}\boldsymbol{B}\cdot\boldsymbol{\Omega}\Big)\langle\psi_{\boldsymbol{k},r_1}|\frac{1}{2}\{\hat{\mathcal{O}},(\hat{\boldsymbol{r}}-\boldsymbol{r}_1)_\alpha\}|\psi_{\boldsymbol{k},r_1}\rangle, \tag{A.8}$$

with sum over $\alpha$ implied.

Now, let us calculate the $\mu$-th component of the electric current, $\langle\hat{\boldsymbol{j}}^e\rangle$. We take $\hat{\mathcal{O}}=-e\frac{\hat{p}_\mu}{m}$. We denote $[d\boldsymbol{k}]=\frac{d\boldsymbol{k}}{(2\pi)^d}(1+\frac{e}{\hbar}\boldsymbol{B}\cdot\boldsymbol{\Omega})$. We get

$$-\partial_\alpha\int[d\boldsymbol{k}](f+g)\mathcal{M}_{\alpha\mu}=-\partial_\alpha\int[d\boldsymbol{k}](f+g)m_\nu\epsilon_{\nu\alpha\mu}=+\partial_\alpha\int[d\boldsymbol{k}](f+g)m_\nu\epsilon_{\alpha\nu\mu}. \tag{A.9}$$

This is the $\mu$-th component of $\nabla\times\int\frac{d\boldsymbol{k}}{(2\pi)^d}(f+g)(1+\frac{e}{\hbar}\boldsymbol{B}\cdot\boldsymbol{\Omega})\boldsymbol{m}_{\boldsymbol{k}}$. When the electric field, the magnetic field, the chemical potential gradient and the temperature gradients are constant, we can drop the term $\nabla\times\int d\boldsymbol{k}\,g(1+\frac{e}{\hbar}\boldsymbol{B}\cdot\boldsymbol{\Omega})\boldsymbol{m}_{\boldsymbol{k}}$, because $g$ is already a linear function in the spatially uniform fields $\boldsymbol{E}$, $\nabla\mu$ and $\nabla T$.

While calculating electric current using Eq. (A.5) (with $\hat{\mathcal{O}}$ taken to be the current operator, $\hat{j}^e=-e\frac{\hat{p}_\mu}{m}$), the first term, which is the contribution of the center of the wavepacket, can be simplified (up to linear order),

$$\int\frac{d\boldsymbol{k}}{(2\pi)^d}(f+g)(1+\frac{e}{\hbar}\boldsymbol{B}\cdot\boldsymbol{\Omega})\langle\psi_{\boldsymbol{k},r_1}|\hat{\boldsymbol{j}}^e|\psi_{\boldsymbol{k},r_1}\rangle=-e\int\frac{d\boldsymbol{k}}{(2\pi)^d}\Big[g_{\boldsymbol{k}}\frac{1}{\hbar}\frac{\partial\varepsilon_{\boldsymbol{k}}}{\partial\boldsymbol{k}}+f_{\boldsymbol{k}}\frac{e}{\hbar}(\boldsymbol{E}\times\boldsymbol{\Omega}(\boldsymbol{k}))\Big]. \tag{A.10}$$

Note that the phase space correction factor $(1+\frac{e}{\hbar}\boldsymbol{B}\cdot\boldsymbol{\Omega})$ gets canceled. Finally we obtain, up to linear order,

$$\langle\hat{\boldsymbol{j}}^e\rangle=\boldsymbol{j}^e_{\text{total}}=\sum_s\Big[-e\int\frac{d\boldsymbol{k}}{(2\pi)^d}\Big[g_{\boldsymbol{k}}\frac{1}{\hbar}\frac{\partial\varepsilon_{\boldsymbol{k}}}{\partial\boldsymbol{k}}+f_{\boldsymbol{k}}\frac{e}{\hbar}(\boldsymbol{E}\times\boldsymbol{\Omega}(\boldsymbol{k}))\Big]+\nabla\times\int\frac{d\boldsymbol{k}}{(2\pi)^d}f_{\boldsymbol{k}}\Big(1+\frac{e}{\hbar}\boldsymbol{B}\cdot\boldsymbol{\Omega}\Big)\boldsymbol{m}_{\boldsymbol{k}}\Big]. \tag{A.11}$$

This equation has a simple interpretation. The net current is the contribution of the movement of the center the wavepackets, as well as due to the rotation about their individual centers.

# B  Demonstration that $m_\alpha\epsilon_{\alpha\mu\nu}=\mathcal{M}_{\mu\nu}$

**GOAL**: *The goal of this appendix is to demonstrate that the tensor $\mathcal{M}_{\mu\nu}$ defined in Eq. (A.6) is related to the components of the orbital magnetic moment $\boldsymbol{m}_{\boldsymbol{k}}$ defined in Eq. (6) as, $m_\alpha\epsilon_{\alpha\mu\nu}=\mathcal{M}_{\mu\nu}$. This result is crucial in obtaining Eq. (A.11).*

## B.1  We demonstrate that $\mathcal{M}_{\mu\nu}=\langle\psi_{\boldsymbol{k},r_1}|\frac{1}{2}\{-e\frac{\hat{p}_\nu}{m},(\hat{\boldsymbol{r}}-\boldsymbol{r}_1)_\mu\}|\psi_{\boldsymbol{k},r_1}\rangle$ is a totally anti-symmetric tensor

The effective Hamiltonian [52] acting on $u_{n,\boldsymbol{k}}$ in its effective Schrodinger equation is

$$H(\boldsymbol{k})=\Big[\frac{\hbar^2}{2m}(\boldsymbol{k}-i\nabla)^2+V(\boldsymbol{r})\Big]=e^{-i\boldsymbol{k}\cdot\boldsymbol{r}}\hat{H}e^{i\boldsymbol{k}\cdot\boldsymbol{r}}, \tag{B.1}$$

with $\hat{H}$ being the original Hamiltonian. Let us define

$$\tilde{\boldsymbol{P}}(\boldsymbol{k})=\frac{m}{\hbar}\frac{\partial H(\boldsymbol{k})}{\partial\boldsymbol{k}}=\hbar(\boldsymbol{k}-i\nabla)=e^{-i\boldsymbol{k}\cdot\boldsymbol{r}}\hat{\boldsymbol{P}}e^{i\boldsymbol{k}\cdot\boldsymbol{r}}. \tag{B.2}$$

We would first prove some preliminary results which we would need later on.

### B.1.1 Calculation of $\langle u_k | \tilde{P}(k) | u_k \rangle$

The idea of this calculation is based on [4].

$$
\begin{aligned}
\left\langle \tilde{P}(k) \right\rangle = \langle u_k | \tilde{P}(k) | u_k \rangle &= \frac{m}{\hbar} \langle u_k | \frac{\partial H(k)}{\partial k} | u_k \rangle \\
&= \frac{m}{\hbar} \left[ \langle u | \frac{\partial}{\partial k} (H(k) | u \rangle) - \langle u | H(k) \left| \frac{\partial}{\partial k} u \right\rangle \right] \\
&= \frac{m}{\hbar} \left[ \langle u | \frac{\partial}{\partial k} (\varepsilon_k | u \rangle) - \varepsilon_k \left\langle u \left| \frac{\partial}{\partial k} u \right\rangle \right. \right] \\
&= \frac{m}{\hbar} \left[ \langle u | \left( \varepsilon_k \left| \frac{\partial}{\partial k} u \right\rangle \right) + \langle u | \left( \frac{\partial \varepsilon_k}{\partial k} | u \rangle \right) - \varepsilon_k \left\langle u \left| \frac{\partial}{\partial k} u \right\rangle \right. \right] \\
&= \frac{m}{\hbar} \frac{\partial \varepsilon_k}{\partial k},
\end{aligned}
$$

(B.3)

which is just the (bare) electron mass times its velocity. This expression is nothing but an application of the Hellmann–Feynman theorem [82]. Note that the Berry curvature term does not appear here because we took a single Bloch eigenstate, not an wavepacket.

### B.1.2 Calculation of $\left\langle \frac{\partial}{\partial k} u_k \right| \tilde{P}(k) | u_k \rangle$

This calculation is based on [4].

$$
\begin{aligned}
\left\langle \frac{\partial}{\partial k_\mu} u_k \right| \tilde{P}_\nu(k) | u_k \rangle &= \frac{m}{\hbar} \left\langle \frac{\partial}{\partial k_\mu} u_k \right| \frac{\partial H(k)}{\partial k_\nu} | u_k \rangle \\
&= \frac{m}{\hbar} \left[ \left\langle \frac{\partial}{\partial k_\mu} u \right| \frac{\partial}{\partial k_\nu} (H(k) | u \rangle) - \left\langle \frac{\partial}{\partial k_\mu} u \right| H(k) \left| \frac{\partial}{\partial k_\nu} u \right\rangle \right] \\
&= \frac{m}{\hbar} \left[ \left\langle \frac{\partial}{\partial k_\mu} u \right| \frac{\partial}{\partial k_\nu} (\varepsilon_k | u \rangle) - \left\langle \frac{\partial}{\partial k_\mu} u \right| H(k) \left| \frac{\partial}{\partial k_\nu} u \right\rangle \right] \\
&= \frac{m}{\hbar} \left[ \left\langle \frac{\partial}{\partial k_\mu} u \right| \left( \varepsilon_k \left| \frac{\partial}{\partial k_\nu} u \right\rangle \right) + \left\langle \frac{\partial}{\partial k_\mu} u \right| \left( \frac{\partial \varepsilon_k}{\partial k_\nu} | u \rangle \right) - \varepsilon_k \left\langle \frac{\partial}{\partial k_\mu} u \left| \frac{\partial}{\partial k_\nu} u \right\rangle \right. \right] \\
&= \frac{m}{\hbar} \left[ \left\langle \frac{\partial}{\partial k_\mu} u \right| \varepsilon_k - H(k) \left| \frac{\partial}{\partial k_\nu} u \right\rangle \right] + \frac{m}{\hbar} \frac{\partial \varepsilon_k}{\partial k_\nu} \left\langle \frac{\partial}{\partial k_\mu} u \right| u \rangle \\
&= \frac{m}{\hbar} \left[ \left\langle \frac{\partial}{\partial k_\mu} u \right| \varepsilon_k - H(k) \left| \frac{\partial}{\partial k_\nu} u \right\rangle \right] + i A_\mu(k) \left\langle \tilde{P}_\nu(k) \right\rangle.
\end{aligned}
$$

(B.4)

### B.1.3 Calculation of $\left\langle u_{k_1} \right| e^{-ik \cdot (\hat{r} - r_0)} (r - r_0)_\mu \hat{P}_\nu e^{ik \cdot (r - r_0)} \left| u_{k_2} \right\rangle_{\text{all space}}$ (let us denote this quantity as $\mathcal{S}$)

**Notation**: We define

$$
\langle \psi_1 | \psi_2 \rangle_{\text{all space}} = \int_{\text{all space}} dr \, \psi_1^* \psi_2,
$$

(B.5a)

and

$$
\langle \psi_1 | \psi_2 \rangle_{\text{cell}} = \int_{\text{unit cell}} dr \, \psi_1^* \psi_2.
$$

(B.5b)

If $\psi_1$, $\psi_2$ are periodic over unit cells, then

$$\langle \psi_1 | \psi_2 \rangle_{\text{all space}} = N \langle \psi_1 | \psi_2 \rangle_{\text{cell}} , \tag{B.6}$$

where $N$ is the total number of unit cells.

We use the relation,

$$\hat{P}(e^{i\boldsymbol{k}\cdot(\boldsymbol{r}-\boldsymbol{r}_0)} |u_{\boldsymbol{k}_2}\rangle) = e^{i\boldsymbol{k}\cdot(\boldsymbol{r}-\boldsymbol{r}_0)} \tilde{P}(\boldsymbol{k}_2) |u_{\boldsymbol{k}_2}\rangle . \tag{B.7}$$

Then,

$$\begin{aligned}
S &= \langle u_{\boldsymbol{k}_1,n} | e^{-i\boldsymbol{k}\cdot(\hat{\boldsymbol{r}}-\boldsymbol{r}_0)} (\boldsymbol{r}-\boldsymbol{r}_0)_\mu \hat{P}_\nu e^{i\boldsymbol{k}\cdot(\boldsymbol{r}-\boldsymbol{r}_0)} |u_{\boldsymbol{k}_2,n}\rangle_{\text{all space}} \\
&= \langle u_{\boldsymbol{k}_1,n} | e^{i(\boldsymbol{k}_2-\boldsymbol{k}_1)\cdot(\hat{\boldsymbol{r}}-\boldsymbol{r}_0)} (\boldsymbol{r}-\boldsymbol{r}_0)_\mu \tilde{P}_\nu(\boldsymbol{k}_2) |u_{\boldsymbol{k}_2,n}\rangle_{\text{all space}} .
\end{aligned} \tag{B.8}$$

Since $|u_{\boldsymbol{k}_2,n}\rangle$ for different values of $n$ are eigenstates of $H(\boldsymbol{k}_2)$ for fixed value of $\boldsymbol{k}_2$, they form a complete set. Then, $\mathcal{I} = \sum_n |u_{\boldsymbol{k}_2,n}\rangle_{\text{cell cell}} \langle u_{\boldsymbol{k}_2,n}|$, and due to periodicity, $\frac{1}{N} \sum_n |u_{\boldsymbol{k}_2,n}\rangle_{\text{all space all space}} \langle u_{\boldsymbol{k}_2,n}| = \mathcal{I}$. We insert this in the expression above.

$$\begin{aligned}
S &= \langle u_{\boldsymbol{k}_1,n} | e^{i(\boldsymbol{k}_2-\boldsymbol{k}_1)\cdot(\hat{\boldsymbol{r}}-\boldsymbol{r}_0)} (\boldsymbol{r}-\boldsymbol{r}_0)_\mu \tilde{P}_\nu(\boldsymbol{k}_2) |u_{\boldsymbol{k}_2,n}\rangle_{\text{all space}} \\
&= \langle u_{\boldsymbol{k}_1,n} | i \left( \frac{\partial}{\partial k_{1\mu}} e^{i(\boldsymbol{k}_2-\boldsymbol{k}_1)\cdot(\boldsymbol{r}-\boldsymbol{r}_0)} \right) \tilde{P}_\nu(\boldsymbol{k}_2) |u_{\boldsymbol{k}_2,n}\rangle_{\text{all space}} \\
&= \frac{1}{N} \sum_{n'} \langle u_{\boldsymbol{k}_1,n} | i \left( \frac{\partial}{\partial k_{1\mu}} e^{i(\boldsymbol{k}_2-\boldsymbol{k}_1)\cdot(\boldsymbol{r}-\boldsymbol{r}_0)} \right) |u_{\boldsymbol{k}_2,n'}\rangle_{\text{all space}} \langle u_{\boldsymbol{k}_2,n'} | \tilde{P}_\nu(\boldsymbol{k}_2) |u_{\boldsymbol{k}_2,n}\rangle_{\text{all space}} ,
\end{aligned} \tag{B.9}$$

where we used the periodicity of the Bloch wavefunctions to absorb the normalization factor. Now we would manipulate the derivative with respect to $\boldsymbol{k}_{1_\mu}$,

$$\begin{aligned}
S &= \frac{i}{N} \sum_{n'} \left( \frac{\partial}{\partial k_{1\mu}} \langle u_{\boldsymbol{k}_1,n} | e^{i(\boldsymbol{k}_2-\boldsymbol{k}_1)\cdot(\boldsymbol{r}-\boldsymbol{r}_0)} \right) |u_{\boldsymbol{k}_2,n'}\rangle_{\text{all space}} \langle u_{\boldsymbol{k}_2,n'} | \tilde{P}_\nu(\boldsymbol{k}_2) |u_{\boldsymbol{k}_2,n}\rangle_{\text{all space}} \\
&\quad - \frac{i}{N} \sum_{n'} \left\langle \frac{\partial}{\partial k_{1\mu}} u_{\boldsymbol{k}_1,n} \right| e^{i(\boldsymbol{k}_2-\boldsymbol{k}_1)\cdot(\boldsymbol{r}-\boldsymbol{r}_0)} |u_{\boldsymbol{k}_2,n'}\rangle_{\text{all space}} \langle u_{\boldsymbol{k}_2,n'} | \tilde{P}_\nu(\boldsymbol{k}_2) |u_{\boldsymbol{k}_2,n}\rangle_{\text{all space}} .
\end{aligned} \tag{B.10}$$

We make use of the following two identities respectively, to simplify the first and the second terms in the right hand side of Eq. (B.10).

$$\frac{1}{N} \langle u_{\boldsymbol{k}_2,n'} | \tilde{P}_\nu(\boldsymbol{k}_2) |u_{\boldsymbol{k}_2,n}\rangle_{\text{all space}} = \langle u_{\boldsymbol{k}_2,n'} | \tilde{P}_\nu(\boldsymbol{k}_2) |u_{\boldsymbol{k}_2,n}\rangle_{\text{cell}} , \tag{B.11}$$

$$\frac{1}{N} \sum_n |u_{\boldsymbol{k}_2,n}\rangle_{\text{all space all space}} \langle u_{\boldsymbol{k}_2,n}| = \mathcal{I}. \tag{B.12}$$

Then, from Eq. (B.10),

$$\begin{aligned}
S &= i \sum_{n'} \frac{\partial}{\partial k_{1\mu}} \left( {}_{\text{all space}}\langle u_{\boldsymbol{k}_1,n} | e^{i(\boldsymbol{k}_2-\boldsymbol{k}_1)\cdot(\boldsymbol{r}-\boldsymbol{r}_0)} |u_{\boldsymbol{k}_2,n'}\rangle_{\text{all space}} \right) \langle u_{\boldsymbol{k}_2,n'} | \tilde{P}_\nu(\boldsymbol{k}_2) |u_{\boldsymbol{k}_2,n}\rangle_{\text{cell}} \\
&\quad - i \left\langle \frac{\partial}{\partial k_{1\mu}} u_{\boldsymbol{k}_1,n} \right| e^{i(\boldsymbol{k}_2-\boldsymbol{k}_1)\cdot(\boldsymbol{r}-\boldsymbol{r}_0)} \tilde{P}_\nu(\boldsymbol{k}_2) |u_{\boldsymbol{k}_2,n}\rangle_{\text{all space}} .
\end{aligned} \tag{B.13}$$

When the number of sites in a lattice is very large, the points in reciprocal space are dense, and we can approximate the sum over the real space lattice sites,

$$\sum_{\boldsymbol{R}} e^{i(\boldsymbol{k}_1-\boldsymbol{k}_2)\cdot\boldsymbol{R}} \rightarrow \frac{1}{V_c} \int d\boldsymbol{r}\, e^{i(\boldsymbol{k}_1-\boldsymbol{k}_2)\cdot\boldsymbol{r}} = \frac{(2\pi)^d}{V_c} \delta(\boldsymbol{k}_1-\boldsymbol{k}_2), \tag{B.14}$$

where $V_c$ is the volume of the $d$ dimensional unit cell. We can further write $\frac{(2\pi)^d}{V_c} = V_{\text{BZ}}$, where $V_{\text{BZ}}$ is the volume of the $1^{st}$ Brillouin zone.

For any periodic function over the unit cells,

$$
\int_{\text{all space}} e^{i(k_1-k_2)\cdot r} f(\text{periodic}) = \left( \sum_{R \in \text{lattice sites}} e^{i(k_1-k_2)\cdot R} \right) \left( \int_{\text{unit cell}} f(\text{periodic}) \right)
$$
$$
= [V_{\text{BZ}} \delta(k_1-k_2)] \left( \int_{\text{unit cell}} f(\text{periodic}) \right). \tag{B.15}
$$

Now we choose units such that $V_{\text{BZ}} = 1$ for the simplicity of the subsequent calculations (anyway it can always be absorbed in the normalization). From Eq. (B.13), we get

$$
\begin{aligned}
S &= i \sum_{n'} \frac{\partial}{\partial k_{1\mu}} \left( \delta(k_1-k_2) \langle u_{k_1,n} | u_{k_2,n'} \rangle_{\text{cell}} \right) \langle u_{k_2,n'} | \tilde{P}_\nu(k_2) | u_{k_2,n} \rangle_{\text{cell}} \\
&\quad - i \delta(k_1-k_2) \left\langle \frac{\partial}{\partial k_{1\mu}} u_{k_1,n} \middle| \tilde{P}_\nu(k_2) | u_{k_2,n} \right\rangle_{\text{cell}} \\
&= i \sum_{n'} \frac{\partial}{\partial k_{1\mu}} (\delta(k_1-k_2)) \langle u_{k_1,n} | u_{k_2,n'} \rangle_{\text{cell}} \langle u_{k_2,n'} | \tilde{P}_\nu(k_2) | u_{k_2,n} \rangle_{\text{cell}} \\
&\quad + i \sum_{n'} \delta(k_1-k_2) \left\langle \frac{\partial}{\partial k_{1\mu}} u_{k_1,n} \middle| u_{k_2,n'} \right\rangle_{\text{cell}} \langle u_{k_2,n'} | \tilde{P}_\nu(k_2) | u_{k_2,n} \rangle_{\text{cell}} \\
&\quad - i \delta(k_1-k_2) \left\langle \frac{\partial}{\partial k_{1\mu}} u_{k_1,n} \middle| \tilde{P}_\nu(k_2) | u_{k_2,n} \right\rangle_{\text{cell}}.
\end{aligned} \tag{B.16}
$$

After summing over $n'$ in the second term of the final line in Eq. (B.16), the second and the third terms cancel each other,

$$
\begin{aligned}
S &= i \sum_{n'} \left[ \frac{\partial}{\partial k_{1\mu}} \delta(k_1-k_2) \right] \langle u_{k_1,n} | u_{k_2,n'} \rangle_{\text{cell}} \langle u_{k_2,n'} | \tilde{P}_\nu(k_2) | u_{k_2,n} \rangle_{\text{cell}} \\
&= i \left[ \frac{\partial}{\partial k_{1\mu}} \delta(k_1-k_2) \right] \langle u_{k_1,n} | \tilde{P}_\nu(k_2) | u_{k_2,n} \rangle_{\text{cell}}.
\end{aligned} \tag{B.17}
$$

Therefore,

$$
\left\langle u_{k_1} \middle| e^{-ik\cdot(\hat{r}-r_0)} (r-r_0)_\mu \hat{P}_\nu e^{ik\cdot(r-r_0)} \middle| u_{k_2} \right\rangle_{\text{all space}} = i \left[ \frac{\partial}{\partial k_{1\mu}} \delta(k_1-k_2) \right] \langle u_{k_1,n} | \tilde{P}_\nu(k_2) | u_{k_2,n} \rangle_{\text{cell}}. \tag{B.18}
$$

### B.1.4 To show that $\left\langle \psi_{k,r_0} \middle| \{ (\hat{r}-r_0)_\mu, \hat{P}_\nu \} \middle| \psi_{k,r_0} \right\rangle$ is anti-symmetric in $\mu$, $\nu$ for a Bloch wavepacket $\psi_{k,r_0}$, which implies $\mathcal{M}_{\mu\nu}$ is totally anti-symmetric

It is mentioned in Ref. [40] that $\left\langle \psi_{k,r_0} \middle| \frac{1}{2} \{ (r-r_0)_\mu, \hat{P}_\nu \} \middle| \psi_{k,r_0} \right\rangle$ is a completely anti-symmetric tensor. Here we prove it. A similar calculation to find the orbital angular momentum of a wavepacket can be found at Appendix B of Ref. [4]. First we calculate the quantity without the anticommutator, and denote is as $S_1$.

$$
\begin{aligned}
S_1 &= \left\langle \psi_{k,r_0} \middle| (r-r_0)_\mu \hat{P}_\nu \middle| \psi_{k,r_0} \right\rangle \\
&= \int \int dk_1 dk_2 w(k_1-k) w(k_2-k) e^{-iA(k)\cdot(k_1-k)} e^{iA(k)\cdot(k_2-k)} \\
&\quad \times \left\langle u_{k_1} \middle| e^{-ik\cdot(\hat{r}-r_0)} (r-r_0)_\mu \hat{P}_\nu e^{ik\cdot(r-r_0)} \middle| u_{k_2} \right\rangle_{\text{all space}}.
\end{aligned} \tag{B.19}
$$

Now we substitute the boxed result (Eq. (B.18)) from the previous section into the above equation,

$$
\mathcal{S}_1 = \int\int d\mathbf{k}_1 d\mathbf{k}_2 w(\mathbf{k}_1-\mathbf{k})w(\mathbf{k}_2-\mathbf{k})e^{-iA(\mathbf{k})\cdot(\mathbf{k}_1-\mathbf{k})}e^{iA(\mathbf{k})\cdot(\mathbf{k}_2-\mathbf{k})}
$$
$$
\times i\left[\frac{\partial}{\partial k_{1\mu}}\delta(\mathbf{k}_1-\mathbf{k}_2)\right]\langle u_{\mathbf{k}_1,n}|\tilde{\mathbf{P}}_\nu(\mathbf{k}_2)|u_{\mathbf{k}_2,n}\rangle_{\text{cell}}\ . \tag{B.20}
$$

After integrating by parts to shift the derivative from the delta function to the other quantities,

$$
\mathcal{S}_1 = \int\int d\mathbf{k}_1 d\mathbf{k}_2 w(\mathbf{k}_1-\mathbf{k})w(\mathbf{k}_2-\mathbf{k}_0)e^{-iA(\mathbf{k})\cdot(\mathbf{k}_1-\mathbf{k})}e^{iA(\mathbf{k})\cdot(\mathbf{k}_2-\mathbf{k})}
$$
$$
\times i\left[\frac{\partial}{\partial k_{1\mu}}\delta(\mathbf{k}_1-\mathbf{k}_2)\right]\langle u_{\mathbf{k}_1,n}|\tilde{\mathbf{P}}_\nu(\mathbf{k}_2)|u_{\mathbf{k}_2,n}\rangle_{\text{cell}}
$$
$$
= -i\int\int d\mathbf{k}_1 d\mathbf{k}_2 w(\mathbf{k}_1-\mathbf{k})w(\mathbf{k}_2-\mathbf{k})e^{-iA(\mathbf{k}_0)\cdot(\mathbf{k}_1-\mathbf{k}_0)}e^{iA(\mathbf{k})\cdot(\mathbf{k}_2-\mathbf{k})} \tag{B.21}
$$
$$
\times\delta(\mathbf{k}_1-\mathbf{k}_2)\left[\left\langle\frac{\partial}{\partial k_{1\mu}}u_{\mathbf{k}_1,n}\middle|\tilde{\mathbf{P}}_\nu(\mathbf{k}_2)\middle|u_{\mathbf{k}_2,n}\right\rangle_{\text{cell}}-iA_\mu(\mathbf{k})\langle u_{\mathbf{k}_1,n}|\tilde{\mathbf{P}}_\nu(\mathbf{k}_2)|u_{\mathbf{k}_2,n}\rangle_{\text{cell}}\right]\ .
$$

Integrating over the delta function, and using Eq.(B.4), we get

$$
\mathcal{S}_1 = -i\int d\mathbf{k}_1 [w(\mathbf{k}_1-\mathbf{k})]^2 \frac{m}{\hbar}\left[\left\langle\frac{\partial}{\partial k_{1\mu}}u\middle|\varepsilon_{\mathbf{k}_1}-H(\mathbf{k}_1)\middle|\frac{\partial}{\partial k_{1\nu}}u\right\rangle\right]
$$
$$
= -i\frac{m}{\hbar}\left[\left\langle\frac{\partial}{\partial k_\mu}u\middle|\varepsilon_{\mathbf{k}}-H(\mathbf{k})\middle|\frac{\partial}{\partial k_\nu}u\right\rangle\right]. \tag{B.22}
$$

The last line follows because $w(\mathbf{k}_1-\mathbf{k})$ being sharply peaked at $\mathbf{k}_1=\mathbf{k}$, picks up the value of the integrand at that point, and due to normalization of $\psi(\mathbf{r})_{n,\mathbf{k}}$ in Eq. (A.1), $\int d\mathbf{k}_1 [w(\mathbf{k}_1-\mathbf{k})]^2 = 1$ (In other words, $[w(\mathbf{k}_1-\mathbf{k})]^2$ effectively behaves like $\delta(\mathbf{k}_1-\mathbf{k})$). Therefore,

$$
\boxed{\langle\psi_{\mathbf{k},\mathbf{r}_0}|(\mathbf{r}-\mathbf{r}_0)_\mu\hat{P}_\nu|\psi_{\mathbf{k},\mathbf{r}_0}\rangle = -i\frac{m}{\hbar}\left[\left\langle\frac{\partial}{\partial k_\mu}u\middle|\varepsilon_{\mathbf{k}}-H(\mathbf{k})\middle|\frac{\partial}{\partial k_\nu}u\right\rangle\right].} \tag{B.23}
$$

**Case**: $\mu\neq\nu$.

In this case, $(\mathbf{r}-\mathbf{r}_0)_\mu\hat{P}_\nu$ is Hermitian (because $(\mathbf{r}-\mathbf{r}_0)_\mu$ and $\hat{P}_\nu$ commute, and they are individually Hermitian), and its expectation value must be real. Then, the quantity

$$
\left\langle\frac{\partial}{\partial k_\mu}u\middle|\varepsilon_{\mathbf{k}}-H(\mathbf{k})\middle|\frac{\partial}{\partial k_\nu}u\right\rangle = \frac{i\hbar}{m}\langle\psi_{\mathbf{k},\mathbf{r}_0}|\{(\mathbf{r}-\mathbf{r}_0)_\mu\hat{P}_\nu\}|\psi_{\mathbf{k},\mathbf{r}_0}\rangle \tag{B.24}
$$

must be purely imaginary.

Now, $\varepsilon_{\mathbf{k}}-H(\mathbf{k})$ is Hermitian. Then,

$$
\left\langle\frac{\partial}{\partial k_\mu}u\middle|\varepsilon_{\mathbf{k}}-H(\mathbf{k})\middle|\frac{\partial}{\partial k_\nu}u\right\rangle^* = \left\langle\frac{\partial}{\partial k_\nu}u\middle|\varepsilon_{\mathbf{k}}-H(\mathbf{k})\middle|\frac{\partial}{\partial k_\mu}u\right\rangle. \tag{B.25}
$$

But since this is purely imaginary,

$$
\left\langle\frac{\partial}{\partial k_\mu}u\middle|\varepsilon_{\mathbf{k}}-H(\mathbf{k})\middle|\frac{\partial}{\partial k_\nu}u\right\rangle^* = -\left\langle\frac{\partial}{\partial k_\mu}u\middle|\varepsilon_{\mathbf{k}}-H(\mathbf{k})\middle|\frac{\partial}{\partial k_\nu}u\right\rangle. \tag{B.26}
$$

Combining the two, we get

$$\left\langle \frac{\partial}{\partial k_\mu} u \middle| \varepsilon_k - H(k) \middle| \frac{\partial}{\partial k_\nu} u \right\rangle = -\left\langle \frac{\partial}{\partial k_\nu} u \middle| \varepsilon_k - H(k) \middle| \frac{\partial}{\partial k_\mu} u \right\rangle. \tag{B.27}$$

Also, since $(r - r_0)_\mu$ and $\hat{P}_\nu$ commute, for $\mu \neq \nu$,

$$(r - r_0)_\mu \hat{P}_\nu = \frac{1}{2}\{(r - r_0)_\mu, \hat{P}_\nu\}. \tag{B.28}$$

Consequently, for $\mu \neq \nu$,

$$\left\langle \psi_{k,r_0} \middle| \frac{1}{2}\{(r - r_0)_\mu, \hat{P}_\nu\} \middle| \psi_{k,r_0} \right\rangle = -\left\langle \psi_{k,r_0} \middle| \frac{1}{2}\{(r - r_0)_\nu, \hat{P}_\mu\} \middle| \psi_{k,r_0} \right\rangle, \tag{B.29}$$

i.e.,

$$\boxed{\mathcal{M}_{\mu\nu} = -\mathcal{M}_{\nu\mu}.} \tag{B.30}$$

**Case**: $\mu = \nu$. In this subsection, no sum is implied for repeated indices.

In this case, $(r - r_0)_\mu \hat{P}_\mu$ is not anymore Hermitian. However, $\left[\left\langle \frac{\partial}{\partial k_\mu} u \middle| \varepsilon_k - H(k) \middle| \frac{\partial}{\partial k_\mu} u \right\rangle\right]$ must be real. Consequently,

$$-i\frac{m}{\hbar}\left[\left\langle \frac{\partial}{\partial k_\mu} u \middle| \varepsilon_k - H(k) \middle| \frac{\partial}{\partial k_\mu} u \right\rangle\right] = \left\langle (r - r_0)_\mu \hat{P}_\mu \right\rangle \tag{B.31}$$

is purely imaginary, and its complex conjugate $\left\langle \hat{P}_\mu (r - r_0)_\mu \right\rangle$ would be exactly the negative of that. Thus,

$$\left\langle (r - r_0)_\mu \hat{P}_\mu + \hat{P}_\mu (r - r_0)_\mu \right\rangle = 0. \tag{B.32}$$

Therefore, for all $\mu$,

$$\boxed{\mathcal{M}_{\mu\mu} = 0,} \tag{B.33}$$

i.e., all the diagonal terms are individually zero.

Note: Since we know that $\left\langle (r - r_0)_\mu \hat{P}_\mu \right\rangle$ is purely imaginary, and also,

$$(r - r_0)_\mu \hat{P}_\mu - \hat{P}_\mu (r - r_0)_\mu = i\hbar, \tag{B.34}$$

it follows that

$$\left\langle (r - r_0)_\mu \hat{P}_\mu \right\rangle = \frac{i\hbar}{2}. \tag{B.35}$$

Therefore, $\mathcal{M}$ is a *completely anti-symmetric* tensor.

## B.2 We demonstrate that $\tilde{m}_\alpha = \frac{1}{2}\mathcal{M}_{\mu\nu}\epsilon_{\alpha\mu\nu}$ is identical to the magnetic moment defined in Eq. (6)

Since only the $\mu \neq \nu$ terms contribute in $\mathcal{M}_{\mu\nu}\epsilon_{\alpha\mu\nu}$, and $\hat{p}_\nu$ commutes with $\hat{r}_\mu$ for $\mu \neq \nu$, we can get rid of the anticommutator,

$$\tilde{m}_\alpha = \frac{1}{2}\left\langle \psi_{k,r_0} \middle| \frac{1}{2}\{-e\frac{\hat{p}_\nu}{m}, (\hat{r} - r_0)_\mu\} \middle| \psi_{k,r_0} \right\rangle \epsilon_{\alpha\mu\nu} = -\frac{e}{2m}\epsilon_{\alpha\mu\nu}\left\langle \psi_{k,r_0} \middle| (\hat{r} - r_0)_\mu \hat{p}_\nu \middle| \psi_{k,r_0} \right\rangle$$
$$= -\frac{e}{2m}\left\langle \psi_{k,r_0} \middle| ((\hat{r} - r_0) \times \hat{p})_\alpha \middle| \psi_{k,r_0} \right\rangle, \tag{B.36}$$

which is the $\alpha$-th component of magnetic moment $m_k$, defined in Eq. (6), that is, $\tilde{m}_i = m_i$.

**B.3 We demonstrate that $\mathcal{M}_{\gamma\sigma} = m_\alpha \epsilon_{\alpha\gamma\sigma}$**

Multiplying both sides of the equation

$$m_\alpha = \frac{1}{2}\mathcal{M}_{\mu\nu}\epsilon_{\alpha\mu\nu}, \tag{B.37}$$

with $\epsilon_{\alpha\gamma\sigma}$, and summing over $\alpha$, we get

$$\begin{aligned}
m_\alpha \epsilon_{\alpha\gamma\sigma} &= \frac{1}{2}\mathcal{M}_{\mu\nu}\epsilon_{\alpha\mu\nu}\epsilon_{\alpha\gamma\sigma} \\
&= \frac{1}{2}\mathcal{M}_{\mu\nu}\left(\delta_{\mu\gamma}\delta_{\nu\sigma} - \delta_{\mu\sigma}\delta_{\nu\gamma}\right) \\
&= \frac{1}{2}\left(\mathcal{M}_{\gamma\sigma} - \mathcal{M}_{\sigma\gamma}\right) \\
&= \mathcal{M}_{\gamma\sigma}
\end{aligned} \tag{B.38}$$

due to anti-symmetry of $\mathcal{M}$.

# C Derivation of transport heat current density

 **GOAL**: *The goal of this appendix is to derive the complete expressions of currents presented in Section 4.*

Since the energy magnetization $\boldsymbol{M}_0^E$ is a function of the temperature $T$ and the chemical potential $\mu$, it follows from Eq. (13) that the magnetization energy current is

$$\begin{aligned}
\boldsymbol{j}_M^E &= \boldsymbol{\nabla}\mu \times \frac{\partial \boldsymbol{M}_0^E}{\partial \mu} + \boldsymbol{\nabla}T \times \frac{\partial \boldsymbol{M}_0^E}{\partial T} - \boldsymbol{E} \times \boldsymbol{M}^e \\
&= \boldsymbol{\nabla}\mu \times \frac{\partial \boldsymbol{M}_0^E}{\partial \mu} + \boldsymbol{\nabla}T \times \frac{\partial \boldsymbol{M}_0^E}{\partial T} \\
&\quad - \boldsymbol{E} \times \sum_s \int \frac{d\boldsymbol{k}}{(2\pi)^d}\left[\boldsymbol{m}(\boldsymbol{k})\left(1 + \frac{e}{\hbar}\boldsymbol{B}\cdot\boldsymbol{\Omega}(\boldsymbol{k})\right)f_{\boldsymbol{k}} + k_B T \frac{e\boldsymbol{\Omega}}{\hbar}\log\left(1 + e^{-\beta(\varepsilon_{\boldsymbol{k}}-\mu)}\right)\right].
\end{aligned} \tag{C.1}$$

Then, the transport energy current density is

$$\begin{aligned}
\boldsymbol{j}_{\text{transport}}^E &= \boldsymbol{j}_{\text{total}}^E - \boldsymbol{j}_M^E \\
&= \sum_s \Bigg[ \int \frac{d\boldsymbol{k}}{(2\pi)^d}\varepsilon_{\boldsymbol{k}}\left[g_{\boldsymbol{k}}\frac{1}{\hbar}\frac{\partial\varepsilon_{\boldsymbol{k}}}{\partial\boldsymbol{k}} + \frac{e}{\hbar}f_{\boldsymbol{k}}(\boldsymbol{E}\times\boldsymbol{\Omega}(\boldsymbol{k}))\right] \\
&\quad + \boldsymbol{E}\times\int\frac{d\boldsymbol{k}}{(2\pi)^d}k_B T\frac{e\boldsymbol{\Omega}}{\hbar}\log\left(1 + e^{-\beta(\varepsilon_{\boldsymbol{k}}-\mu)}\right) \\
&\quad - \boldsymbol{\nabla}\mu\times\frac{\partial}{\partial\mu}\int\frac{d\boldsymbol{k}}{(2\pi)^d}\varepsilon_{\boldsymbol{k}}f_{\boldsymbol{k}}\left(\frac{\boldsymbol{m}_{\boldsymbol{k}}}{e}\right)\left(1 + \frac{e}{\hbar}\boldsymbol{B}\cdot\boldsymbol{\Omega}(\boldsymbol{k})\right) \\
&\quad - \boldsymbol{\nabla}T\times\frac{\partial}{\partial T}\int\frac{d\boldsymbol{k}}{(2\pi)^d}\varepsilon_{\boldsymbol{k}}f_{\boldsymbol{k}}\left(\frac{\boldsymbol{m}_{\boldsymbol{k}}}{e}\right)\left(1 + \frac{e}{\hbar}\boldsymbol{B}\cdot\boldsymbol{\Omega}(\boldsymbol{k})\right)\Bigg] \\
&\quad - \boldsymbol{\nabla}\mu\times\frac{\partial\boldsymbol{M}_0^E}{\partial\mu} - \boldsymbol{\nabla}T\times\frac{\partial\boldsymbol{M}_0^E}{\partial T}.
\end{aligned} \tag{C.2}$$

Substituting the following expression of the bare energy magnetization,

$$M_0^E = \sum_{n,s} \left[ \int \frac{d\boldsymbol{k}}{(2\pi)^d} \varepsilon_{n,k} f_{nk} \left( \frac{\boldsymbol{m}_k}{-e} \right) \left( 1 + \frac{e}{\hbar} \boldsymbol{B} \cdot \boldsymbol{\Omega}(\boldsymbol{k}) \right) \right.$$
$$\left. - \int \frac{d\boldsymbol{k}}{(2\pi)^d} \frac{\boldsymbol{\Omega}_n(\boldsymbol{k})}{\hbar} \int_{\tilde{\mu}=-\infty}^{\mu} d\tilde{\mu} \left\{ \varepsilon_{n,k}(\boldsymbol{k}) f_{nk}(\tilde{\mu}) + k_B T \log\left( 1 + e^{-\beta(\varepsilon_{n,k}-\tilde{\mu})} \right) \right\} \right] \tag{C.3}$$

in the above Eq. (C.2), we get

$$j_{\text{transport}}^E = \sum_s \left[ \int \frac{d\boldsymbol{k}}{(2\pi)^d} \varepsilon_k g_k \frac{1}{\hbar} \frac{\partial \varepsilon_k}{\partial \boldsymbol{k}} + (e\boldsymbol{E} + \boldsymbol{\nabla}\mu) \int \frac{d\boldsymbol{k}}{(2\pi)^d} \frac{\boldsymbol{\Omega}}{\hbar} \left( \varepsilon_k f_k + k_B T \log\left( 1 + e^{-\beta(\varepsilon_k-\mu)} \right) \right) \right.$$
$$\left. + \boldsymbol{\nabla}T \times \frac{\partial}{\partial T} \int \frac{d\boldsymbol{k}}{(2\pi)^d} \frac{\boldsymbol{\Omega}}{\hbar} \int_{\tilde{\mu}=-\infty}^{\mu} \left( \varepsilon_k f_k(\tilde{\mu}) + k_B T \log\left( 1 + e^{-\beta(\varepsilon_k-\tilde{\mu})} \right) \right) \right] . \tag{C.4}$$

The circulating magnetization electric current is

$$j_M^e = \boldsymbol{\nabla} \times \boldsymbol{M}^e$$
$$= \sum_s \left[ \boldsymbol{\nabla} \times \int \frac{d\boldsymbol{k}}{(2\pi)^d} f \boldsymbol{m}_k \left( 1 + \frac{e}{\hbar} \boldsymbol{B} \cdot \boldsymbol{\Omega}(\boldsymbol{k}) \right) + \boldsymbol{\nabla}\mu \times \int \frac{d\boldsymbol{k}}{(2\pi)^d} \frac{e}{\hbar} \boldsymbol{\Omega}(\boldsymbol{k}) f \right.$$
$$\left. + \frac{\boldsymbol{\nabla}T}{T} \times \int \frac{d\boldsymbol{k}}{(2\pi)^d} \frac{e}{\hbar} \boldsymbol{\Omega}(\boldsymbol{k}) f_k(\varepsilon_k - \mu) + \frac{\boldsymbol{\nabla}T}{T} \times \int \frac{d\boldsymbol{k}}{(2\pi)^d} \frac{e}{\hbar} \boldsymbol{\Omega}(\boldsymbol{k}) k_B T \log\left( 1 + e^{-\beta(\varepsilon_k-\mu)} \right) \right] . \tag{C.5}$$

The electric transport current is [56]

$$j^e{}_{\text{transport}} = \sum_s \left[ \int \frac{d\boldsymbol{k}}{(2\pi)^d} (-e) g_k \frac{1}{\hbar} \frac{\partial \varepsilon_k}{\partial \boldsymbol{k}} - f_k \frac{e^2}{\hbar} \left( \left[ \boldsymbol{E} + \frac{\boldsymbol{\nabla}\mu}{e} \right] \times \boldsymbol{\Omega}(\boldsymbol{k}) \right) \right.$$
$$\left. - \frac{\boldsymbol{\nabla}T}{T} \times \int \frac{d\boldsymbol{k}}{(2\pi)^d} \frac{e}{\hbar} \boldsymbol{\Omega}(\boldsymbol{k}) \left[ f_k(\varepsilon_k - \mu) + k_B T \log\left( 1 + e^{-\beta(\varepsilon_k-\mu)} \right) \right] \right]$$
$$= \sum_s \left[ \int \frac{d\boldsymbol{k}}{(2\pi)^d} (-e) g_k \frac{1}{\hbar} \frac{\partial \varepsilon_k}{\partial \boldsymbol{k}} - f_k \frac{e^2}{\hbar} \left( \left[ \boldsymbol{E} + \frac{\boldsymbol{\nabla}\mu}{e} \right] \times \boldsymbol{\Omega}(\boldsymbol{k}) \right) \right.$$
$$\left. - \boldsymbol{\nabla}T \times \frac{\partial}{\partial T} \int \frac{d\boldsymbol{k}}{(2\pi)^d} \frac{e}{\hbar} \boldsymbol{\Omega}(\boldsymbol{k}) \left[ k_B T \log\left( 1 + e^{-\beta(\varepsilon_k-\mu)} \right) \right] \right] . \tag{C.6}$$

The number density transport current is

$$j_{\text{transport}}^N = \frac{j_{\text{transport}}^e}{-e}, \tag{C.7}$$

and the full expression for the transport heat current density (presented in Eq. (21b)) is

$$j_{\text{transport}}^Q = j_{\text{transport}}^E - \mu j_{\text{transport}}^N$$
$$= \sum_s \left[ \int \frac{d\boldsymbol{k}}{(2\pi)^d} (\varepsilon_k - \mu) \frac{1}{\hbar} \frac{\partial \varepsilon_k}{\partial \boldsymbol{k}} g_k + (e\boldsymbol{E} + \boldsymbol{\nabla}\mu) \times \int \frac{d\boldsymbol{k}}{(2\pi)^d} \frac{\boldsymbol{\Omega}}{\hbar} \left( \varepsilon_k f_k + k_B T \log\left( 1 + e^{-\beta(\varepsilon_k-\mu)} \right) \right) \right.$$
$$\left. + \boldsymbol{\nabla}T \times \frac{\partial}{\partial T} \int \frac{d\boldsymbol{k}}{(2\pi)^d} \frac{\boldsymbol{\Omega}}{\hbar} \left[ \left\{ \int_{\tilde{\mu}=-\infty}^{\mu} \left( \varepsilon_k f_k(\tilde{\mu}) + k_B T \log\left( 1 + e^{-\beta(\varepsilon_k-\tilde{\mu})} \right) \right) \right\} - \mu k_B T \log\left( 1 + e^{-\beta(\varepsilon_k-\mu)} \right) \right] \right] . \tag{C.8}$$

# D   Simplification of the condition on energy magnetization at low temperature

**GOAL**: *In this appendix, we describe the approximations utilized to obtain Eq.(16) from Eq.(15).*
    At very low temperatures (in the limit $\beta\mu \gg 1$), for $|\boldsymbol{k}| < k_f$, we have $f_{\boldsymbol{k}} \approx 1$, and

$$\log\left(1 + e^{-\beta(\varepsilon_k - \mu)}\right) \approx \log\left(e^{-\beta(\varepsilon_k - \mu)}\right) = \beta(\mu - \varepsilon_k). \tag{D.1}$$

Thus, for $|\boldsymbol{k}| < k_f$,

$$\varepsilon_k f_{\boldsymbol{k}} + k_B T \log\left(1 + e^{-\beta(\varepsilon_k - \mu)}\right) \approx \mu. \tag{D.2}$$

And for $|\boldsymbol{k}| > k_f$, $f_{\boldsymbol{k}} \approx e^{-\beta(\varepsilon_k - \mu)} \ll 1$, and

$$\log\left(1 + e^{-\beta(\varepsilon_k - \mu)}\right) \approx e^{-\beta(\varepsilon_k - \mu)} \ll 1. \tag{D.3}$$

In this case ($|\boldsymbol{k}| > k_f$), the quantity $\left[\varepsilon_k f_{\boldsymbol{k}} + k_B T \log\left(1 + e^{-\beta(\varepsilon_k - \mu)}\right)\right]$ is negligibly small compared to $\mu$.
Combining both the cases, we can write

$$\varepsilon_k f_{\boldsymbol{k}} + k_B T \log\left(1 + e^{-\beta(\varepsilon_k - \mu)}\right) \approx \mu\Theta(\mu - \varepsilon_k) \approx \mu f_{\boldsymbol{k}}, \tag{D.4}$$

where $\Theta$ is the Heaviside step function.
We also use the fact that, at very low temperatures,

$$\frac{\partial f_{\boldsymbol{k}}}{\partial \mu} \approx \delta(\varepsilon_k - \mu). \tag{D.5}$$

# E   Calculations for the Boltzmann transport equation

**GOAL**: *In this appendix, we demonstrate in detail how we solve the Boltzmann Transport Equation.*

In this appendix, the subscripts $\boldsymbol{k}$ of $\varepsilon$, $g$ and $\tau$ are not explicitly written. It is to be understood that they are all functions of the crystal momentum.

## E.1   Simplification of right-hand side of Eq. (22)

Since $f = \frac{1}{e^{\beta(\varepsilon - \mu)} + 1}$, it follows that

$$\frac{\partial f}{\partial \boldsymbol{r}} = \frac{\partial f}{\partial \varepsilon}\left[-\frac{\boldsymbol{\nabla} T}{T}(\varepsilon - \mu) - \boldsymbol{\nabla}\mu\right], \tag{E.1a}$$

and,

$$\frac{\partial f}{\partial \boldsymbol{k}} = \frac{\partial f}{\partial \varepsilon}\frac{\partial \varepsilon}{\partial \boldsymbol{k}}. \tag{E.1b}$$

We will now calculate the quantity $\dot{\boldsymbol{r}} \cdot \frac{\partial}{\partial \boldsymbol{r}} f + \dot{\boldsymbol{k}} \cdot \frac{\partial}{\partial \boldsymbol{k}} f$ using the decoupled equations of motion in 2D, which are

$$\dot{\boldsymbol{r}} = \frac{\frac{1}{\hbar}\frac{\partial \varepsilon}{\partial \boldsymbol{k}} + \frac{e}{\hbar}(\boldsymbol{E} \times \boldsymbol{\Omega})}{1 + \frac{e}{\hbar}\boldsymbol{B} \cdot \boldsymbol{\Omega}}, \tag{E.2a}$$

$$\dot{\boldsymbol{k}} = -\frac{\frac{e}{\hbar}\boldsymbol{E} + \frac{e}{\hbar^2}\frac{\partial \varepsilon}{\partial \boldsymbol{k}} \times \boldsymbol{B}}{1 + \frac{e}{\hbar}\boldsymbol{B} \cdot \boldsymbol{\Omega}}. \tag{E.2b}$$

Here, $\frac{\partial f}{\partial \boldsymbol{r}}$ is linear in external fields, and $\dot{\boldsymbol{r}}$ has a term $\boldsymbol{E} \times \boldsymbol{\Omega}$, and we neglect the product of these terms, which is a second order quantity.

Therefore, we get

$$-\dot{\boldsymbol{r}} \cdot \frac{\partial}{\partial \boldsymbol{r}} f - \dot{\boldsymbol{k}} \cdot \frac{\partial}{\partial \boldsymbol{k}} f = \frac{\partial f}{\partial \varepsilon} \frac{1}{1 + \frac{e}{\hbar} \boldsymbol{B} \cdot \boldsymbol{\Omega}} \frac{1}{\hbar} \frac{\partial \varepsilon}{\partial \boldsymbol{k}} \cdot \left[ e\boldsymbol{E} + \boldsymbol{\nabla}\mu + \frac{\boldsymbol{\nabla} T}{T}(\varepsilon - \mu) \right]. \qquad (\text{E.3})$$

## E.2 Solution of the BTE for $\boldsymbol{E} \neq 0$, $\boldsymbol{\nabla}\mu = \boldsymbol{\nabla} T = 0$

In this case, the Boltzmann Transport Equation (Eq. (22)) becomes

$$\frac{g}{\tau} - \frac{\frac{e}{\hbar^2}}{1 + \frac{e}{\hbar} \boldsymbol{B} \cdot \boldsymbol{\Omega}} \frac{\partial \varepsilon}{\partial \boldsymbol{k}} \times \boldsymbol{B} \cdot \frac{\partial}{\partial \boldsymbol{k}} g = \frac{\partial f}{\partial \varepsilon} \frac{1}{1 + \frac{e}{\hbar} \boldsymbol{B} \cdot \boldsymbol{\Omega}} \frac{1}{\hbar} \frac{\partial \varepsilon}{\partial \boldsymbol{k}} \cdot e\boldsymbol{E}. \qquad (\text{E.4})$$

We solve this equation, treating the second term in LHS as a perturbation (the second term is of the order $\omega g$ with $\omega$ being the cyclotron frequency, whereas the first term is of the order $\frac{g}{\tau}$. As long as $\omega\tau \ll 1$, treating the second term as a perturbation is valid). We write

$$g = g_0 + g_1, \qquad (\text{E.5a})$$

such that,

$$\frac{g_0}{\tau} = \frac{\partial f}{\partial \varepsilon} \frac{1}{1 + \frac{e}{\hbar} \boldsymbol{B} \cdot \boldsymbol{\Omega}} \frac{1}{\hbar} \frac{\partial \varepsilon}{\partial \boldsymbol{k}} \cdot e\boldsymbol{E}, \qquad (\text{E.5b})$$

and,

$$\frac{g_1}{\tau} - \frac{\frac{e}{\hbar^2}}{1 + \frac{e}{\hbar} \boldsymbol{B} \cdot \boldsymbol{\Omega}} \frac{\partial \varepsilon}{\partial \boldsymbol{k}} \times \boldsymbol{B} \cdot \frac{\partial}{\partial \boldsymbol{k}} g_0 = 0. \qquad (\text{E.5c})$$

From the last equation, it is evident that $g_1$ is a linear function of $\boldsymbol{B}$, and it is justified to discard the term $\frac{\frac{e}{\hbar^2}}{1 + \frac{e}{\hbar} \boldsymbol{B} \cdot \boldsymbol{\Omega}} \frac{\partial \varepsilon}{\partial \boldsymbol{k}} \times \boldsymbol{B} \cdot \frac{\partial}{\partial \boldsymbol{k}} g_1$, as it would be quadratic in $\boldsymbol{B}$. Then,

$$g_0 = \frac{\partial f}{\partial \varepsilon} \frac{1}{1 + \frac{e}{\hbar} \boldsymbol{B} \cdot \boldsymbol{\Omega}} \frac{e\tau}{\hbar} \frac{\partial \varepsilon}{\partial \boldsymbol{k}} \cdot \boldsymbol{E}. \qquad (\text{E.6})$$

To find $g_1$, we need to calculate $\frac{\partial g_0}{\partial \boldsymbol{k}}$, i.e., the quantity $\frac{\partial}{\partial \boldsymbol{k}} \left[ \frac{\partial f}{\partial \varepsilon} \frac{1}{1 + \frac{e}{\hbar} \boldsymbol{B} \cdot \boldsymbol{\Omega}} \frac{e\tau}{\hbar} \frac{\partial \varepsilon}{\partial \boldsymbol{k}} \cdot \boldsymbol{E} \right]$.

To calculate it, let us first evaluate an expression of the form $\boldsymbol{\nabla}[\phi \boldsymbol{A} \cdot \boldsymbol{C}]$, where $\phi$ is a scalar function, $\boldsymbol{A}$ is a vector function, and $\boldsymbol{C}$ is a constant vector.

$$\boldsymbol{\nabla}[\phi \boldsymbol{A} \cdot \boldsymbol{C}] = (\boldsymbol{\nabla}\phi)(\boldsymbol{A} \cdot \boldsymbol{C}) + \phi(\boldsymbol{C} \cdot \boldsymbol{\nabla})\boldsymbol{A} + \phi(\boldsymbol{C} \times (\boldsymbol{\nabla} \times \boldsymbol{A})). \qquad (\text{E.7})$$

In our calculation, $\phi \sim \frac{\partial f}{\partial \varepsilon} \frac{1}{1 + \frac{e}{\hbar} \boldsymbol{B} \cdot \boldsymbol{\Omega}} \frac{e\tau}{\hbar}$, $\boldsymbol{A} \sim \frac{\partial \varepsilon}{\partial \boldsymbol{k}}$, and $\boldsymbol{C} \sim \boldsymbol{E}$.

Then,

$$\frac{\partial}{\partial \boldsymbol{k}} \left[ \frac{\partial f}{\partial \varepsilon} \frac{1}{1 + \frac{e}{\hbar} \boldsymbol{B} \cdot \boldsymbol{\Omega}} \frac{e\tau}{\hbar} \frac{\partial \varepsilon}{\partial \boldsymbol{k}} \cdot \boldsymbol{E} \right] = \frac{\partial}{\partial \boldsymbol{k}} \left[ \frac{\frac{\partial f}{\partial \varepsilon}\tau}{1 + \frac{e}{\hbar} \boldsymbol{B} \cdot \boldsymbol{\Omega}} \right] \frac{e\boldsymbol{E}}{\hbar} \cdot \frac{\partial \varepsilon}{\partial \boldsymbol{k}}$$

$$+ \frac{\frac{\partial f}{\partial \varepsilon}\tau}{1 + \frac{e}{\hbar} \boldsymbol{B} \cdot \boldsymbol{\Omega}} \left[ \left( \frac{e}{\hbar}\boldsymbol{E} \cdot \frac{\partial}{\partial \boldsymbol{k}} \right) \frac{\partial \varepsilon}{\partial \boldsymbol{k}} + \frac{e}{\hbar}\boldsymbol{E} \times \left( \underbrace{\frac{\partial}{\partial \boldsymbol{k}} \times \frac{\partial \varepsilon}{\partial \boldsymbol{k}}}_{0} \right) \right].$$

$$(\text{E.8})$$

The last term is zero because it is the curl of a gradient. Finally, the solution is obtained by adding $g_0$ and $g_1$,

$$
g = \frac{\partial f}{\partial \varepsilon} \frac{1}{1 + \frac{e}{\hbar} \boldsymbol{B} \cdot \boldsymbol{\Omega}} \frac{\tau}{\hbar} \frac{\partial \varepsilon}{\partial \boldsymbol{k}} \cdot e\boldsymbol{E}
$$
$$
+ \frac{\frac{e\tau}{\hbar^2}}{1 + \frac{e}{\hbar} \boldsymbol{B} \cdot \boldsymbol{\Omega}} \frac{\partial \varepsilon}{\partial \boldsymbol{k}} \cdot \boldsymbol{B} \times \left[ \frac{\partial}{\partial \boldsymbol{k}} \left[ \frac{\frac{\partial f}{\partial \varepsilon} \tau}{1 + \frac{e}{\hbar} \boldsymbol{B} \cdot \boldsymbol{\Omega}} \right] \left( \frac{e\boldsymbol{E}}{\hbar} \cdot \frac{\partial \varepsilon}{\partial \boldsymbol{k}} \right) + \frac{\frac{\partial f}{\partial \varepsilon} \tau}{1 + \frac{e}{\hbar} \boldsymbol{B} \cdot \boldsymbol{\Omega}} \left[ \left( \frac{e}{\hbar} \boldsymbol{E} \cdot \frac{\partial}{\partial \boldsymbol{k}} \right) \frac{\partial \varepsilon}{\partial \boldsymbol{k}} \right] \right].
$$
(E.9)

### E.3 Solution of the BTE for $\boldsymbol{E} = 0$, $\boldsymbol{\nabla}\mu \neq 0$, $\boldsymbol{\nabla}T \neq 0$

Under these conditions, the Boltzmann Transport Equation becomes,

$$
\frac{g}{\tau} - \frac{\frac{e}{\hbar^2} \frac{\partial \varepsilon}{\partial \boldsymbol{k}} \times \boldsymbol{B}}{1 + \frac{e}{\hbar} \boldsymbol{B} \cdot \boldsymbol{\Omega}} \cdot \frac{\partial}{\partial \boldsymbol{k}} g = \frac{\partial f}{\partial \varepsilon} \frac{1}{1 + \frac{e}{\hbar} \boldsymbol{B} \cdot \boldsymbol{\Omega}} \frac{1}{\hbar} \frac{\partial \varepsilon}{\partial \boldsymbol{k}} \cdot \left[ \boldsymbol{\nabla}\mu + \boldsymbol{\nabla}T \frac{\varepsilon - \mu}{T} \right].
$$
(E.10)

Similar to the previous section, let

$$
g = g_0 + g_1,
$$
(E.11a)

with

$$
g_0 = \frac{\partial f}{\partial \varepsilon} \frac{1}{1 + \frac{e}{\hbar} \boldsymbol{B} \cdot \boldsymbol{\Omega}} \frac{\tau}{\hbar} \frac{\partial \varepsilon}{\partial \boldsymbol{k}} \cdot \left[ \boldsymbol{\nabla}\mu + \boldsymbol{\nabla}T \frac{\varepsilon - \mu}{T} \right],
$$
(E.11b)

and,

$$
\frac{g_1}{\tau} - \frac{\frac{e}{\hbar^2} \frac{\partial \varepsilon}{\partial \boldsymbol{k}} \times \boldsymbol{B}}{1 + \frac{e}{\hbar} \boldsymbol{B} \cdot \boldsymbol{\Omega}} \cdot \frac{\partial}{\partial \boldsymbol{k}} g_0 = 0.
$$
(E.11c)

The form of the equations (E.11b), (E.11c) is otherwise similar to the form of the equations (E.5b), (E.5c) in Appendix E.2, but here we have a factor of $(\varepsilon - \mu)$ with $\boldsymbol{\nabla}T$. As a result, we would get *additional factors* like $\frac{\partial \varepsilon}{\partial \boldsymbol{k}} \cdot \frac{\boldsymbol{\nabla}T}{T}$ when we calculate $\frac{\partial g_0}{\partial \boldsymbol{k}}$ in order to calculate $g_1$. One might think that such additional factors may violate the Onsager relation. But such factors would cancel due to a vector triple product being zero (Eq. (E.14)), and the Onsager relation continues to hold.

We have, from Eq. (E.11b),

$$
\frac{\partial g_0}{\partial \boldsymbol{k}} = \frac{\partial}{\partial \boldsymbol{k}} \left[ \frac{\frac{\partial f}{\partial \varepsilon} \tau}{1 + \frac{e}{\hbar} \boldsymbol{B} \cdot \boldsymbol{\Omega}} \right] \frac{\boldsymbol{\nabla}\mu + \boldsymbol{\nabla}T \frac{\varepsilon - \mu}{T}}{\hbar} \cdot \frac{\partial \varepsilon}{\partial \boldsymbol{k}}
$$
$$
+ \frac{\frac{\partial f}{\partial \varepsilon} \tau}{1 + \frac{e}{\hbar} \boldsymbol{B} \cdot \boldsymbol{\Omega}} \left[ \left( \frac{\boldsymbol{\nabla}\mu + \boldsymbol{\nabla}T \frac{\varepsilon - \mu}{T}}{\hbar} \cdot \frac{\partial}{\partial \boldsymbol{k}} \right) \frac{\partial \varepsilon}{\partial \boldsymbol{k}} + \frac{1}{\hbar} \left( \boldsymbol{\nabla}\mu + \boldsymbol{\nabla}T \frac{\varepsilon - \mu}{T} \right) \times \underbrace{\left( \frac{\partial}{\partial \boldsymbol{k}} \times \frac{\partial \varepsilon}{\partial \boldsymbol{k}} \right)}_{0} \right]
$$
$$
+ \frac{\frac{\partial f}{\partial \varepsilon} \tau}{1 + \frac{e}{\hbar} \boldsymbol{B} \cdot \boldsymbol{\Omega}} \left[ \frac{1}{\hbar} \left( \frac{\partial \varepsilon}{\partial \boldsymbol{k}} \cdot \frac{\partial}{\partial \boldsymbol{k}} \right) \left( \frac{\varepsilon \boldsymbol{\nabla}T}{T} \right) + \frac{1}{\hbar} \frac{\partial \varepsilon}{\partial \boldsymbol{k}} \times \left( \frac{\partial}{\partial \boldsymbol{k}} \times \left( \frac{\varepsilon \boldsymbol{\nabla}T}{T} \right) \right) \right].
$$
(E.12)

The term inside the last third bracket of Eq. (E.12) can be further simplified.

$$\left(\frac{\partial\varepsilon}{\partial\boldsymbol{k}}\cdot\frac{\partial}{\partial\boldsymbol{k}}\right)\left(\frac{\varepsilon\boldsymbol{\nabla}T}{T}\right)+\frac{\partial\varepsilon}{\partial\boldsymbol{k}}\times\left(\frac{\partial}{\partial\boldsymbol{k}}\times\left(\frac{\varepsilon\boldsymbol{\nabla}T}{T}\right)\right)=\left(\frac{\partial\varepsilon}{\partial\boldsymbol{k}}\cdot\frac{\partial\varepsilon}{\partial\boldsymbol{k}}\right)\left(\frac{\boldsymbol{\nabla}T}{T}\right)+\frac{\partial\varepsilon}{\partial\boldsymbol{k}}\times\left(\frac{\partial\varepsilon}{\partial\boldsymbol{k}}\times\left(\frac{\boldsymbol{\nabla}T}{T}\right)\right)$$

$$=\left(\frac{\partial\varepsilon}{\partial\boldsymbol{k}}\cdot\frac{\partial\varepsilon}{\partial\boldsymbol{k}}\right)\left(\frac{\boldsymbol{\nabla}T}{T}\right)+\left(\frac{\partial\varepsilon}{\partial\boldsymbol{k}}\cdot\frac{\boldsymbol{\nabla}T}{T}\right)\frac{\partial\varepsilon}{\partial\boldsymbol{k}}-\left(\frac{\partial\varepsilon}{\partial\boldsymbol{k}}\cdot\frac{\partial\varepsilon}{\partial\boldsymbol{k}}\right)\left(\frac{\boldsymbol{\nabla}T}{T}\right)$$

$$=\left(\frac{\partial\varepsilon}{\partial\boldsymbol{k}}\cdot\frac{\boldsymbol{\nabla}T}{T}\right)\frac{\partial\varepsilon}{\partial\boldsymbol{k}}.$$

(E.13)

When we calculate $g_1\left(=\tau\frac{\frac{e}{\hbar^2}\frac{\partial\varepsilon}{\partial\boldsymbol{k}}\times\boldsymbol{B}}{1+\frac{e}{\hbar}\boldsymbol{B}\cdot\boldsymbol{\Omega}}\cdot\frac{\partial}{\partial\boldsymbol{k}}g_0\right)$ in Eq. (E.11c), the above term cancels because

$$\left(\frac{\partial\varepsilon}{\partial\boldsymbol{k}}\times\boldsymbol{B}\right)\cdot\frac{\partial\varepsilon}{\partial\boldsymbol{k}}=0.$$

(E.14)

Then, the full solution is

$$g=g_0+g_1$$

$$=\frac{\partial f}{\partial\varepsilon}\frac{1}{1+\frac{e}{\hbar}\boldsymbol{B}\cdot\boldsymbol{\Omega}}\frac{\tau}{\hbar}\frac{\partial\varepsilon}{\partial\boldsymbol{k}}\cdot\left[\boldsymbol{\nabla}\mu+\boldsymbol{\nabla}T\frac{\varepsilon-\mu}{T}\right]+\frac{\frac{e\tau}{\hbar^2}}{1+\frac{e}{\hbar}\boldsymbol{B}\cdot\boldsymbol{\Omega}}\frac{\partial\varepsilon}{\partial\boldsymbol{k}}\cdot$$

$$\left[\boldsymbol{B}\times\left[\frac{\partial}{\partial\boldsymbol{k}}\left[\frac{\frac{\partial f}{\partial\varepsilon}\tau}{1+\frac{e}{\hbar}\boldsymbol{B}\cdot\boldsymbol{\Omega}}\right]\frac{\boldsymbol{\nabla}\mu+\boldsymbol{\nabla}T\frac{\varepsilon-\mu}{T}}{\hbar}\cdot\frac{\partial\varepsilon}{\partial\boldsymbol{k}}+\frac{\frac{\partial f}{\partial\varepsilon}\tau}{1+\frac{e}{\hbar}\boldsymbol{B}\cdot\boldsymbol{\Omega}}\left[\left(\frac{\boldsymbol{\nabla}\mu+\boldsymbol{\nabla}T\frac{\varepsilon-\mu}{T}}{\hbar}\cdot\frac{\partial}{\partial\boldsymbol{k}}\right)\frac{\partial\varepsilon}{\partial\boldsymbol{k}}\right]\right]\right].$$

(E.15)

Since Eq. (25) is linear, when the electric field, the temperature gradient and the chemical potential gradient are each non-zero, we can add the two solutions in appendix E.2 and appendix E.3, and we get the solution in Eq. (26).

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
