# Peer review of "Energy magnetization and transport in systems with a non-zero Berry curvature in a magnetic field"

_SciPost Physics Core, doi:SciPost Phys. Core 6, 052 (2023)_

## Round 1 · Referee Report · Anonymous · 2022-6-27

Strengths
1. Proposes an alternative way of deriving the expressions for the
charge and energy magnetization of Bloch electrons [Eqs. (8) and (14)
in the manuscript].
2. A useful and detailed Introduction, with abundant references to
previous works.
3. Detailed derivations are given in the Appendices. See however point
4 in the Weaknesses section.
Weaknesses
1. A siginificant part of the manuscript is devoted to rederiving in a
new way results that were previously obtained in the literature,
namely Eqs. (8) and (14).
2. The authors claim to have identified a new type of regular Hall
response in time reversal invariant systems coming from the Berry
curvature and from the nonequilibrium part of the distribution
function given by Eq.(18). However, an actual expression for that Hall
response is never written down. As detailed in the Report, I believe
that such a contribution to the regular Hall response cannot exist.
3. The formal results obtained in the manuscript are not illustrated
by any explicit calculations on model systems.
4. The main text is not sufficiently self contained. I had to
constantly flip back and forth between the main text and the numerous
appendices (as well as the numerous footnotes) to keep up with the
manuscript.
Report
The authors have shown that requiring the transport charge and energy
currents to obey the Einstein relation provides an alternative route
for deriving the known expressions for the charge and energy
magnetization of Bloch electrons at finite temperature [Eqs.(8) and
(14) in the manuscript].
This approach complements the standard approach [e.g., Ref. 58] where
one starts from those expressions for the magnetizations, subtracts
their curl from the expressions for the local (or total) current
densities to obtain the transport current densities, and then verifies
a posteriori that the resulting transport coefficients satisfy the
Einstein relation.
The new approach in this manuscript provides a fresh perspective. But
that fact that it is only used to recover previously known results
limits its impact somewhat.
In addition to the new derivation of Eqs.(8) and (14), the other main
result of the manuscript is the expression in Eq.(18), valid for 2D
systems, of the correction to the distribution function to linear
order in the electric field, chemical potential gradient, and
temperature gradient. In previous works, the emphasis had been on the
intrinsic part of the response in ferromagnetic systems, associated
with the equilibrium distribution function.
The authors claim that plugging Eq.(18) into the expressions in
Eqs.(15a,15b) for the transport currents yields a regular Hall
response in systems like bilayer graphene, which possess a nonzero
Berry curvature due to broken inversion symmetry, but display no
anomalous Hall effect due to time reversal invariance.
However, the expression for that Hall response is never written down
explicitly. From which of the various terms in Eq.(18) does it arise?
Is it from the first term which is linear in tau, or from the second
or third term, both of which are quadratic in tau?
I don't quite see how such a Hall response will come about. For time
reversal invariant systems, it follows from the Onsager relation that
the Hall response is odd in B, see
https://doi.org/10.1016/0038-1098(65)90178-X
On the other hand, for time reversal invariant systems one can easily
show that terms in the magnetoconductivity with odd powers of B have
even powers of the relaxation time tau. See, e.g., below Eq.(9) in
https://doi.org/10.1103/PhysRevB.105.045421
This seems to rule out the first term in Eq.(18) as a candidate for a
Hall response in time reversal invariant systems, since it it is
linear (odd) in tau. The other two terms in Eq.(18) are quadratic in
tau, but at linear order in B they are independent of the Berry
curvature: the Berry curvature only contributes at second order in B
via the energy denominators 1+(e/hbar)B.Omega, since there is already
a factor of B in the numerator of each of those terms.
The above considerations seem to rule out any regular (linear in B)
Hall response in time reversal invariant systems coming from the
Berry-curvature and from the nonequilibrium distribution function.
Note that there is actually a regular Hall response in time reversal
invariant systems coming from the product of the Berry curvature with
the orbital moment. It is however associated with the equilibrium part
of the distribution function, not with the nonequilibrium part. That
type of response was first identified in
http://dx.doi.org/10.1103/PhysRevLett.112.166601
[see the second term in Eq.(12) therein], and is also discussed in
https://doi.org/10.1103/PhysRevB.103.125432
That response is zero-th order in tau and linear in B, and so it
complies with the above-mentioned constraints for Hall responses in
time reversal invariant systems. The authors appear to have missed
this contribution in their analysis.
Overall, I feel that this submission does not quite meet the strict
acceptance criteria of SciPost Physics. However, I would recommend
publication in SciPost Physics Core once the manuscript has been
revised taking this report into account.
Requested changes
Concerning the physics content of the manuscript, my main request
would be to revise the discussion of the types of responses (Hall vs
Ohmic) that arise from the equilibrium and non-equilibrium parts of
the distribution function, taking into account the above comments.
Concerning the presentation, I wonder if it would make more sense to
put the paragraph containing Eq.(14) immediately after Eq.(11). The
paragraphs containing Eqs.(12,13) and Fig. 2 could then be placed in a
separate subsection III.C, where the physical interpretation for Chern
insulators is discussed.
Minor suggestions:
* * *
* The wording of the sentence "We further obtain [...]" in the
abstract suggests that the expression obtained for the energy
magnetization is new. In view of the references given below Eq.(14),
that sentence should probably be revised. The same comment applies to
the corresponding sentences in the Conclusions section.
* The inline equation below Eq.(F2) is the same as Eq.(8), no need to
write it explicitly.
Suggestions on the formatting of the manuscript:
* * *
* The authors should consider numbering all equations that are not
in-line.
* The manuscript contains a large number of footnotes, which are not
readily identified by the reader as such since they are placed in the
references. It would be helpful if the footnotes were placed on the
page where they are called from, and if their overall number was
reduced.
* A related issue is that several of the Appendices are only mentioned
in footnotes. It would be better if the Appendices were called from
the main text.
* The number of Appendices is quite large. For example, I wonder if
the two very short appendices A and H could be disposed of, and their
contents inserted at the appropriate places in the main text.
Some Appendices have a high concentration of in-line equations,
sometime with several of them in the same paragraph: see top of p. 12,
beginning of Appendix G, Appendix I.1, and below Eq.(I1). This is a
matter of personal taste, but I find this format hard to read.
* In Appendix D, the notation with "all space" or "cell" on both sides
of the inner products is somewhat cumbersome. Maybe place them on
the right-hand side only?
Typos:
* * *
* In the last paragraph of the 1st column of p. 2 , it is written
"the the validity" .
* In the first two lines below Eq. (3), remove the band indices from
the cell-periodic Bloch states. Same for Eq. (C1) and the in-line
equations below it.
* At the beginning of Sec. III.B, "The expression in Eq.(6)" should
read "The expression in Eq.(6b)".
* In footnote 79, one reference appears (twice) as a question mark.
Author: Subroto Mukerjee on 2022-09-01 [id 2777]
(in reply to Report 1 on 2022-06-27)
Response to referee 1:
We thank the referee for their very careful reading of our manuscript and for highlighting the strengths and weaknesses in it along with providing several suggestions for its improve- ments. Please find below our response to the referee’s comments, suggestions and criticisms.
The referee says: Strengths 1. Proposes an alternative way of deriving the expressions for the charge and energy mag- netization of Bloch electrons [Eqs. (8) and (14) in the manuscript]. 2. A useful and detailed Introduction, with abundant references to previous works. 3. Detailed derivations are given in the Appendices. See however point 4 in the Weaknesses section.
Our response: We thank the referee for highlighting the strengths of our manuscript.
The referee says: Weaknesses
Our response: We thank the referee for also pointing out the weaknesses in our manuscript. However, as we explain below, we respectfully disagree with the referee that these are necessarily weaknesses.
The referee says: 1. A significant part of the manuscript is devoted to rederiving in a new way results that were previously obtained in the literature, namely Eqs. (8) and (14).
Our response: It is indeed true that the expression for the energy magnetization has been derived before in the literature, a fact that we did acknowledge in our manuscript. The utility of our work is that it provides a derivation that is perhaps more straightforward to understand and adapt to other situations, based as it is on general considerations of the Einstein relation rather than specific microscopics.
The referee says: 2. The authors claim to have identified a new type of regular Hall response in time reversal invariant systems coming from the Berry curvature and from the nonequilibrium part of the distribution function given by Eq.(18). However, an actual expression for that Hall response is never written down. As detailed in the Report, I believe that such a contribution to the regular Hall response cannot exist.
Our response: As we have clearly mentioned in the manuscript, we have derived an expression for the non-equilibrium distribution function in the presence of both a non-zero Berry curvature and non-zero magnetic field. This expression can be used to obtain all charge and heat transport coefficients, in linear response in temperature and potential gradients and not just the Hall response. The terms involving the scalar product of the field and Berry curvature, i.e. B.Ω provide contributions to these coefficients (including the Hall conductivity) in addition to those already derived in the literature in the presence of either B or Ω alone. The scope of the current work is limited to deriving only the expression for the non-equilibrium distribution function and not providing detailed expressions for all the transport coefficients. Nevertheless, the reason we singled out the Hall response for mention is that it is the most widely studied of the transport coefficients in topological systems. Thus, the additional contributions to transport that we obtain in our work are perhaps best highlighted using the Hall response as an example. The referee’s contention that the additional contribution to the regular Hall response cannot exist is perhaps due to the expectation that the term “Hall response” should only be used to describe an effect that is linear in the magnetic field. If that is the case, we agree. However, we have not claimed anywhere that the additional terms arising from our calculations are linear in the magnetic field (since they are not) and have thus also assiduously avoided any mention of the Hall resistance, as opposed to the Hall response since the former term is usually employed in situations where the response is linear in the field.
The referee says: 3. The formal results obtained in the manuscript are not illustrated by any explicit calculations on model systems.
Our response: Our intention here is to provide a treatment of transport and diamagnetic currents in systems with both B ̸= 0 and Ω ̸= 0 in the most general terms that would be applicable to any system rather than focus on specific systems. In this regard, it is very similar in scope to previous papers on topological systems such as Refs. 58 and 62, which too did not perform calculations on model systems. We thus believe our work will also be a valuable contribution to the broad field like those papers.
The referee says: 4. The main text is not sufficiently self contained. I had to constantly flip back and forth between the main text and the numerous appendices (as well as the numerous footnotes) to keep up with the manuscript.
Our response: We have followed the referee’s very useful suggestions for improvement of the presenta tion of the manuscript and made modifications accordingly.
The referee says: Report The authors have shown that requiring the transport charge and energy currents to obey the Einstein relation provides an alternative route for deriving the known expressions for the charge and energy magnetization of Bloch electrons at finite temperature [Eqs.(8) and (14) in the manuscript]. This approach complements the standard approach [e.g., Ref. 58] where one starts from those expressions for the magnetizations, subtracts their curl from the expressions for the local (or total) current densities to obtain the transport current densities, and then verifies a posteriori that the resulting transport coefficients satisfy the Einstein relation. The new approach in this manuscript provides a fresh perspective. But that fact that it is only used to recover previously known results limits its impact somewhat.
Our response: We are happy to note that the referee feels that our approach provides a fresh perspective. As we have also mentioned above, we do not claim that the result obtained is entirely new. Rather, it is the approach based on a general application of the Einstein relation as opposed to an appeal to microscopics for specific models that is the focus of our work.
The referee says: In addition to the new derivation of Eqs.(8) and (14), the other main result of the manuscript is the expression in Eq.(18), valid for 2D systems, of the correction to the distribu- tion function to linear order in the electric field, chemical potential gradient, and temperature gradient. In previous works, the emphasis had been on the intrinsic part of the response in ferromagnetic systems, associated with the equilibrium distribution function. The authors claim that plugging Eq.(18) into the expressions in Eqs.(15a,15b) for the transport currents yields a regular Hall response in systems like bilayer graphene, which possess a nonzero Berry curvature due to broken inversion symmetry, but display no anomalous Hall effect due to time reversal invariance.
Our response: We thank the referee for this summary of the significance of our work. As they say, previous calculations have indeed focused on systems with a non-zero Ω due to broken time reversal but no magnetic field. Our calculation applies to general situations with non-zero Ω and non-zero B. In particular, as the referee mentions, it is of interest for systems with non-zero Ω due to broken inversion and not time reversal which require a magnetic field to produce off-diagonal transport responses.
The referee says: However, the expression for that Hall response is never written down explicitly. From which of the various terms in Eq.(18) does it arise? Is it from the first term which is linear in tau, or from the second or third term, both of which are quadratic in tau? I don’t quite see how such a Hall response will come about. For time reversal invariant systems, it follows from the Onsager relation that the Hall response is odd in B, see https://doi.org/10.1016/0038-1098(65)90178-X On the other hand, for time reversal invariant systems one can easily show that terms in the magnetoconductivity with odd powers of B have even powers of the relaxation time tau. See, e.g., below Eq.(9) in https://doi.org/10.1103/PhysRevB.105.045421 This seems to rule out the first term in Eq.(18) as a candidate for a Hall response in time reversal invariant systems, since it it is linear (odd) in tau. The other two terms in Eq.(18) are quadratic in tau, but at linear order in B they are independent of the Berry curvature: the Berry curvature only contributes at secondorder in Bv iathe energydenominators 1+(e/hbar)B⃗.Ω⃗,since there is already a factor of B in the numerator of each of those terms. The above considerations seem to rule out any regular (linear in B) Hall response in time reversal invariant systems coming from the Berry-curvature and from the nonequilibrium distribution function.
Our response: We thank the referee for taking the time to carefully analyze the expression we have obtained term by term to determine its contribution to the Hall response. As we have mentioned earlier, we are not using the term “Hall response” to mean a response that is linear in B, which is the sense in which the referee is presumably using it. Hence, their conclusion that our expression produces no Hall response. We have clearly indicated in the new version of the manuscript that the term “Hall response” does not imply linear in B response. We have also commented on the contribution of each of the different terms that appears in the expression for the non-equilibrium distribution function to the Hall response.
The referee says: Note that there is actually a regular Hall response in time reversal invariant systems coming from the product of the Berry curvature with the orbital moment. It is however associated with the equilibrium part of the distribution function, not with the nonequilibrium part. That type of response was first identified in http://dx.doi.org/10.1103/PhysRevLett.112.166601 [see the second term in Eq.(12) therein], and is also discussed in https://doi.org/10.1103/PhysRevB.103.125432 That response is zeroth order in tau and linear in B, and so it complies with the above- mentioned constraints for Hall responses in time reversal invariant systems. The authors appear to have missed this contribution in their analysis.
Our response: We thank the referee for bringing up this point. The intrinsic equilibrium contribution the Hall response is present in our analysis through the second term of Eqn. 17a. We had not highlighted this fact earlier since the intrinsic equilibrium Hall response has been a subject of detailed investigation already as the referee points out and we did not have anything to add to what is known. Our focus, as mentioned earlier, is on the contribution to transport arising from the non-equilibrium distribution function. However, in the interest of clarity, we have now explicitly mentioned exactly how our calculation also includes the equilibrium response after En. 19.
The referee says: Overall, I feel that this submission does not quite meet the strict acceptance criteria of SciPost Physics. However, I would recommend publication in SciPost Physics Core once the manuscript has been revised taking this report into account.
Our response: We respectfully disagree that our work does not meet the criteria for publication in Sci- Post Physics. We hope we have managed to convey from our responses above that both of the important results of our work are of significance to the field of transport in topologi- cal systems. And that our results are correct and produce the already studied equilibrium Hall response in addition to the hitherto unexplored non-equilibrium response for Ω ̸= 0 and B ̸= 0. To recap the significance of our work: The energy magnetization is a rather opaque quantity compared to its charge counterpart. We feel that our derivation based on general considerations related to the Einstein relation simplifies its understanding more than those that appeal to microscopics and currently exist in the literature. Further, our expression for the non-equilibrium distribution function in systems with Ω ̸= 0 and B ̸= 0 is a new result, which can be used to obtain any charge or heat transport coefficient. We have clarified that this distribution function does indeed provide a Hall response, just not one that is linear in B, and our calculation also contains the intrinsic equilibrium Hall response. These seem to have been the referee’s main concerns about the validity of our results and we believe that we have addressed them.
The referee says: Requested Changes Concerning the physics content of the manuscript, my main request would be to revise the discussion of the types of responses (Hall vs Ohmic) that arise from the equilibrium and non-equilibrium parts of the distribution function, taking into account the above comments. Concerning the presentation, I wonder if it would make more sense to put the paragraph containing Eq.(14) immediately after Eq.(11). The paragraphs containing Eqs.(12,13) and Fig. 2 could then be placed in a separate subsection III.C, where the physical interpretation for Chern insulators is discussed. Minor suggestions • The wording of the sentence ”We further obtain [...]” in the abstract suggests that the expression obtained for the energy magnetization is new. In view of the references given below Eq.(14), that sentence should probably be revised. The same comment applies to the corresponding sentences in the Conclusions section. • The inline equation below Eq.(F2) is the same as Eq.(8), no need to write it explicitly.
Our response: We thank the referee for these suggestions and have now incorporated them in the manuscript.
The referee says: Suggestions on the formatting of the manuscript: 1. The authors should consider numbering all equations that are not in-line. 2. The manuscript contains a large number of footnotes, which are not readily identified by the reader as such since they are placed in the references. It would be helpful if the footnotes were placed on the page where they are called from, and if their overall number was reduced. 3. A related issue is that several of the Appendices are only mentioned in footnotes. It would be better if the Appendices were called from the main text. 4. The number of Appendices is quite large. For example, I wonder if the two very short appendices A and H could be disposed of, and their contents inserted at the appropriate places in the main text. 5. Some Appendices have a high concentration of in-line equations, sometime with several of them in the same paragraph: see top of p. 12, beginning of Appendix G, Appendix I.1, and below Eq.(I1). This is a matter of personal taste, but I find this format hard to read. 6. In Appendix D, the notation with ”all space” or ”cell” on both sides of the inner products is somewhat cumbersome. Maybe place them on the right-hand side only?
Our response: We thank the referee for these formatting suggestions and have now incorporated them in the manuscript. Please see the list of changes for details.
The referee says: Typos 1. In the last paragraph of the 1st column of p. 2 , it is written ”the the validity” . 2. In the first two lines below Eq. (3), remove the band indices from the cell-periodic Bloch states. Same for Eq. (C1) and the in-line equations below it. 3. At the beginning of Sec. III.B, ”The expression in Eq.(6)” should read ”The expression in Eq.(6b)”. 4. In footnote 79, one reference appears (twice) as a question mark. 7
Our response: We thank the referee for pointing out these typos and have now corrected them and a few others that we found upon rereading the manuscript.
Author: Subroto Mukerjee on 2022-09-01 [id 2778]
(in reply to Report 2 on 2022-07-21)We thank the referee for their very careful reading of our manuscript and for highlighting the strengths and weaknesses in it along with providing several suggestions for its improvements. Please find below our response to the referee’s comments, suggestions and criticisms.
The referee says :
Strengths: The strength of the manuscript lies on the detailed derivation of the charge and energy magnetization of Bloch electrons using an alternative approach based on Einstein relation.
Our response: We thank the referee for highlighting the strengths of our manuscript.
The referee says: Weaknesses
Our response: We thank the referee for also pointing out the weaknesses in our manuscript. However, as we explain below, we respectfully disagree with the referee that these are necessarily weaknesses.
The referee says: 1. Although the authors provide detailed derivation of charge and energy magnetization, they have not applied their formalism to any particular system.
Our response: As we have also mentioned in our response to a similar comment by referee 1, our intention here is to provide a treatment of transport and diamagnetic currents in systems with both B ̸= 0 and Ω ̸= 0 in the most general terms that would be applicable to any system rather than focus on specific systems. In this regard, it is very similar in scope to previous papers on topological systems such as Refs. 58 and 62, which too did not perform calculations on model systems. We thus believe our paper will also be a valuable contribution to the broad field like those other papers.
The referee says: 2. They do not provide the explicit expression of Hall conductivity (which is one of their main results) in the main text.
Our response: This point too has been brought up by referee 1. As we have clearly mentioned in the manuscript, we have derived an expression for the non-equilibrium distribution function in the presence of both a non-zero Berry curvature and non-zero magnetic field. This expression can be used to obtain all charge and heat transport coefficients, in linear response in temperature and potential gradients and not just the Hall response. The terms involving the scalar product of the field and Berry curvature, i.e. B.Ω provide contributions to these coefficients (including the Hall conductivity) in addition to those already derived in the literature in the presence of either B or Ω alone. The scope of the current work is limited to deriving only the expression for the non-equilibrium distribution function and not providing detailed expressions for all the transport coef-ficients. Nevertheless, the reason we singled out the Hall response for mention is that it is the most widely studied of the transport coefficients in topological systems and so the existence of the additional contributions to transport in general is perhaps best highlighted using the Hall response as an example.
The referee says: Report The authors provide an alternative approach (based on the Einstein relation) to derive charge and energy magnetization. They use the approach only to recover old results. There- fore, I feel there are not many new results in the manuscript. Another point is that the authors did not compare their approach with the old one (why one should use their approach to derive charge and energy magnetization?).
Our response: We thank the referee for their comments. As we have already emphasized, our approach is based on general considerations related to the Einstein relations rather than based on specifics of the microscopics of model systems. The energy magnetization is a rather opaque quantity compared to its charge counterpart. We feel that our derivation based on the above general considerations simplifies its understanding compared to those that appeal to microscopics, which currently exist in the literature. We have added a line to this effect in the modified manuscript.
The referee says: The authors claim that the non-equilibrium distribution function obtained in this work can generate a new type of regular Hall response in time-reversal invariant samples with a non-zero Berry curvature. Although the authors have presented detailed derivation for the rest of the calculations, however, they did not provide the expression of Hall conductivity anywhere. Regarding that I have few questions
Our response: Please find below the responses to the questions brought up by the referee.
The referee says: 1. Please write down the explicit expression of the different components of the Hall conductivity.
Our response: As we have mentioned earlier, the Hall response is not the focus of this work. Rather, it is the expression for the non-equilibrium distribution function in the presence of both Ω ̸= 0 and B ̸= 0 from which, in principle, one can derive all the heat and charge trans- port coefficients by integrating with appropriate kernels. The Hall response is non-linear in the magnetic field (as also pointed out in our response to referee 1) and thus does not have a simple closed form expression. Nevertheless, we have added a discussion after Eqn. 24 on page 7 explaining the effect of each of the terms to Hall response.
The referee says: 2. How one can distinguish different Hall components from each other in experiment?
Our response: The different components of the Hall response contribute with different functional dependences on the magnetic field. Since the field in the non-equilibrium distribution functions appears only through the combination e Ω(k).B, with the momentum k be- ing integrated over, the functional dependence on B will be determined by the precise dependence of Ω(k) on k. There is thus no simple way to parse the contributions experimentally without doing an explicit calculation for a specific microscopic system. However, a general feature of the field dependence coming from the presence of the term e Ω(k).B is that it is non-linear in B as opposed to the Hall response in a regular metal or even Chern insulator, in which it is linear. The deviation from linearity in the field could be an experimental signature of the contribution from the non-equilibrium distribution function that we have derived.
The referee says: 3. What is τ scaling for the new regular Hall effect? What are symmetry constraints for this Hall conductivity? Our response: In the discussion added after Eqn. 24, we have commented on the τ dependence of the different terms in the expression for the Hall conductivity. We are not sure what the referee means by “symmetry constraints” for this Hall conductivity but it obeys the usual Onsager symmetry relations like the Hall conductivity in any other situation.
The referee says: 4. The authors should calculate the new regular Hall conductivity for bilayer graphene system and then compare with other coexisting Hall components.
Our response: As we have emphasized before, the Hall conductivity is not the focus of this work. It is the calculation of the non-equilibrium distribution function from which all charge and energy transport coefficients can, in principle, be obtained. The Hall conductivity is just one of these, even if it is the most commonly studied one in the literature. A calculation of only the Hall conductivity detracts from the significance of the distribution function to the calculation of all transport coefficients. At the same time, a calculation of all the transport coefficients is too involved to be within the scope of what we are trying to achieve here. Also, as mentioned before, our work develops a general formalism without focusing on any specific microscopic system along the lines of other works such as Refs. 58 and 62. We mention bilayer graphene since it is an example of a system in which Ω ̸= 0 due to broken inversion and not time reversal symmetry and thus is of the type that our formalism might be useful to study. However, there could be other such systems and so we do not think that studying this specific system as an example is necessary. This is more so because the calculation for Dirac materials can be technically quite involved due to the singular behavior of the Berry curvature, as we have pointed out in the discussion added after Eqn. 24. It should thus be the subject of a self-contained investigation of its own well beyond the scope of our general treatment here.
The referee says: 5. The authors consider relaxation time approximation. Please discuss the regime of validity of this approximation.
Our response: The relaxation time approximation employed here has the same regime of validity as in all other such studies based on a non-interacting description of topological systems. The main assumption is that transport can be described in terms of non-interacting electrons or quasiparticles with momentum relaxation in the bulk due to interactions with other degrees of freedom like phonons and impurities. The approximation breaks down in the so-called “ hydrodynamic regime” in which the electrons collectively behave as a fluid with bulk momentum conservation. This regime is usually accessible in a rather restricted region of electron density and temperature in which the time scale obtained from the shear viscosity of the electron fluid is shorter than that from interactions between the electrons and other degrees of freedom. The relaxation time approximation thus applies in a much broader regime of parameters.
The referee says: Requested changes Major changes 1. The authors should write down the explicit expression of different Hall components using their formalism and explain how they can be distinguished from each other in experiment. 2. They should calculate the new regular Hall response for the bilayer graphene system.
Our response: We respectfully disagree with the referee that either of these changes is required since, as we have emphasized above, our work is about the derivation of the non-equilibrium distribution function from which all transport coefficients can be derived and not just the Hall response. Further, our formalism is also meant to be general in scope and not focused on any particular system. We have added a short discussion after Eqn. 24 on why the calculation of the Hall conductivity for specific Dirac systems can be technically involved. We thus think it should be the subject of a self-contained investigation of its own and it is not very reasonable to expect us to fit it within the scope of our general treatment here.
The referee says: Minor changes 1. Appendix A and Appendix J can be included in the main text. 2. Appendix B is very well known. So the authors can remove it from the manuscript. 3. There are several typos and grammatical errors in the manuscript which authors should fix in the revised version. 4. Some of the equations in the main text do not align properly. The authors should consider them to align properly within the margin.
Our response: We thank the referee for suggesting these minor changes, which we have now incorporated in the manuscript.

---

## Round 1 · Referee Report · Anonymous · 2022-7-21

Strengths
The strength of the manuscript lies on the detailed derivation of the charge and energy magnetization of Bloch electrons using an alternative approach based on Einstein relation.
Weaknesses
1) Although the authors provide detailed derivation of charge and energy magnetization, they have not applied their formalism to any particular system.
2) They do not provide the explicit expression of Hall conductivity (which is one of their main results) in the main text.
Report
The authors provide an alternative approach (based on the Einstein relation) to derive charge and energy magnetization. They use the approach only to recover old results. Therefore, I feel there are not many new results in the manuscript. Another point is that the authors did not compare their approach with the old one (why one should use their approach to derive charge and energy magnetization?).
The authors claim that the non-equilibrium distribution function obtained in this work can generate a new type of regular Hall response in time-reversal invariant samples with a non-zero Berry curvature. Although the authors have presented detailed derivation for the rest of the calculations, however, they did not provide the expression of Hall conductivity anywhere. Regarding that I have few questions
i) Please write down the explicit expression of the different components of the Hall conductivity.
ii) How one can distinguish different Hall components from each other in experiment?
iii) What is \tau scaling for the new regular Hall effect? What are symmetry constraints for this Hall conductivity?
iv) The authors should calculate the new regular Hall conductivity for bilayer graphene system and then compare with other coexisting Hall components.
v) The authors consider relaxation time approximation. Please discuss the regime of validity of this approximation.
I feel the current version of the manuscript does not meet the criteria for the publication in SciPost.
Requested changes
Major Changes:
1) The authors should write down the explicit expression of different Hall components using their formalism and explain how they can be distinguished from each other in experiment.
2) The should calculate the new regular Hall response for the bilayer graphene system.
Minor Changes:
1) Appendix A and Appendix J can be included in the main text.
2) Appendix B is very well known. So the authors can remove it from the manuscript.
3) There are several typos and grammatical errors in the manuscript which authors should fix in the revised version.
4) Some of the equations in the main text do not align properly. The authors should consider them to align properly within the margin.

---

## Round 2 · Referee Report · Anonymous (Referee 1) · 2022-9-27

Report

Warnings issued while processing user-supplied markup:

  • Inconsistency: plain/Markdown and reStructuredText syntaxes are mixed. Markdown will be used.
    Add "#coerce:reST" or "#coerce:plain" as the first line of your text to force reStructuredText or no markup.
    You may also contact the helpdesk if the formatting is incorrect and you are unable to edit your text.

The resubmitted manuscript does not differ substancially from that of the original submission, and I feel that it still does not meet the stringent novelty and clarity criteria for publication in SciPost Physics. I mantain the opinion that it could be published in SciPost Physics Core, once the issues mentioned below are resolved.

Concerning novelty, I would like to add the following remarks, which should probably have been included in my original report. The authors point to Eq.(24) as a significant new result. However, there have been several works in recent years dealing with the perturbative solution of the "Berry-Boltzmann" equations to higher orders in applied fields, and these are not referenced in the manuscript. See, for example, Sec. 3 of the following review paper:

https://doi.org/10.1007/s11467-019-0887-2

How does Eq.(24) in the manuscript relate to Eqs.(67-69) in the above paper? The perturbative expansion in Eq.(64) therein seems to provide a systematic procedure for obtaining corrections to the distribution function in powers of $E$ and $B$. Why did the authors not use this standard approach? It would be important to make contact with the existing literature on evaluating nonlinear currents starting from the Berry-Boltzmann equations, in order to understand in what ways the current work goes beyond previous ones, and whether or not there is agreement with the published results.

I am also skeptical about the claim, made in the second-to-last paragraph before the Conclusions, that an expansion in powers of $B$ of the conductivity cannot be made for Dirac materials . At least in the case of the linear anomalous Hall effect in 3D Weyl semimetals, it is well known that the intrinsic anomalous Hall conductivity behaves perfectly smoothly as the chemical potential is tuned across a Weyl node, in spite of the singular behavior of the Berry curvature at the node. If the authors claim that the situation is fundamentally different in Dirac materials, they should explain why that is so.

The authors object to the suggestions of applying their formalism to a specific system or model Hamiltonian, and to discussing in more detail what are the novel aspects of the Hall or Ohmic responses that come out of it. However, the absence of the concrete physical insights that such additions to the manuscript might bring (for example, an illustration of the claim that an expansion in powers of $B$ cannot be made for Dirac materials) is one of the arguments for publishing in SciPost Physics Core rather than in SciPost Physics.

Concerning presentation matters, I still find the manuscript quite hard to follow, with the main text relying too heavily on long appendices to arrive at the main results (this goes against the claim made by the authors that their approach is simpler and more transparent than those in the previous literature). The number of footnotes disrupting the flow of the main text is still too high, and numerous typos remain. Finally, I would have liked to see a more polished revised version of the manuscript.

The appendices are a mixture of rederivations of well-known results with new derivations. It would be useful to remove or strongly condense the former, so that the reader can more easily identify and focus on the latter. At times, the appendices look like research notes that have not been fully converted into publishable material. Finally, most of the equations in the appendices remain unnumbered. That is a problem already for writing this report, since I need to refer to several unnumbered equations.

In the following I make some specific suggestions concerning the issues raised above, along with other miscellaneous comments.

In the abstract, write "in Chern insulators" after "chiral edge states" (and insert a comma after the word "condition" a bit earlier).

Footnotes 1 and 3 are identical. I suggest removing both and replacing them with the following change in the 2nd line of the 2nd column of p.1:

"by the relation $j^e=-ej^n$."

could become something like

"by the relation $j^e=-ej^n$, where $-e$ is the electron charge."

The authors have now clarified that by "regular Hall" response they do not mean a Hall response linear in $B$ (which is the meaning I was associating with those words). However, they do so in footnote 13, by which time the expression "regular Hall" has appeared nine times already in the manuscript (starting with the abstract). That explanation should be given at the earliest occurrence of the expression "regular Hall" in the main text, namely on p.2.

I do not understand how footnote 4 relates to the text about spin degeneracy from where it is called. Seems to me that the content of that footnote is instead related to the equation

$\epsilon_k = \epsilon_0(k) - m_k.B$

which by the way is written twice as an inline equation: first below Eq.(3), and then again in footnote 5. I suggest writing that equation only once, as a numbered equation below Eq.(3). Then, instead of footnote 4 simply say in the main text that $m_k=m^\text{orb}_k + m^\text{spin}_k$.

Coming back to the sentence

"The factor of 2 is for the spin degeneracy."

that appears below Eq.(4). The abstract mentions "time-reversal invariant samples with a non-zero Berry curvature". Such materials must break inversion symmetry, but then the bands are no longer spin degenerate. Should that sentence be removed, along with all the factors of 2 that it implies throughout the text?

Why write Eq. (5) for the intrinsic orbital moment $m_k$? That equation is never used in practice to evaluate $m_k$. The more useful expression is the one in terms of the antisymmetric part of the unnumbered boxed equation on top of p.13 in Appendix B. One possibility would be to write two equalities in Eq.(5). The first one would be the current Eq.(5), and the second would be the more useful "quantum-geometric" expression involving a cross product of $k$ derivatives cell-periodic Bloch states, with $(H-E)$ sandwiched in between.

I am also puzzled about the purpose of Appendix B. Is it to rederive the well-known quantum-geometric formula for $m_k$? If so, you should write down clearly that formula somewhere (possibly in Eq.(5), as suggested mentioned). You should also explain why there is the need for presenting a new derivation of a rather well known result. More specific comments about Appendix B can be found below.

In writing Eq.(6), you specialize to $B=0$. I do not understand the motivation for doing so, given that in the abstract you emphasize the case where $B$ is nonzero. Instead of relegating the $B\not= 0$ case to footnote 5, what about replacing or supplementing Eq.(6) with an equation in the main text that is valid for $B\not= 0$?

Appendices A and B are called from footnote 6. I suggest moving that footnote to the main text, somewhere below Eq.(6). Seems to me that it is only Appendix A that should be called there, and that Appendix B is only really invoked directly from Appendix A, not from the main text.

The title of Sec. III is rather long, and it is unnecessarily repeated in the titles of the subsections. Here are some suggestions for streamling the titles:

III. Charge and energy magnetizations from the Einstein relation

III.A Charge magnetization

III.B Energy magnetization

The opening paragraph of Sec. III.A contains two almost identical sentences, namely the first and the fourth:

"In this section we demonstrate that the known expression of the charge magnetization can be recovered by demanding that the Einstein relation holds for the electric transport current."

"Here we show that the same expression of the charge magnetization can be obtained by demanding that the Einstein relation holds for the electric transport current."

Above Eq.(8), the authors refer to Appendix C for its derivation. That appendix is unnecessarily long, with many equations that already appear elsewhere in the manuscript. Once that redundancy is removed the appendix becomes quite short, and it would improve the readibility of the manuscript if it was brought to the main text.

Concerning Appendix C itself:

The first equality in Eq.(C1) is identical to Eq.(A4), which is in turn identical to Eq.(6a).

Eq. (C2) is the same as Eq.(7).

The first equality in Eq.(C3) is the same as the inline equation below Eq.(7).

The closing sentence of Appendix C should end with something like ", which gives Eq.(9)".

With these changes Appendix C can be made quite short, and inserted in the main text of Sec. III.A.

Incidentally, the title of Appendix C has "derivation of the" appearing twice, which is awkward.

Immediately below Fig. 2, the long inline equation should probably be a numbered equation. In the following sentence, "the known expressions [56,63,64] for the charge magnetization" refers, I presume, to Eq.(9). If that is the case, the sentence should refer to that equation directly instead of refering to the literature (which has already been cited in connection with that equation).

On the 2nd column of p. 5, the second line containts "is appears". Remove "is"?

The long paragraph where the above typo occurs contains a very large number of parenthetical remarks, and does not read well.

The two-sentence paragraph below the one discussed above mentions "this quantity". Which quantity? The one given by Eq.(15)? Please specify.

Similar problem with the short paragraph below the one containing Eq.(23). Therein, "this term" should be specified clearly. Presumably, it is the second term on the left-hand side of Eq.(23).

(Incidentally, the acronym LHS is used without having been defined.)

The sentence straddling pages 5 and 6 could be improved. As written, it sounds as if the results obtained from the different methods that are listed were different from one another. Instead, I believe the authors want to say that Eq.(16) agrees with the result obtained in the previous literature in different ways: using gauge theories of gravity, or by introducing either an inhomogeneous disorder field or a gravitomagnetic field.

In footnote 11, a quantity $\omega$ appears without being defined. Is it the same as the cyclotron frequency $\omega_c$ in footnote 12? Or is it the frequency of an applied ac electric field?

Concerning the long paragraph on the 2nd column of p.5, where Eq.(24) for the nonequilibrium distribution function is discussed. As already mentioned, it would be important to relate Eq.(24) to similar expressions that are givern in the literature. At present, that long paragraph makes almost no contact with prior work. And yet, there has been a lot of activity recently in calculating nonequilibrium distribution functions in the presence of external fields and Berry curvature. Itt would therefore be important to provide some perspective on Eq.(24), by making contact with the literature. In addition to the review paper indicated earlier, here are some other papers that might be relevant:

https://doi.org/10.1103/PhysRevB.95.165135

https://doi.org/10.1103/PhysRevB.105.045421

https://doi.org/10.1103/PhysRevB.105.205126

The appendices need to be heavily edited, as they read more like research notes than publication-grade material. Every equation should be numbered, and redundancies should be identified and removed.

In particular, the equation below (A2) defining the quantity ${\cal M}_{ij}$ needs to be numbered and refered to later on. It is repeated in the title of subsection 1 of Appendix B, and in fact that quantity features prominently throughout Appendix B.

(I am also unsure if the quantity ${\cal M}_{\mu\nu}$ introduced in the first sentence of Appendix B.1.d the same as ${\cal M}_{ij}$.)

Several of the titles of appendices, or subsections thereof, are quite long and contain long equations. See, for example, Appendices B.1.c and B.1.d.

Above Eq.(A4) [which, as already mentioned, is the same as Eq.(6a)] it is written: "We want to calculate the properties of the anomalous Hall effect, where $B=0$." This seems to contradict what is stated in the abstract, where the focus seems to be on the $B\not= 0$ case.

Specific comments about Appendix B:

The (unnumbered) multiline equation on the bottom of p.10 is just the well-known Hellmann-Feynman theorem, which in the present context gives the intraband velocity. It may not be necessary to spell out its derivation in such detail.

Once the long equation on top of p. 11 is numbered, it does not have to be repeated as an inline equation on top of p. 13 (in the sentence "Integrating over [...]"), but simply referred to.

On p.13, the sentence starting with "In this case [...]" contains seven inline equations, which make it quite unwieldy. There is also an "is" missing before the first occurrence of "Hermitian".

The second half of p.12 is quite confusing: the "quantity without the commutator" is introduced, and it would probably help to associate a symbol with it. Then there are several equations that start with an equal sign, and one could insert that symbol on the (nonexisting) left-hand side of those equations. When we get to the first boxed equation on p.13, are we still talking about that quantity without a symbol? If so, the proposed symbol could be used there once again.

I would think that the end result of this entire appendix should be the quantum-geometric expression for the intrinsic orbital moment. However, that equation is never written down explicitly.

Below Eq.(F1), there is a

"(see Appendix ?? for justification)"

Please insert the correct appendix number in place of "??".

  • validity: -
  • significance: -
  • originality: -
  • clarity: -
  • formatting: -
  • grammar: -

Author:  Subroto Mukerjee  on 2023-05-12  [id 3665]

(in reply to Report 1 on 2022-09-27)

We thank the referee for going through our revised manuscript and submitting their second report. We appreciate the extreme care with which they have read our revised submission and their suggestions for the improvement of our presentation. We have incorporated most of these suggestions (Please see this list of changes). We also apologise for the delay in responding which was partly on account of the large number of changes we have made to the manuscript upon the referee's suggestions. We believe that the changes have improved the presentation of our work.

The referee says that our manuscript does not meet the stringent novelty and clarity criterion for publication in SciPost Physics. We hope that by making the changes suggested by the referee, we have addressed the questions about clarity. As for novelty, the referee has pointed us to a review paper, which they claim contains a summary of previous work on the perturbative solution of the Berry-Boltzmann equations to high orders in applied fields and which we have not cited. The referee says that the existence of this paper detracts from our claim of the novelty of our approach. We thank the referee for bringing this paper to our notice, which we had indeed missed earlier. However, as we explain below, there are questions about the correctness of the expressions in this paper. Moreover, the approach described in it does not account for temperature gradients, the inclusion of which is an important aspect of our calculations. We thus believe that our approach is indeed novel within the scope of general transport theory.

A further criticism of the referee is that we do not perform a calculation for a specific microscopic model and this fact disqualifies our paper from consideration for publication on SciPost Phys. We reiterate that our intention is to present a calculation of the energy magnetization in the most general terms. And, as we have pointed out in our previous response, there have been several papers in the recent past, which too have presented general treatments of different aspects of transport theory without performing calculations for specific models. While we once again, respectfully disagree with the referee that this should be a disqualification for publication on SciPost Phys., we are willing to accept the referee's recommendation of publication in Sci Post Phys. core instead.

Please find below a detailed point by point response to the referee's comments. We hope that this version of the manuscript will be considered acceptable for publication in Sci Post Phys. Core.

The referee says: The resubmitted manuscript does not differ substantially from that of
the original submission, and I feel that it still does not meet the
stringent novelty and clarity criteria for publication in SciPost Physics.
I mantain the opinion that it could be published in SciPost Physics
Core, once the issues mentioned below are resolved.

Our response: We thank the referee for going through our revised manuscript and submitting their second report. We appreciate the extreme care with which they have read our revised submission and their suggestions for the improvement of our presentation. We have incorporated most of these suggestions (Please see this list of changes).

The referee says that our manuscript does not meet the stringent novelty and clarity criterion for publication in SciPost Physics. We hope that by making the changes suggested by the referee, we have addressed the questions about clarity. As for novelty, the referee has pointed us to a review paper, which they claim contains a summary of previous work on the perturbative solution of the Berry-Boltzmann equations to high orders in applied fields and which we have not cited. The referee says that the existence of this paper detracts from our claim of the novelty of our approach. We thank the referee for bringing this paper to our notice, which we had indeed missed earlier. However, as we explain below, there are questions about the correctness of the expressions in this paper. Moreover, the approach described in it does not account for temperature gradients, the inclusion of which is an important aspect of our calculations. We thus believe that our approach is indeed novel within the scope of general transport theory.

A further criticism of the referee is that we do not perform a calculation for a specific microscopic model and this fact disqualifies our paper from consideration for publication on SciPost Phys. We reiterate that our intention is to present a calculation of the energy magnetization in the most general terms. And, as we have pointed out in our previous response, there have been several papers in the recent past, which too have presented general treatments of different aspects of transport theory without performing calculations for specific models. While we once again, respectfully disagree with the referee that this should be a disqualification for publication on SciPost Phys., we are willing to accept the referee's recommendation of publication in Sci Post Phys. core instead.

Please find below a detailed point by point response to the referee's comments. We hope that this version of the manuscript will be considered acceptable for publication in Sci Post Phys. Core.

The referee says: Concerning novelty, I would like to add the following remarks, which
should probably have been included in my original report. The authors
point to Eq.(24) as a significant new result. However, there have been
several works in recent years dealing with the perturbative solution
of the "Berry-Boltzmann" equations to higher orders in applied fields,
and these are not referenced in the manuscript. See, for example,
Sec. 3 of the following review paper:

https://doi.org/10.1007/s11467-019-0887-2

How does Eq.(24) in the manuscript relate to Eqs.(67-69) in the above
paper? The perturbative expansion in Eq.(64) therein seems to provide
a systematic procedure for obtaining corrections to the distribution
function in powers of E and B. Why did the authors not use this
standard approach? It would be important to make contact with the
existing literature on evaluating nonlinear currents starting from the
Berry-Boltzmann equations, in order to understand in what ways the
current work goes beyond previous ones, and whether or not there is
agreement with the published results.

Our response: We thank the referee for pointing this reference out to us, which we had indeed missed in the previous versions of our manuscript and which we have now cited. The paper in question does indeed describe a perturbative expansion of the distribution function in powers of $E$ and $B$. Note however, that the ``mechanistic'' approach of this paper and those cited in it cannot account for temperature gradients (unless a gravitational potential is included in the semi-classical equations of motion a la Luttinger), which is a crucial aspect of our calculation. Thus, the existence of this paper and the others cited in it does not detract from the novelty of our work. The referee asks how Eqn. 24 in our manuscript relates to Eqns. 67-69 of the review paper. The expressions in Eqns. 67-69 appears to have mistakes. For instance, there are no factors of $1 + {\bf B}. {\Omega}$ and higher powers of it in the denominators of $f_2$ and $f_3$, which should be present. Since the paper claims to obtain the non-equlibrium distribution function to non-linear order in the magnetic field, the presence of these factors is crucial for the correctness of the expressions. We are working only to linear order in the electric field and have verified that our expressions match those in the review paper (modulo the missing factors of ($1 + {\bf B}. {\Omega}$ and higher powers) to that order. We also respectfully disagree with the referee that this is a standard approach. If it were, it would be found in standard references on transport theory, which it is not. However, the approach is certainly a useful one. Nevertheless, we would like to point out that the perturbation technique employed here is different from the one we have used. In our approach ${\bf v} \times {\bf B}$ is a perturbation, whereas in the above reference ${\bf E}$ and ${\bf v} \times {\bf B}$ are treated on equal footing.

The referee says: I am also skeptical about the claim, made in the second-to-last
paragraph before the Conclusions, that an expansion in powers of B
of the conductivity cannot be made for Dirac materials . At least in the
case of the linear anomalous Hall effect in 3D Weyl semimetals, it is
well known that the intrinsic anomalous Hall conductivity behaves
perfectly smoothly as the chemical potential is tuned across a Weyl
node, in spite of the singular behavior of the Berry curvature at the
node. If the authors claim that the situation is fundamentally
different in Dirac materials, they should explain why that is so.

Our response: Suppose we have a single Dirac cone in the Brillouin Zone (BZ). Suppose we want to integrate the expression $1/(1 + e B.\Omega/ \hbar)$ over the BZ.
If we expand it in powers of B and integrate order by order, then it will be finite upto linear order in B (since the integral of Ω is finite – the Chern number). However, if we want to calculate the terms corresponding to higher powers of B, the corresponding integrals will blow up (e.g. $\Omega$^2 integrated over the BZ will diverge).
We did not make any claim that the Hall coefficient of the Weyl semimetal will diverge. Our point is, since the solution of the Boltzmann Transport Equation contains the expression $1/(1 + e B.\Omega/\hbar)$, it needs to be treated carefully when we integrate over the BZ. It should also be noted that in the limit that the chemical potential (or Fermi energy) approaches a Dirac (or Well) point, the semiclassical approximation breaks down. This is because the density of states goes to zero, which causes scattering times used in the semiclassical approximation to, in principle, diverge from a consideration of the Fermi golden rule. This indicates that in this limit a full quantum treatment is required and this quantum intervention produces smooth behavior of transport coefficients, as the referee points out, even though the Berry curvature is singular at the Dirac point. In the presence of a magnetic field, the full quantum treatment requires a consideration of the Landau bands produced by the field. We have now added a brief discussion of this to the first paragraph on page 8.

The referee says: The authors object to the suggestions of applying their formalism to a
specific system or model Hamiltonian, and to discussing in more detail
what are the novel aspects of the Hall or Ohmic responses that come
out of it. However, the absence of the concrete physical insights that
such additions to the manuscript might bring (for example, an
illustration of the claim that an expansion in powers of B
cannot be made for Dirac materials) is one of the arguments for publishing in
SciPost Physics Core rather than in SciPost Physics.

Our response: As we mentioned in our previous response, the aim here is to present a general formalism for magnetotransport in systems with non-trivial Berry curvature without reference to microscopic in the spirit of previous work like in in Refs. 58 and 62. While we still maintain that this is a worthwhile exercise, suitable for publication in SciPost Phys., we are happy to have our manuscript published in SciPost Phys. core instead as the referee suggests.

The referee says: Concerning presentation matters, I still find the manuscript quite
hard to follow, with the main text relying too heavily on long
appendices to arrive at the main results (this goes against the claim
made by the authors that their approach is simpler and more
transparent than those in the previous literature). The number of
footnotes disrupting the flow of the main text is still too high, and
numerous typos remain. Finally, I would have liked to see a more
polished revised version of the manuscript.

The appendices are a mixture of rederivations of well-known results
with new derivations. It would be useful to remove or strongly
condense the former, so that the reader can more easily identify and
focus on the latter. At times, the appendices look like research notes
that have not been fully converted into publishable material. Finally,
most of the equations in the appendices remain unnumbered. That is a
problem already for writing this report, since I need to refer to
several unnumbered equations.

In the following I make some specific suggestions concerning the
issues raised above, along with other miscellaneous comments.

Our response: We thank the referee for their suggestions for improving the presentation. We have made modifications to the manuscript in accordance with these suggestions as we detail below. We have numbered all the equations in the appendices and improved the presentation. We have also removed some of the appendices and stated the goal of each appendix at its beginning.

The referee says: In the abstract, write "in Chern insulators" after "chiral edge
states" (and insert a comma after the word "condition" a bit earlier).

Our response: We have made this change.

The referee says: Footnotes 1 and 3 are identical. I suggest removing both and replacing
them with the following change in the 2nd line of the 2nd column of
p.1:

"by the relation je=−ejn." could become something like
"by the relation je=−ejn, where −e
is the electron charge."

Our response: We have made this change

The referee says: The authors have now clarified that by "regular Hall" response they do
not mean a Hall response linear in B
(which is the meaning I was associating with those words). However, they do so in footnote 13, by
which time the expression "regular Hall" has appeared nine times
already in the manuscript (starting with the abstract). That
explanation should be given at the earliest occurrence of the
expression "regular Hall" in the main text, namely on p.2.

Our response: We have added this explanation to paragraph 2 of the second page.

The referee says: I do not understand how footnote 4 relates to the text about spin
degeneracy from where it is called. Seems to me that the content of
that footnote is instead related to the equation $\epsilon_k = \epsilon_0(k)-m_k.B$
which by the way is written twice as an inline equation: first below
Eq.(3), and then again in footnote 5. I suggest writing that equation
only once, as a numbered equation below Eq.(3). Then, instead of
footnote 4 simply say in the main text that $m_k = m^{orb}_k +m^{spin}_k$.

Our response: The footnote has been removed, and has been replaced by an explanation in section II of how the band energy is modified due to spin and orbital angular momentum.

The referee says: Coming back to the sentence

"The factor of 2 is for the spin degeneracy."

that appears below Eq.(4). The abstract mentions "time-reversal
invariant samples with a non-zero Berry curvature". Such materials
must break inversion symmetry, but then the bands are no longer spin
degenerate. Should that sentence be removed, along with all the
factors of 2 that it implies throughout the text?

Our response: We have now replaced factors of 2 with sums over spins, $\Sigma_s$.

The referee says: Why write Eq. (5) for the intrinsic orbital moment $m_k$? That equation
is never used in practice to evaluate $m_k$. The more useful expression
is the one in terms of the antisymmetric part of the unnumbered boxed
equation on top of p.13 in Appendix B. One possibility would be to
write two equalities in Eq.(5). The first one would be the current
Eq.(5), and the second would be the more useful "quantum-geometric"
expression involving a cross product of $k$
derivatives cell-periodic
Bloch states, with $(H−E)$
sandwiched in between.

Our response: We are using the formula in Eq.(5) to motivate the analogy of the orbital angular momentum with the usual notion of angular momentum, $r \times p$. While the “quantum geometric” formula is useful, we do not actually need it in our paper, as we are not doing any specific band structure calculation.

The referee says: I am also puzzled about the purpose of Appendix B. Is it to rederive
the well-known quantum-geometric formula for $m_k$ ?

Our response: We have slightly changed the presentation. The purpose of appendix B is to prove equation (A7), which we will require in obtaining equation (A9).

The referee says: If so, you should
write down clearly that formula somewhere (possibly in Eq.(5), as
suggested mentioned). You should also explain why there is the need
for presenting a new derivation of a rather well known result. More
specific comments about Appendix B can be found below.

Our response: This derivation is indeed not new. Parts of the derivation (of Equation (A7)) are scattered across many papers, and some of them make use of a lot of properties of the quantity $M_{mu nu}$ without deriving them. However, we could not find a complete derivation of equation (A7) anywhere and so decided to compile the derivation in a single place for the convenience of the readers.

The referee says: In writing Eq.(6), you specialize to $B=0$. I do not understand the
motivation for doing so, given that in the abstract you emphasize the
case where B is nonzero. Instead of relegating the $B \neq 0$ case to
footnote 5, what about replacing or supplementing Eq.(6) with an
equation in the main text that is valid for $B \neq 0$?

Our response: We have now replaced all these equations with those for non-zero B.

The referee says: Appendices A and B are called from footnote 6. I suggest moving that
footnote to the main text, somewhere below Eq.(6). Seems to me that it
is only Appendix A that should be called there, and that Appendix B is
only really invoked directly from Appendix A, not from the main text.

Our response: We have made this change

The referee says: The title of Sec. III is rather long, and it is unnecessarily repeated
in the titles of the subsections. Here are some suggestions for 
streamling the titles:
III. Charge and energy magnetizations from the Einstein relation
III.A Charge magnetization
III.B Energy magnetization

Our response: We have made this change

The referee says: The opening paragraph of Sec. III.A contains two almost identica l
sentences, namely the first and the fourth:
"In this section we demonstrate that the known expression of the 
charge magnetization can be recovered by demanding that the Einstein 
relation holds for the electric transport current."
"Here we show that the same expression of the charge magnetization can 
be obtained by demanding that the Einstein relation holds for the 
electric transport current."

Our response: We have now reworded the opening paragraph to so that there is only one sentence with the relevant point.

The referee says: Above Eq.(8), the authors refer to Appendix C for its derivation. That 
appendix is unnecessarily long, with many equations that already
appear elsewhere in the manuscript. Once that redundancy is removed, 
the appendix becomes quite short, and it would improve the readibility
 of the manuscript if it was brought to the main text.

Concerning Appendix C itself:
The first equality in Eq.(C1) is identical to Eq.(A4), which is in 
turn identical to Eq.(6a).
Eq. (C2) is the same as Eq.(7).
The first equality in Eq.(C3) is the same as the inline equation below
 Eq.(7).
The closing sentence of Appendix C should end with something like
", which gives Eq.(9)".
With these changes Appendix C can be made quite short, and inserted in 
the main text of Sec. III.A.

Incidentally, the title of Appendix C has "derivation of the"
appearing twice, which is awkward.

Our response: This appendix has been removed, and the necessary equations have been included in the section III A of the main text instead

The referee says: Immediately below Fig. 2, the long inline equation should probably be 
a numbered equation. In the following sentence, "the known expressions
[56,63,64] for the charge magnetization" refers, I presume, to
 Eq.(9). If that is the case, the sentence should refer to that
 equation directly instead of referring to the literature (which has
 already been cited in connection with that equation).

Our response: We have made this change

The referee says: On the 2nd column of p. 5, the second line contains "is
 appears". Remove "is"?

Our response: We have made this change

The referee says: The long paragraph where the above typo occurs contains a very large
number of parenthetical remarks, and does not read well.

Our response: We have removed a lot of the parenthetical remarks so that the paragraph reads better now.

The referee says: The two-sentence paragraph below the one discussed above mentions
"this quantity". Which quantity? The one given by Eq.(15)? Please 
specify.

Our response: We have now clarified this and removed the reference to "this quantity"

The referee says: Similar problem with the short paragraph below the one containing
Eq.(23). Therein, "this term" should be specified clearly. Presumably,
it is the second term on the left-hand side of Eq.(23).

Our response: We have clarified this as well.

The referee says: (Incidentally, the acronym LHS is used without having been defined.)

Our response: We have now expanded the acronym the first time it appears

The referee says: The sentence straddling pages 5 and 6 could be improved. As written,
it sounds as if the results obtained from the different methods that
 are listed were different from one another. Instead, I believe the
 authors want to say that Eq.(16) agrees with the result obtained in
 the previous literature in different ways: using gauge theories of
 gravity, or by introducing either an inhomogeneous disorder field or a
 gravitomagnetic field.

Our response: We have reworded the sentence fo read “This expression agrees with the results obtained in
the previous scientific literature using different methods,
namely, gauge theories of gravity [69], introduction
of an inhomogeneous disorder field [58], and introduction
of a gravitomagnetic field [61]”.

The referee says: In footnote 11, a quantity $\omega$
appears without being defined. Is it
the same as the cyclotron frequency $\omega_c$
in footnote 12? Or is it
the frequency of an applied ac electric field?

Our response: We have replaced all occurrences of $\omega_c$ with $\omega$

The referee says: Concerning the long paragraph on the 2nd column of p.5, where Eq.(24)
for the nonequilibrium distribution function is discussed. As already
mentioned, it would be important to relate Eq.(24) to similar
expressions that are given in the literature. At present, that long
paragraph makes almost no contact with prior work. And yet, there has
been a lot of activity recently in calculating nonequilibrium
 distribution functions in the presence of external fields and Berry
curvature. It would therefore be important to provide some
perspective on Eq.(24), by making contact with the literature. In
addition to the review paper indicated earlier, here are some other
papers that might be relevant:

https://doi.org/10.1103/PhysRevB.95.165135
https://doi.org/10.1103/PhysRevB.105.045421
https://doi.org/10.1103/PhysRevB.105.205126

Our response: We have cited these papers in the latest revision. We have explained that these works did not take the momentum dependence of the scattering timescale into effect, which our solution did. Also, in these works, the authors have Taylor expanded the factor $(1 + e B.\Omega/ \hbar)$ in the denominator, which is non-analytic at the Weyl/Dirac points. We have kept this term in the denominator for consistency, as the non-analytic nature of the Berry curvature may cause issues if we Taylor expand and try to calculate the effects order-by-order in B.

The referee says: The appendices need to be heavily edited, as they read more like
research notes than publication-grade material. Every equation should
be numbered, and redundancies should be identified and removed.

Our response: All equations have been numbered. Some of the appendices have been absorbed into the main tex

The referee says: In particular, the equation below (A2) defining the quantity $M_{ij}$

needs to be numbered and referred to later on. It is repeated in the
title of subsection 1 of Appendix B, and in fact that quantity 
features prominently throughout Appendix B.
(I am also unsure if the quantity $M_{\mu \nu}$
introduced in the 
first sentence of Appendix B.1.d the same as $M_{ij}$.)

Our response: We have made this change and removed the redundancy.

The referee says: Several of the titles of appendices, or subsections thereof, are quite 
long and contain long equations. See, for example, Appendices B.1.c
and B.1.d.

Our response: We understand that the titles of some of the appendices and their subsections might seem long. However, we have decided to retain their titles in the revised submission as well since we believe that the titles describe the content of the appendices well and further their length does not interfere with the flow of the main part of the paper.

The referee says: Above Eq.(A4) [which, as already mentioned, is the same as Eq.(6a)] it
is written: "We want to calculate the properties of the anomalous Hall
effect, where B=0
." This seems to contradict what is stated in the
abstract, where the focus seems to be on the B≠0
case.

Our response: We have rewritten all the relevant equations for non-zero magnetic fields.

The referee says: Specific comments about Appendix B:
The (unnumbered) multiline equation on the bottom of p.10 is just the
well-known Hellmann-Feynman theorem, which in the present context
 gives the intraband velocity. It may not be necessary to spell out its
 derivation in such detail.
Once the long equation on top of p. 11 is numbered, it does not have
to be repeated as an inline equation on top of p. 13 (in the sentence
"Integrating over [...]"), but simply referred to.
On p.13, the sentence starting with "In this case [...]" contains
 seven inline equations, which make it quite unwieldy. There is also an
"is" missing before the first occurrence of "Hermitian".
The second half of p.12 is quite confusing: the "quantity without the
commutator" is introduced, and it would probably help to associate a
symbol with it. Then there are several equations that start with an
 equal sign, and one could insert that symbol on the (nonexisting)
left-hand side of those equations. When we get to the first boxed 
equation on p.13, are we still talking about that quantity without a 
symbol? If so, the proposed symbol could be used there once again.
I would think that the end result of this entire appendix should be
 the quantum-geometric expression for the intrinsic orbital
 moment. However, that equation is never written down explicitly.

Our response: We have made some changes along the lines suggested by the referee to make the appendix more readable. The end result of the appendix is $M_{\mu\nu} = m_\alpha \epsilon_{\alpha\mu\nu}$

The referee says: Below Eq.(F1), there is a
"(see Appendix ?? for justification)"
Please insert the correct appendix number in place of "??"

Our response: We have made this change.

---

## Round 2 · Author Response

Dear Editor,

We thank you for sending our manuscript out for review and also thank both referees for their careful reading of the manuscript. Referee 1 says that our paper provides a “fresh perspective” on the derivation of the energy magnetization even though the expression we derive has been obtained before. They contend that this limits the impact of our work. Further, they say that we have not performed a calculation of the additional Hall response arising from the non-equilibrium distribution function that we have derived for any model system and also argue that such an additional response, in fact, does not exist. Referee 2’s criticisms are also along similar lines, i.e. we do not perform a calculation for a model system. They thus find that our paper does not meet the standards of Scipost Phys. and referee 2 recommends it be considered for Scipost Phys. Core instead, which also appears toe the current editorial recommendation. We respectfully disagree with the decision and provide detailed point by point responses below to the comments, questions and criticisms of both referees. Briefly (we have provided elaborations in our detailed response), the reasons for our belief in the suitability of our work for publication in Scipost Phys. and not Scipost Phys. Core are

  1. While it is true that the expression for the energy magnetization is not new, our deriva- tion relies on general considerations related to the Einstein relations rather the specifics of microscopic systems. We believe that this makes the importance of the energy mag- netization, which is a more opaque quantity than its charge counterpart, easier to com- prehend compared to existing treatments in the literature. This more than compensates for the fact that the expression we have obtained has been derived before in other ways.

  2. Our treatment is also not meant to be specific to a particular system (like bilayer graphene) or indeed even to a particular transport quantity like the Hall response. Instead, the non-equilibrium distribution function for systems with Ω ̸= 0 and B ̸= 0 can be used generally to obtain all charge and heat transport coefficients. In spirit, our treatment is similar to that in Refs. 58 and 62, which also do not focus on microscopics. The Hall conductivity is specifically mentioned at a few points in the manuscript since it is the most studied of all transport coefficients. However, we have made it clear that all transport coefficients are on the same footing for us. A derivation of all such coefficients would be the subject of an extensive and self-contained calculation in itself and well beyond the scope of what we are trying to achieve here. We believe the expression in Eqn. 24 of the non-equilibirum distribution function is sufficiently novel to be of interest by itself. Similarly, a calculation for a specific microscopic system like bilayer graphene is also fairly involved (as we have commented in the modified manuscript) and a subject of a separate detailed calculation in itself.

  3. Referee 1’s contention that the additional Hall response that arises from our calculation does not exist appears to be based on the expectation that it is going to be linear in the magnetic field. We have not made this claim anywhere and the response we obtain is, in fact non-linear, and very much present in the systems our calculations apply to. There is thus, as far as we can tell, no issue about the correctness of our work.

  4. Our manuscript contains two different important contributions: 1) A new derivation of an existing result related to a rather opaque physical quantity based on simple and very general arguments and 2) An expression for the non-equilibrium distribution function for systems with a non-zero Berry curvature and in a non-zero magnetic field, which is a new result.

We are resubmitting our manuscript with modifications in response to the referees’ com- ments and suggestions. We hope that it can still be considered for publication in Scipost Phy. in light of the points that we have made above.

Thank you very much.

Best regards,

Archisman Panigrahi Subroto Mukerjee

Response to referee 1:

We thank the referee for their very careful reading of our manuscript and for highlighting the strengths and weaknesses in it along with providing several suggestions for its improve- ments. Please find below our response to the referee’s comments, suggestions and criticisms.

The referee says: Strengths 1. Proposes an alternative way of deriving the expressions for the charge and energy mag- netization of Bloch electrons [Eqs. (8) and (14) in the manuscript]. 2. A useful and detailed Introduction, with abundant references to previous works. 3. Detailed derivations are given in the Appendices. See however point 4 in the Weaknesses section.

Our response: We thank the referee for highlighting the strengths of our manuscript.

The referee says: Weaknesses

Our response: We thank the referee for also pointing out the weaknesses in our manuscript. However, as we explain below, we respectfully disagree with the referee that these are necessarily weaknesses.

The referee says: 1. A significant part of the manuscript is devoted to rederiving in a new way results that were previously obtained in the literature, namely Eqs. (8) and (14).

Our response: It is indeed true that the expression for the energy magnetization has been derived before in the literature, a fact that we did acknowledge in our manuscript. The utility of our work is that it provides a derivation that is perhaps more straightforward to understand and adapt to other situations, based as it is on general considerations of the Einstein relation rather than specific microscopics.

The referee says: 2. The authors claim to have identified a new type of regular Hall response in time reversal invariant systems coming from the Berry curvature and from the nonequilibrium part of the distribution function given by Eq.(18). However, an actual expression for that Hall response is never written down. As detailed in the Report, I believe that such a contribution to the regular Hall response cannot exist.

Our response: As we have clearly mentioned in the manuscript, we have derived an expression for the non-equilibrium distribution function in the presence of both a non-zero Berry curvature and non-zero magnetic field. This expression can be used to obtain all charge and heat transport coefficients, in linear response in temperature and potential gradients and not just the Hall response. The terms involving the scalar product of the field and Berry curvature, i.e. B.Ω provide contributions to these coefficients (including the Hall conductivity) in addition to those already derived in the literature in the presence of either B or Ω alone. The scope of the current work is limited to deriving only the expression for the non-equilibrium distribution function and not providing detailed expressions for all the transport coefficients. Nevertheless, the reason we singled out the Hall response for mention is that it is the most widely studied of the transport coefficients in topological systems. Thus, the additional contributions to transport that we obtain in our work are perhaps best highlighted using the Hall response as an example. The referee’s contention that the additional contribution to the regular Hall response cannot exist is perhaps due to the expectation that the term “Hall response” should only be used to describe an effect that is linear in the magnetic field. If that is the case, we agree. However, we have not claimed anywhere that the additional terms arising from our calculations are linear in the magnetic field (since they are not) and have thus also assiduously avoided any mention of the Hall resistance, as opposed to the Hall response since the former term is usually employed in situations where the response is linear in the field.

The referee says: 3. The formal results obtained in the manuscript are not illustrated by any explicit calculations on model systems.

Our response: Our intention here is to provide a treatment of transport and diamagnetic currents in systems with both B ̸= 0 and Ω ̸= 0 in the most general terms that would be applicable to any system rather than focus on specific systems. In this regard, it is very similar in scope to previous papers on topological systems such as Refs. 58 and 62, which too did not perform calculations on model systems. We thus believe our work will also be a valuable contribution to the broad field like those papers.

The referee says: 4. The main text is not sufficiently self contained. I had to constantly flip back and forth between the main text and the numerous appendices (as well as the numerous footnotes) to keep up with the manuscript.

Our response: We have followed the referee’s very useful suggestions for improvement of the presenta tion of the manuscript and made modifications accordingly.

The referee says: Report The authors have shown that requiring the transport charge and energy currents to obey the Einstein relation provides an alternative route for deriving the known expressions for the charge and energy magnetization of Bloch electrons at finite temperature [Eqs.(8) and (14) in the manuscript]. This approach complements the standard approach [e.g., Ref. 58] where one starts from those expressions for the magnetizations, subtracts their curl from the expressions for the local (or total) current densities to obtain the transport current densities, and then verifies a posteriori that the resulting transport coefficients satisfy the Einstein relation. The new approach in this manuscript provides a fresh perspective. But that fact that it is only used to recover previously known results limits its impact somewhat.

Our response: We are happy to note that the referee feels that our approach provides a fresh perspective. As we have also mentioned above, we do not claim that the result obtained is entirely new. Rather, it is the approach based on a general application of the Einstein relation as opposed to an appeal to microscopics for specific models that is the focus of our work.

The referee says: In addition to the new derivation of Eqs.(8) and (14), the other main result of the manuscript is the expression in Eq.(18), valid for 2D systems, of the correction to the distribu- tion function to linear order in the electric field, chemical potential gradient, and temperature gradient. In previous works, the emphasis had been on the intrinsic part of the response in ferromagnetic systems, associated with the equilibrium distribution function. The authors claim that plugging Eq.(18) into the expressions in Eqs.(15a,15b) for the transport currents yields a regular Hall response in systems like bilayer graphene, which possess a nonzero Berry curvature due to broken inversion symmetry, but display no anomalous Hall effect due to time reversal invariance.

Our response: We thank the referee for this summary of the significance of our work. As they say, previous calculations have indeed focused on systems with a non-zero Ω due to broken time reversal but no magnetic field. Our calculation applies to general situations with non-zero Ω and non-zero B. In particular, as the referee mentions, it is of interest for systems with non-zero Ω due to broken inversion and not time reversal which require a magnetic field to produce off-diagonal transport responses.

The referee says: However, the expression for that Hall response is never written down explicitly. From which of the various terms in Eq.(18) does it arise? Is it from the first term which is linear in tau, or from the second or third term, both of which are quadratic in tau? I don’t quite see how such a Hall response will come about. For time reversal invariant systems, it follows from the Onsager relation that the Hall response is odd in B, see https://doi.org/10.1016/0038-1098(65)90178-X On the other hand, for time reversal invariant systems one can easily show that terms in the magnetoconductivity with odd powers of B have even powers of the relaxation time tau. See, e.g., below Eq.(9) in https://doi.org/10.1103/PhysRevB.105.045421 This seems to rule out the first term in Eq.(18) as a candidate for a Hall response in time reversal invariant systems, since it it is linear (odd) in tau. The other two terms in Eq.(18) are quadratic in tau, but at linear order in B they are independent of the Berry curvature: the Berry curvature only contributes at secondorder in Bv iathe energydenominators 1+(e/hbar)B⃗.Ω⃗,since there is already a factor of B in the numerator of each of those terms. The above considerations seem to rule out any regular (linear in B) Hall response in time reversal invariant systems coming from the Berry-curvature and from the nonequilibrium distribution function.

Our response: We thank the referee for taking the time to carefully analyze the expression we have obtained term by term to determine its contribution to the Hall response. As we have mentioned earlier, we are not using the term “Hall response” to mean a response that is linear in B, which is the sense in which the referee is presumably using it. Hence, their conclusion that our expression produces no Hall response. We have clearly indicated in the new version of the manuscript that the term “Hall response” does not imply linear in B response. We have also commented on the contribution of each of the different terms that appears in the expression for the non-equilibrium distribution function to the Hall response.

The referee says: Note that there is actually a regular Hall response in time reversal invariant systems coming from the product of the Berry curvature with the orbital moment. It is however associated with the equilibrium part of the distribution function, not with the nonequilibrium part. That type of response was first identified in http://dx.doi.org/10.1103/PhysRevLett.112.166601 [see the second term in Eq.(12) therein], and is also discussed in https://doi.org/10.1103/PhysRevB.103.125432 That response is zeroth order in tau and linear in B, and so it complies with the above- mentioned constraints for Hall responses in time reversal invariant systems. The authors appear to have missed this contribution in their analysis.

Our response: We thank the referee for bringing up this point. The intrinsic equilibrium contribution the Hall response is present in our analysis through the second term of Eqn. 17a. We had not highlighted this fact earlier since the intrinsic equilibrium Hall response has been a subject of detailed investigation already as the referee points out and we did not have anything to add to what is known. Our focus, as mentioned earlier, is on the contribution to transport arising from the non-equilibrium distribution function. However, in the interest of clarity, we have now explicitly mentioned exactly how our calculation also includes the equilibrium response after En. 19.

The referee says: Overall, I feel that this submission does not quite meet the strict acceptance criteria of SciPost Physics. However, I would recommend publication in SciPost Physics Core once the manuscript has been revised taking this report into account.

Our response: We respectfully disagree that our work does not meet the criteria for publication in Sci- Post Physics. We hope we have managed to convey from our responses above that both of the important results of our work are of significance to the field of transport in topologi- cal systems. And that our results are correct and produce the already studied equilibrium Hall response in addition to the hitherto unexplored non-equilibrium response for Ω ̸= 0 and B ̸= 0. To recap the significance of our work: The energy magnetization is a rather opaque quantity compared to its charge counterpart. We feel that our derivation based on general considerations related to the Einstein relation simplifies its understanding more than those that appeal to microscopics and currently exist in the literature. Further, our expression for the non-equilibrium distribution function in systems with Ω ̸= 0 and B ̸= 0 is a new result, which can be used to obtain any charge or heat transport coefficient. We have clarified that this distribution function does indeed provide a Hall response, just not one that is linear in B, and our calculation also contains the intrinsic equilibrium Hall response. These seem to have been the referee’s main concerns about the validity of our results and we believe that we have addressed them.

The referee says: Requested Changes Concerning the physics content of the manuscript, my main request would be to revise the discussion of the types of responses (Hall vs Ohmic) that arise from the equilibrium and non-equilibrium parts of the distribution function, taking into account the above comments. Concerning the presentation, I wonder if it would make more sense to put the paragraph containing Eq.(14) immediately after Eq.(11). The paragraphs containing Eqs.(12,13) and Fig. 2 could then be placed in a separate subsection III.C, where the physical interpretation for Chern insulators is discussed. Minor suggestions • The wording of the sentence ”We further obtain [...]” in the abstract suggests that the expression obtained for the energy magnetization is new. In view of the references given below Eq.(14), that sentence should probably be revised. The same comment applies to the corresponding sentences in the Conclusions section. • The inline equation below Eq.(F2) is the same as Eq.(8), no need to write it explicitly.

Our response: We thank the referee for these suggestions and have now incorporated them in the manuscript.

The referee says: Suggestions on the formatting of the manuscript: 1. The authors should consider numbering all equations that are not in-line. 2. The manuscript contains a large number of footnotes, which are not readily identified by the reader as such since they are placed in the references. It would be helpful if the footnotes were placed on the page where they are called from, and if their overall number was reduced. 3. A related issue is that several of the Appendices are only mentioned in footnotes. It would be better if the Appendices were called from the main text. 4. The number of Appendices is quite large. For example, I wonder if the two very short appendices A and H could be disposed of, and their contents inserted at the appropriate places in the main text. 5. Some Appendices have a high concentration of in-line equations, sometime with several of them in the same paragraph: see top of p. 12, beginning of Appendix G, Appendix I.1, and below Eq.(I1). This is a matter of personal taste, but I find this format hard to read. 6. In Appendix D, the notation with ”all space” or ”cell” on both sides of the inner products is somewhat cumbersome. Maybe place them on the right-hand side only?

Our response: We thank the referee for these formatting suggestions and have now incorporated them in the manuscript. Please see the list of changes for details.

The referee says: Typos 1. In the last paragraph of the 1st column of p. 2 , it is written ”the the validity” . 2. In the first two lines below Eq. (3), remove the band indices from the cell-periodic Bloch states. Same for Eq. (C1) and the in-line equations below it. 3. At the beginning of Sec. III.B, ”The expression in Eq.(6)” should read ”The expression in Eq.(6b)”. 4. In footnote 79, one reference appears (twice) as a question mark. 7

Our response: We thank the referee for pointing out these typos and have now corrected them and a few others that we found upon rereading the manuscript.

                                                                                                                                  Response to referee 2

We thank the referee for their very careful reading of our manuscript and for highlighting the strengths and weaknesses in it along with providing several suggestions for its improvements. Please find below our response to the referee’s comments, suggestions and criticisms.

The referee says Strengths The strength of the manuscript lies on the detailed derivation of the charge and energy magnetization of Bloch electrons using an alternative approach based on Einstein relation.

Our response: We thank the referee for highlighting the strengths of our manuscript.

The referee says: Weaknesses

Our response: We thank the referee for also pointing out the weaknesses in our manuscript. However, as we explain below, we respectfully disagree with the referee that these are necessarily weaknesses.

The referee says: 1. Although the authors provide detailed derivation of charge and energy magnetization, they have not applied their formalism to any particular system.

Our response: As we have also mentioned in our response to a similar comment by referee 1, our intention here is to provide a treatment of transport and diamagnetic currents in systems with both B ̸= 0 and Ω ̸= 0 in the most general terms that would be applicable to any system rather than focus on specific systems. In this regard, it is very similar in scope to previous papers on topological systems such as Refs. 58 and 62, which too did not perform calculations on model systems. We thus believe our paper will also be a valuable contribution to the broad field like those other papers.

The referee says: 2. They do not provide the explicit expression of Hall conductivity (which is one of their main results) in the main text.

Our response: This point too has been brought up by referee 1. As we have clearly mentioned in the manuscript, we have derived an expression for the non-equilibrium distribution function in the presence of both a non-zero Berry curvature and non-zero magnetic field. This expression can be used to obtain all charge and heat transport coefficients, in linear response in temperature and potential gradients and not just the Hall response. The terms involving the scalar product of the field and Berry curvature, i.e. B.Ω provide contributions to these coefficients (including the Hall conductivity) in addition to those already derived in the literature in the presence of either B or Ω alone. The scope of the current work is limited to deriving only the expression for the non-equilibrium distribution function and not providing detailed expressions for all the transport coef-ficients. Nevertheless, the reason we singled out the Hall response for mention is that it is the most widely studied of the transport coefficients in topological systems and so the existence of the additional contributions to transport in general is perhaps best highlighted using the Hall response as an example.

The referee says: Report The authors provide an alternative approach (based on the Einstein relation) to derive charge and energy magnetization. They use the approach only to recover old results. There- fore, I feel there are not many new results in the manuscript. Another point is that the authors did not compare their approach with the old one (why one should use their approach to derive charge and energy magnetization?).

Our response: We thank the referee for their comments. As we have already emphasized, our approach is based on general considerations related to the Einstein relations rather than based on specifics of the microscopics of model systems. The energy magnetization is a rather opaque quantity compared to its charge counterpart. We feel that our derivation based on the above general considerations simplifies its understanding compared to those that appeal to microscopics, which currently exist in the literature. We have added a line to this effect in the modified manuscript.

The referee says: The authors claim that the non-equilibrium distribution function obtained in this work can generate a new type of regular Hall response in time-reversal invariant samples with a non-zero Berry curvature. Although the authors have presented detailed derivation for the rest of the calculations, however, they did not provide the expression of Hall conductivity anywhere. Regarding that I have few questions

Our response: Please find below the responses to the questions brought up by the referee.

The referee says: 1. Please write down the explicit expression of the different components of the Hall conductivity.

Our response: As we have mentioned earlier, the Hall response is not the focus of this work. Rather, it is the expression for the non-equilibrium distribution function in the presence of both Ω ̸= 0 and B ̸= 0 from which, in principle, one can derive all the heat and charge trans- port coefficients by integrating with appropriate kernels. The Hall response is non-linear in the magnetic field (as also pointed out in our response to referee 1) and thus does not have a simple closed form expression. Nevertheless, we have added a discussion after Eqn. 24 on page 7 explaining the effect of each of the terms to Hall response.

The referee says: 2. How one can distinguish different Hall components from each other in experiment?

Our response: The different components of the Hall response contribute with different functional dependences on the magnetic field. Since the field in the non-equilibrium distribution functions appears only through the combination e Ω(k).B, with the momentum k be- ing integrated over, the functional dependence on B will be determined by the precise dependence of Ω(k) on k. There is thus no simple way to parse the contributions experimentally without doing an explicit calculation for a specific microscopic system. However, a general feature of the field dependence coming from the presence of the term e Ω(k).B is that it is non-linear in B as opposed to the Hall response in a regular metal 􏰀 or even Chern insulator, in which it is linear. The deviation from linearity in the field could be an experimental signature of the contribution from the non-equilibrium distribution function that we have derived.

The referee says: 3. What is τ scaling for the new regular Hall effect? What are symmetry constraints for this Hall conductivity? Our response: In the discussion added after Eqn. 24, we have commented on the τ dependence of the different terms in the expression for the Hall conductivity. We are not sure what the referee means by “symmetry constraints” for this Hall conductivity but it obeys the usual Onsager symmetry relations like the Hall conductivity in any other situation.

The referee says: 4. The authors should calculate the new regular Hall conductivity for bilayer graphene system and then compare with other coexisting Hall components.

Our response: As we have emphasized before, the Hall conductivity is not the focus of this work. It is the calculation of the non-equilibrium distribution function from which all charge and energy transport coefficients can, in principle, be obtained. The Hall conductivity is just one of these, even if it is the most commonly studied one in the literature. A calculation of only the Hall conductivity detracts from the significance of the distribution function to the calculation of all transport coefficients. At the same time, a calculation of all the transport coefficients is too involved to be within the scope of what we are trying to achieve here. Also, as mentioned before, our work develops a general formalism without focusing on any specific microscopic system along the lines of other works such as Refs. 58 and 62. We mention bilayer graphene since it is an example of a system in which Ω ̸= 0 due to broken inversion and not time reversal symmetry and thus is of the type that our formalism might be useful to study. However, there could be other such systems and so we do not think that studying this specific system as an example is necessary. This is more so because the calculation for Dirac materials can be technically quite involved due to the singular behavior of the Berry curvature, as we have pointed out in the discussion added after Eqn. 24. It should thus be the subject of a self-contained investigation of its own well beyond the scope of our general treatment here.

The referee says: 5. The authors consider relaxation time approximation. Please discuss the regime of validity of this approximation.

Our response: The relaxation time approximation employed here has the same regime of validity as in all other such studies based on a non-interacting description of topological systems. The main assumption is that transport can be described in terms of non-interacting electrons or quasiparticles with momentum relaxation in the bulk due to interactions with other degrees of freedom like phonons and impurities. The approximation breaks down in the so-called “ hydrodynamic regime” in which the electrons collectively behave as a fluid with bulk momentum conservation. This regime is usually accessible in a rather restricted region of electron density and temperature in which the time scale obtained from the shear viscosity of the electron fluid is shorter than that from interactions between the electrons and other degrees of freedom. The relaxation time approximation thus applies in a much broader regime of parameters.

The referee says: Requested changes Major changes 1. The authors should write down the explicit expression of different Hall components using their formalism and explain how they can be distinguished from each other in experiment. 2. They should calculate the new regular Hall response for the bilayer graphene system.

Our response: We respectfully disagree with the referee that either of these changes is required since, as we have emphasized above, our work is about the derivation of the non-equilibrium distribution function from which all transport coefficients can be derived and not just the Hall response. Further, our formalism is also meant to be general in scope and not focused on any particular system. We have added a short discussion after Eqn. 24 on why the calculation of the Hall conductivity for specific Dirac systems can be technically involved. We thus think it should be the subject of a self-contained investigation of its own and it is not very reasonable to expect us to fit it within the scope of our general treatment here.

The referee says: Minor changes 1. Appendix A and Appendix J can be included in the main text. 2. Appendix B is very well known. So the authors can remove it from the manuscript. 3. There are several typos and grammatical errors in the manuscript which authors should fix in the revised version. 4. Some of the equations in the main text do not align properly. The authors should consider them to align properly within the margin.

Our response: We thank the referee for suggesting these minor changes, which we have now incorporated in the manuscript.

---

## Round 2 · List of Changes

List of Changes 1. Fixed typos as referee 1 suggested 2. Updated the abstract and the conclusion as per the suggestion of referee 2 3. Added a line highlighting the usefulness of our approach involving the Einstein relation to the derivation of the energy magnetization on page 2. 4. Appendix A has been converted into a footnote 5. Appendix B has been removed 6. Appendix H is now included within the main text. 7. Appendix J is included within the main text. 8. Aligned the contents of several equations in the main text, and fixed equations flowing to the margin 9. Switched the position of the second and the third term in the current Equation 24. 10. The double subscripts ‘cell’ and ‘all space’ are replaced with single subscripts. 11. All equations in the main text that are not in-line have been numbered. 12. The regime of validity of the Boltzmann transport equation is discussed in page 6. 13. The regular Hall response due to the equilibrium part of the distribution function, as discussed in http://dx.doi.org/10.1103/PhysRevLett.112.166601 and https://doi.org/10.1103/PhysRevB.103.125432 was already present in our formalism. We have explicitly shown it in page 6. 14. The role of each term in the solution of the Boltzmann transport equation, and their dependence on the scattering timescale are discussed in detail in page 7.

---

## Round 3 · Referee Report · Anonymous (Referee 1) · 2023-5-15

Report

I thank the authors for having made extensive changes to the manuscript in response to my comments and suggestions, and for the detailed and thoughtful responses to the report.

The revised manuscript has improved significantly in terms of clarity of presentation and overall polish. The novel aspect of the work, and its relation to the previous literature are also much clearer now.

I believe that in its present form the manuscript is a valuable addition to the literature, and I fully recommend its publication in SciPost Physics Core.

Here are a few typos to be corrected:

  • Bottom of p. 2, 1st column: "Note that there is no fundamentally change the regular Hall respose." Please rephrase.

  • Page 3, last sentence of Sec. II: "energy magnetization $m_k$" $\rightarrow$ "orbital magnetization $m_k$"?

  • Below Eq. (15): "there are more than one bands" $\rightarrow$ "there is more than one band"

---

## Round 3 · Author Response

We thank the referee for going through our revised manuscript and submitting their second report. We appreciate the extreme care with which they have read our revised submission and their suggestions for the improvement of our presentation. We have incorporated most of these suggestions (Please see this list of changes). We apologise for the delay in resubmitting which was partly due to the extensive revisions we had to make upon the referee's suggestion. We think that the revisions significantly improve the readability of the manusctipt.

The referee says that our manuscript does not meet the stringent novelty and clarity criterion for publication in SciPost Physics. We hope that by making the changes suggested by the referee, we have addressed the questions about clarity. As for novelty, the referee has pointed us to a review paper, which they claim contains a summary of previous work on the perturbative solution of the Berry-Boltzmann equations to high orders in applied fields and which we have not cited. The referee says that the existence of this paper detracts from our claim of the novelty of our approach. We thank the referee for bringing this paper to our notice, which we had indeed missed earlier. However, as we explain below, there are questions about the correctness of the expressions in this paper. Moreover, the approach described in it does not account for temperature gradients, the inclusion of which is an important aspect of our calculations. We thus believe that our approach is indeed novel within the scope of general transport theory.

A further criticism of the referee is that we do not perform a calculation for a specific microscopic model and this fact disqualifies our paper from consideration for publication on SciPost Phys. We reiterate that our intention is to present a calculation of the energy magnetization in the most general terms. And, as we have pointed out in our previous response, there have been several papers in the recent past, which too have presented general treatments of different aspects of transport theory without performing calculations for specific models. While we once again, respectfully disagree with the referee that this should be a disqualification for publication on SciPost Phys., we are willing to accept the referee's recommendation of publication in Sci Post Phys. core instead.

Please find below a detailed point by point response to the referee's comments. We hope that this version of the manuscript will be considered acceptable for publication in Sci Post Phys. Core.

Best regards,
Archisman Panigrahi
Subroto Mukerjee

---

## Round 3 · List of Changes

We have made the following changes to the manuscript in the resubmission. They have been indicated in red in the manuscript wherever possible.

  1. Added “in Chern insulators” to the abstract, as the referee suggested.

  2. Removed the duplicate footnotes containing “the charge of electron is taken to be −e”, and added that to the main text.

  3. Simplified the titles of the subsections of section III, as the referee suggested.

  4. The acronym LHS (left-hand side) is defined where the first time it is used.

  5. To denote the cyclotron frequency, the symbol ω is consistently used instead of ωc.

  6. The Appendix titled “Derivation of the transport electric current density and derivation of the charge magne- tization” (which was Appendix C in revision 2) is now removed, and its contents are added to section III A of the main text.

  7. The goals of all the appendices are mentioned at their beginning.

  8. All the inline equations in the appendices are converted to numbered equations.

  9. Some prior works related to the solution of Boltzmann Transport Equation (BTE) are cited (references 75, 76, 77, 78), and we have explained how our solution of the BTE addresses some problems which were not addressed previously.

  10. The expression εk = ε0(k) − mk · B is now modified to εk = ε0(k) − mk · B − ms · B, (Eq. (4)) where ms is the spin magnetic moment.

  11. All the factors of 2 arising from spin degeneracy are removed, and the corresponding quantities are written as sum over spins.

  12. In Appendix A and B, the components of the tensor M are rewritten using Greek indices, Mμν.

  13. What we mean by “regular Hall response” is now written in the second page of the main text, where it is mentioned for the first time.

  14. We have explained that the breaking of the inversion symmetry will not affect the spin degeneracy, as the spin-orbit coupling is assumed to be a negligible effect above Eqn. 12 on page 4.

  15. We have replaced all the equations (Eqs. (7a), (7b), (9), (10), (11), (14a), (15), (16), (19)) with those for non-zero magnetic field, as the referee suggested.

  16. We have obtained the formula for the charge and the energy magnetization at a non-zero B, and have demon- strated that they reproduce the previously known formula in the B → 0 limit.

  17. Several footnotes were removed, and their contents were absorbed into the main text.

  18. Appendix B is now invoked only within Appendix A, since the goal of Appendix B is to prove a certain result used in Appendix A.

  19. We have reworded the opening paragraph of section III A, so that it does not anymore contain two similar sentences.

  20. The “long inline equation immediately below figure 2” in the previous revision is now turned into a numbered equation (Eq. 17), as the referee suggested.

  21. In revision 2, the 2nd column in page 5 had a large number of parenthetical remarks, which are removed in this revision.

  22. Whenever we use the phrase “this term” or “this quantity”, where the sentence is not immediately after the corresponding “term” or “quantity”, the relevant equation has been mentioned to clarify which “term” or “quantity” we are referring to.

  23. We have added a brief discussion in the first paragraph on page 8 of what happens to transport coefficients when the Fermi energy approaches a Dirac (or Weyl) point along with two references (79 and 80), to address one of the comments of the referee.

  24. Several typos have been fixed.

---

## Editorial Decision

published